# Volcaniclastic density currents explain widespread and diverse seafloor impacts of the 2022 Hunga Volcano eruption

Sarah Seabrook [1] ✉, Kevin Mackay [1], Sally J. Watson [1,2], Michael A. Clare [3], James E. Hunt [3], Isobel A. Yeo[3], Emily M. Lane [4], Malcolm R. Clark [1], Richard Wysoczanski [1], Ashley A. Rowden[1,5], Taaniela Kula[6], Linn J. Hoffmann [7], Evelyn Armstrong[8] & Michael J. M. Williams [1]

The impacts of large terrestrial volcanic eruptions are apparent from satellite monitoring and direct observations. However, more than three quarters of all volcanic outputs worldwide lie submerged beneath the ocean, and the risks they pose to people, infrastructure, and benthic ecosystems remain poorly understood due to inaccessibility and a lack of detailed observations before and after eruptions. Here, comparing data acquired between 2015 - 2017 and 3 months after the January 2022 eruption of Hunga Volcano, we document the far-reaching and diverse impacts of one of the most explosive volcanic eruptions ever recorded. Almost 10 km$^3$ of seafloor material was removed during the eruption, most of which we conclude was redeposited within 20 km of the caldera by long run-out seafloor density currents. These powerful currents damaged seafloor cables over a length of >100 km, reshaped the seafloor, and caused mass-mortality of seafloor life. Biological (mega-epifaunal invertebrate) seafloor communities only survived the eruption where local topography provided a physical barrier to density currents (e.g., on nearby seamounts). While the longer-term consequences of such a large eruption for human, ecological and climatic systems are emerging, we expect that these previously-undocumented refugia will play a key role in longer-term ecosystem recovery.

Explosive volcanic eruptions can have global impacts, producing large eruption plumes and volcanic deposits, spawning powerful density currents, triggering far-reaching tsunamis, damaging infrastructure, and leading to loss of human life[1–3]. The ash and chemicals released by large volcanic eruptions are also a driving factor in global climate variability, and can affect biogeochemical cycles, ecosystem functioning, associated carbon cycling, and local biological communities[4–7]. Despite their significance, our understanding of the magnitude, extent and diversity of impacts of major eruptions is poor, due to a paucity of pre- and post-eruption observational data[8,9]. This knowledge gap is particularly acute for submarine volcanoes, where eruptions are often undetected or underreported[10]. Despite

[1]National Institute of Water and Atmospheric Research, Wellington, Aotearoa, New Zealand. [2]Institute of Marine Sciences, University of Auckland, Auckland, Aotearoa, New Zealand. [3]Ocean BioGeosciences, National Oceanography Centre, European Way, Southampton SO14 3ZH, UK. [4]National Institute of Water and Atmospheric Research, Christchurch, Aotearoa, New Zealand. [5]Victoria University of Wellington, Wellington, Aotearoa, New Zealand. [6]Natural Resources Division/Tonga Geological Services, P.O. Box 5 Nuku'alofa, Tonga. [7]Department of Botany, University of Otago, PO Box 56 Dunedin, Aotearoa, New Zealand. [8]Department of Marine Science, NIWA/University of Otago Research Centre for Oceanography, University of Otago, Dunedin, Aotearoa, New Zealand. ✉ e-mail: sarah.seabrook@niwa.co.nz

amounting to >75% of all magmatic outputs worldwide, submarine volcanic eruptions account for only 8% of those recorded[11–13]. This observational bias is severe in regions under-represented by scientific study, such as the South Pacific Ocean where the majority of the world's 100,000 uncharted underwater volcanoes of >1 km in elevation lie[14]. In particular, shallow-water, active volcanic centres that are near populated islands expose a major blind spot regarding risk assessment and response preparedness[15,16].

This lack of awareness was illustrated by the highly explosive eruption of the partially-submerged Hunga Tonga-Hunga Ha'apai (hereafter referred to as Hunga) Volcano on 15 January 2022. With a Volcanic Explosivity Index of 5.7, the 15 January eruption was the largest eruption since Mount Pinatubo (Philippines) in 1991 and is one of the most explosive submarine eruptions ever recorded. The Hunga Volcano eruption had global impacts, yet came with little warning[8]. The eruption generated a steam rich eruption plume (a cloud of ash and tephra suspended in gases) that reached the lower mesosphere[17]. Near-term impacts included atmospheric shockwaves, a tsunami that crossed the Pacific Ocean, loss of human life, and damage to seafloor cables that severed telecommunications for the Kingdom of Tonga at a critical time for disaster response[18–21].

The Hunga Volcano eruption highlights the potential risks posed by the 22 mapped volcanoes in the Kingdom of Tonga, hundreds more along the Tonga-Tofua-Kermadec Arc, and many others worldwide. Most submerged volcanoes are poorly mapped, and data from surveys before and after eruptions that can be used to quantify pre- and post-event seafloor conditions are rare[22–25]. No such data previously existed for an event on the scale of the 2022 Hunga Volcano eruption. This data paucity means that the full range of seafloor impacts of large eruptions remain enigmatic. Here, we address this knowledge gap, integrating a multi-disciplinary dataset acquired just 3 months after the 2022 Hunga Volcano eruption[26] with data acquired prior to the eruption[27,28]. Combining these datasets (including multibeam echosounder, geological data, video footage and water column samples), we provide an overview of the wide-reaching ocean impacts of such a large eruption, ranging from topographic changes and damage to critical infrastructure, to widespread loss of seafloor life.

Using seafloor mapping surveys, we show that almost 10 km³ of seafloor material was removed following the eruption and document the fate of the material, providing a first budget for seafloor loss/gain after the eruption. Through integration of seafloor mapping, sediment coring, geochemical analyses, and numerical modelling we find the extensive reshaping of the seafloor was caused by long run-out (>100 km) volcaniclastic density currents. Pre-existing seafloor relief funnelled these flows, explaining spatially variable trends in erosion and deposition. Using seafloor video surveys, we document an apparent widespread loss of biological communities, due to physical disruption or smothering of seafloor habitats, and discuss the broader implications of major submarine eruptions for ecosystem structure, function, and recovery.

## Results and discussion
### Major but localised seafloor reshaping resulting from the 2022 Hunga Volcano eruption
The islands of Hunga Tonga and Hunga Ha'apai are remnants of the rim of a roughly 3- to 4-km wide submarine caldera of Hunga Volcano, with eruptions noted in various parts of the caldera throughout the past century. The most recent eruption series (in 2015) formed a 120 m high and 2 km wide near-circular tephra cone which joined the two islands together[18,27,28]. After this eruption, Hunga Volcano was largely inactive until it began erupting in late 2021, producing a steam-rich gas and ash plume[17,18]. This eruption culminated with the explosive events on the 15 January 2022[29]. Satellite imagery on 15 January, prior to the largest eruptive episode, showed the islands had been disconnected by the eruption sometime before this[29]. After the 15 January eruption, only

small remnants of the islands of Hunga Tonga and Hunga Ha'apai remained above water (Fig. S5). However, the modification of the submerged fraction of the volcanic edifice (~99% of Hunga Volcano) remained unknown.

A map of bathymetric change (derived by subtracting new bathymetric data from pre-eruption surveys between 2015 - 2017) reveal the seafloor morphologic fingerprint of a major eruption. Significant change occurred on the summit, where an increase in depth >800 m was observed in the caldera (Fig. 1). Integrating depth changes over selected areas of the map (see Methods), this change accounted for at least 6.0 km³ seafloor loss from within the caldera (dense rock volume), equivalent to 20 times the eruptive volume of Mount St Helens (USA) in 1980[30]. An additional 3.5 km³ was lost from the outer flanks, without impacting the general morphology of Hunga Volcano (Fig. 1). Many small features present pre-eruption, such as peaks, ridges and gullies, were visible in the post-eruption bathymetry. The deposited volume we report is an uncompacted volume, while the erupted volume was dense rock; hence, direct comparison of these volumes is complicated due to their differing bulk density. Overall, 6.3 km³ of uncompacted material was deposited within 20 km of the caldera rim. Prior studies suggest that ~1.9 km³ of material was ejected into the atmosphere as an eruption plume[31]. The remaining material that is unaccounted for was likely deposited as widespread thin deposits below the detection limit of what can be resolved with ship-based echosounders. This notion is supported by sediment coring that sampled volcanic material linked to the Hunga Volcano eruption (Supplementary Data 1), with some material likely also deposited beyond our survey area[21].

There was a widespread loss of material from Hunga Volcano in water depths shallower than 600 m; however, the seafloor changes in deeper water are more varied and relate to the pre-existing morphology. In water depths deeper than 600 m, which encompasses the flanks of Hunga Volcano and the area around the caldera rim, volume loss is confined to linear chutes that radiate from the caldera rim; all but one of which coincide with pre-existing channels (Fig. 2). Up to 70 m deepening occurred within the incised chutes, which are 5-10 km in length and 2 km in width. At the chute termini, the thickest accumulations of sediment are observed as lobate forms up to 22 m thick, with widespread blanketing of sediment across the pre-eruption seafloor. The most extensive erosion occurred within a new chute, which was not visible in pre-eruption data, that initiated at a topographic low at the NNW edge of the caldera between the two islands (Fig. 2a, Profile A). Most of the measured loss was within 6.5 km of the caldera rim, beyond this deposition dominated. There is no superficial morphology consistent with seafloor landslides (i.e., headscarps and/or scars) on the volcano flanks despite the scale of the eruption and associated seismicity, which makes this event distinct from partially-submerged eruptions that generated large slope collapses such as Anak Krakatau (Indonesia) in 2018[32].

### Volcaniclastic density currents as the primary agents of seafloor change on the volcano flanks
We relate these spatially variable changes in seafloor elevation on the volcano flanks to density currents laden with volcaniclastic material that radiated downslope from Hunga Volcano in multiple directions, exploiting and enhancing pre-existing topography (Figs. 2a, S6; Supplemental Movie 1). We suggest these flows initiated as collapsing eruption column material that plunged into the ocean and mixed with the ambient seawater, becoming turbulent underwater density flows (e.g., turbidity flows) that were steered by the complex topographic relief around the volcano. As we cannot fully discern the point at which flows transitioned from hot gas supported pyroclastic density currents to cooler turbidity currents, we refer to them as volcaniclastic density currents[21,33]. The volcaniclastic density currents most likely initiated from the partial collapse of the eruptive column of Hunga Volcano,

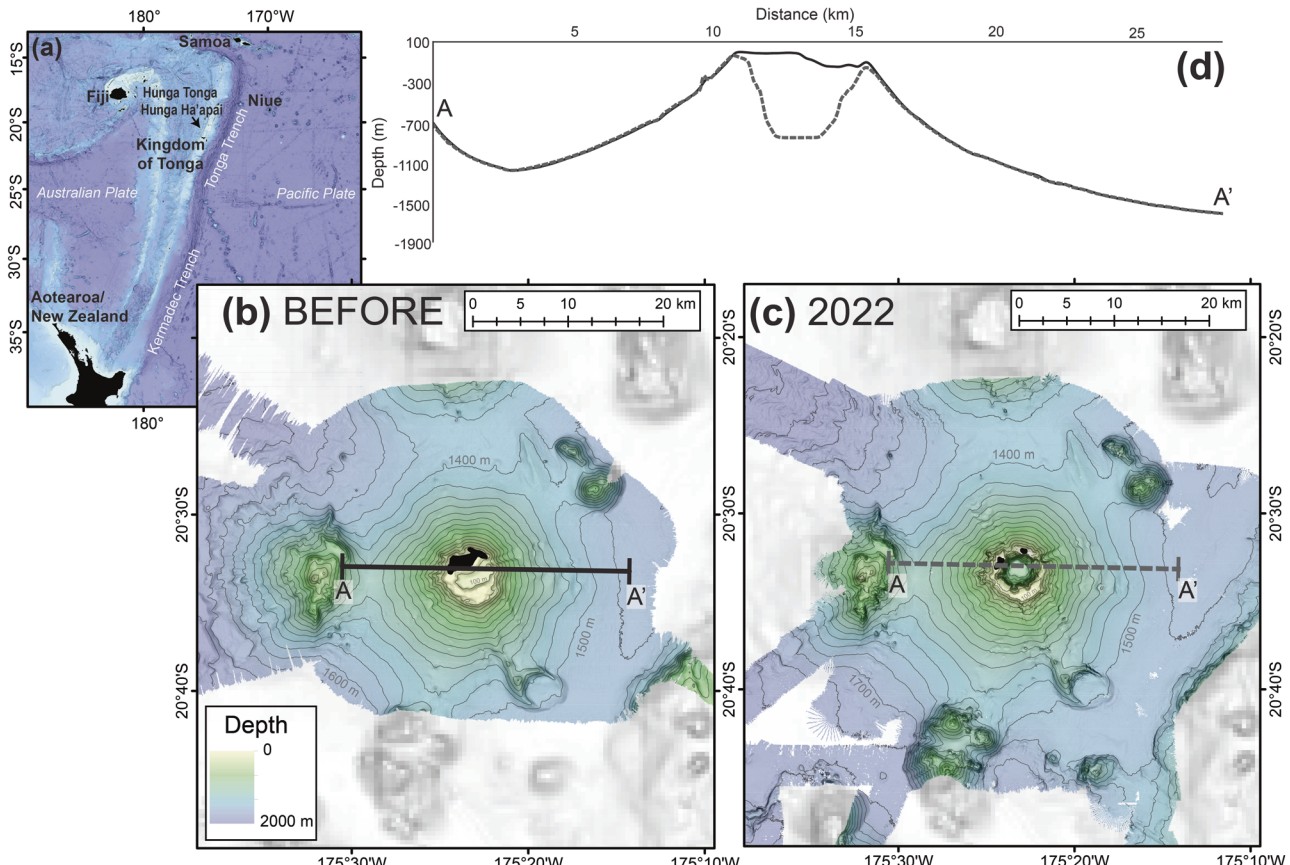

**Fig. 1 | Pre- and post-eruption bathymetry. a** Regional bathymetric map showing the location of the islands of Hunga Tonga and Hunga Ha'apai along the Tonga-Tofu-Kermadec Arc. Onshore regions are shown by black polygons.
**b**, **c** Bathymetric maps of Hunga Volcano acquired before (**b**) and after (**c**) the 2022 eruption. For details pertaining to bathymetry surveys refer to methods. **d** Solid line (A-A') and dashed line (A-A') on insets b & c indicate the location of solid and dash lined profiles shown in (**d**) that show the Hunga Volcano edifice remained intact after the eruption despite the ~800 m difference in the depth of the caldera and drastic changes in island topography (islands shown in black from[28]). The before eruption bathymetry was created from multibeam surveys in 2015[27], 2016[28] and 2017 satellite-derived bathymetry from Land Information New Zealand. This is overlain on GEBCO_2022 bathymetric grid (greyscale; https://doi.org/10.5285/e0f0bb80-ab44-2739-e053-6c86abc0289c).

with this hypothesis similarly supported by the timing and intensity of the tsunami waves and images of the base of the column[21,34]. However, primary magmatic explosivity or hydrovolcanic explosions could also have played a role. Sensitivity tests of our density current model (see Supplementary Material, Fig. S6), however, confirm that whatever initiation mechanism caused them, it most likely occurred within the caldera in order to cause the strong channelisation that was observed on the flanks of Hunga volcano and extends up to the caldera rim.

We observed crescentic scours (up to 30 m deep) and bedforms (up to 1200 m wavelength, 20 m wave height; Fig. 2), commonly seen in other deep-sea settings affected by powerful density flows[35], including density flows triggered by large magnitude earthquakes that severed seafloor cables and attained speeds of up to 20 m/s[36–38]. These observations are further corroborated by seafloor sediment coring and imagery, geochemical analyses and numerical modelling. Multicoring 80 km from the caldera sampled a volcaniclastic deposit with an abrupt basal contact above pre-eruption hemipelagic deposits, overlain by fine volcanic ash deposits tens of centimetres thick (Figs. 3, S1; Table S1). The entire deposit is coarser than the sedimentary layers below, and fines upwards – as expected from a particulate density current deposit. The density flow deposit dominantly comprises sand-sized mixed volcaniclastic material with up to 4 mm-diameter scoria and occasional preserved ripples, which are also observed in seafloor photos (Fig. S9). Similar density current deposits were sampled along a transect approaching the caldera (Fig. S2). Major elemental chemical compositions of volcaniclastic deposits were measured by X-Ray fluorescence spectrometry and overlap with ashfall collected from Tongatapu (main island of Kingdom of Tonga) post-eruption (measured by scanning electron microprobe analysis) as well as recent eruptions from Hunga volcano[39], and are distinct from other regional volcanoes (Supplemental Information; Figs. S3, S4; Supplemental Data 1). A lack of bioturbation within the volcaniclastic deposit, and no indication of oxidation, indicate that it is a relatively fresh deposit, as well as the absence of any hemipelagic layering on top of the volcaniclastic deposit. These combined observations support an interpretation of fast-moving and locally erosional flows that blanketed the flanks of the volcano and beyond, resulting in widespread deposition. Bathymetric difference mapping shows that the density currents were predominantly erosional on the upper slopes of the Hunga Volcano edifice, entraining an additional 3.5 km³ of material; more than the volume of the largest documented historical onshore landslide (Mount St Helens in 1980[40]).

Numerical modelling of the submerged density currents (see Methods section for further details) suggest that ~3–4 km³ of flow is required, originating from inside the caldera rim, to replicate the extent of the observed deposits, especially the overtopping of two submarine knolls to the south heading towards the international telecommunications cable that was damaged extensively after the eruption (Fig. 3). While our relatively simple model does not account for seafloor erosion, incorporation of additional material, nor multi-phase

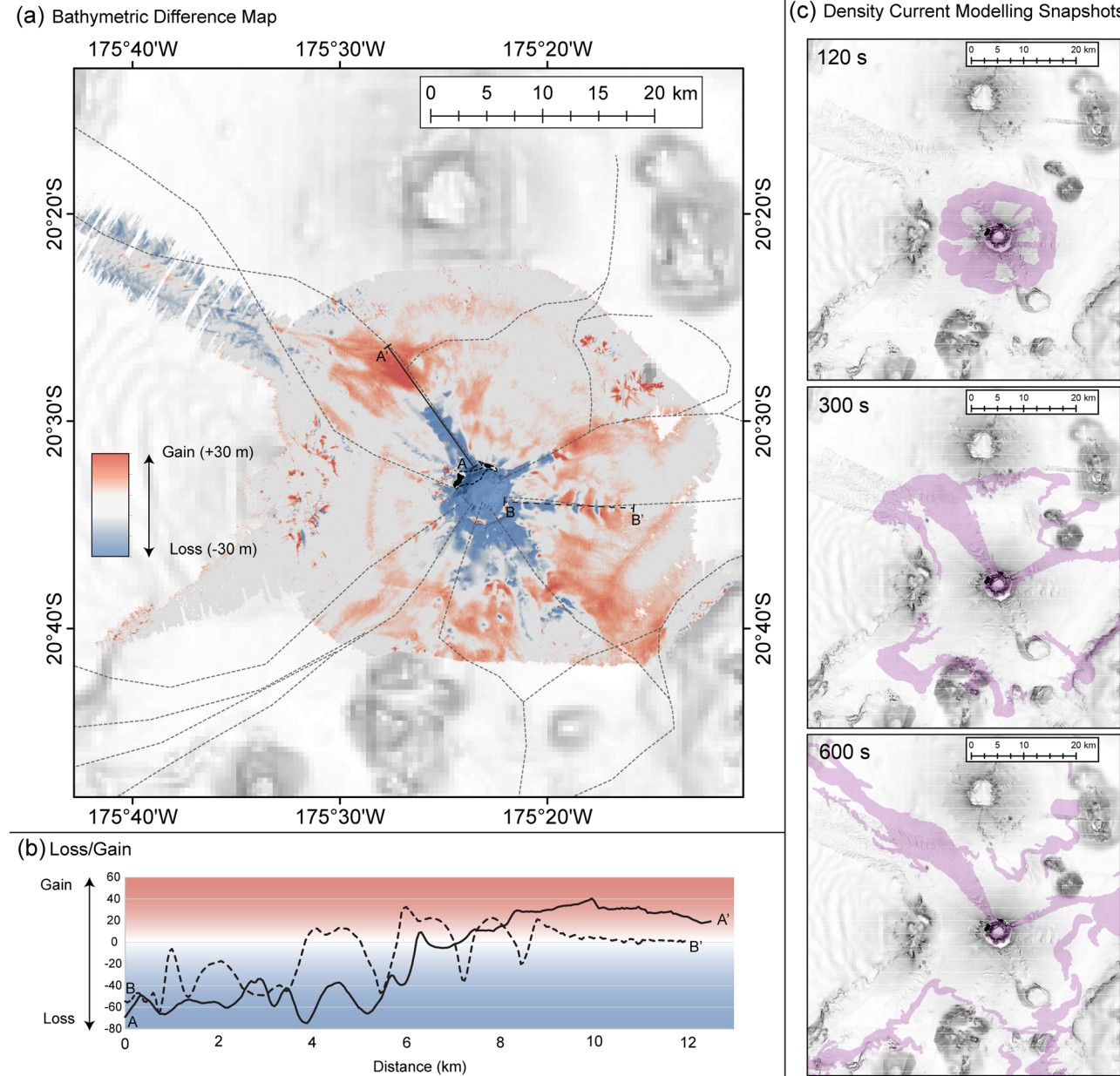

**Fig. 2 | Seafloor difference map and density current modelling snapshots.**
**a** Difference map showing the relative gain (red) and loss (blue) of seafloor when comparing pre-eruption bathymetry with post-eruption bathymetry. Coastlines pre (dashed) and post (solid) eruption are shown. Indicative density current flow paths (i.e., an accessible visualisation of the general flow pathways simplified from the model output animation provided (see Supplemental Movie 1 and Fig. S6)) are indicated with dashed grey lines plotted over the difference map to enable comparison of density current flow paths with the difference map results. **b** Solid (A-A') and dashed (B-B') black lines showing gain/loss along prominent volcaniclastic density current flow pathways, with the location of these profiles shown in (**a**) with corresponding solid and dashed lines. **c** For further context, select snapshots from the model animation (provided in Supplemental Movie 1) are provided at 120, 300, and 600 s into the modelling output. For these density current modelling snapshots, the map frame extent is the same as for the difference map, and the background imagery is 2022 multidirectional hill shade.

flow, the flow pathways that the simulated density currents follow align remarkably well with the areas of most pronounced seafloor change. The model (especially the density current flow paths) is consistent with the locations and extensive damage to the international and domestic telecommunication cables, damaged along 89 km and 105 km length, respectively[21] (Fig. 3). Furthermore, the modelled flow paths taken by the faster moving and thicker (in m) parts of the density current correlate well with the observed spatial trends in erosion and flow paths. The most pronounced erosion occurred immediately to the NNW of the caldera, where the modelling shows part of the density current being funnelled through an area of negative seafloor relief between the two islands and continuing downslope to the northwest. Post-eruption bathymetry suggests this part of the density current went well-beyond the northwest limit of our survey area. Overall, as the simulated density currents flow away from the caldera, the model illustrates how these seafloor flows can be steered by the local topography and how, in some locations, separate flows can overcome bathymetric barriers, converge back together and accelerate. These modelled results illustrating the paths of the simulated density currents corroborate the areas of impact observed in the seafloor surveys presented here (Fig. S19). Flow paths like this have previously been observed subaerially but only inferred in submarine settings[33].

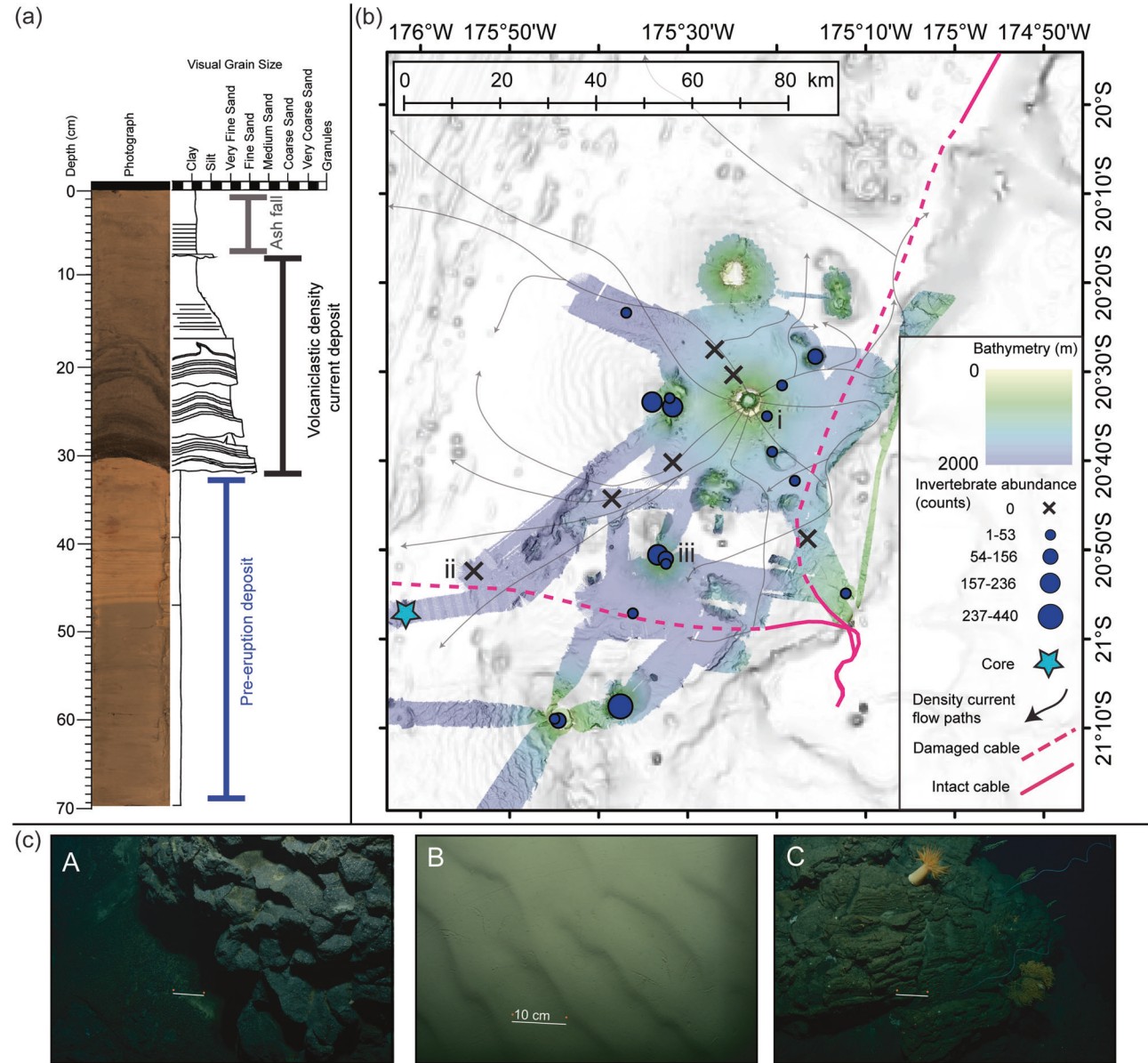

**Fig. 3 | Relation of density current flow paths to impact on the seafloor including the telecommunications cables, sediment system and invertebrate communities. a** Sediment core log with interpreted pre-eruption, volcaniclastic and ash deposits labelled, from the sediment core indicated with the light blue star in (**b**). **b** Compilation map showing seafloor multibeam survey and geologic and biological sampling after the 2022 Hunga Volcano eruption. Indicative density current flow paths taken from modelling results (as in Fig. 2, see Supplemental Movie 1 and Fig. S6) are shown in solid black lines with the arrows indicating directionality of the flow paths interpreted from the modelling simulation. The locations of damaged submarine cable are depicted with dashed pink lines. In order to reach the international cable, the density current must overtop two knolls directly E and ESE of location C, which partially constrains the minimum volume of material in the volcaniclastic density current. The extent of the post-eruption bathymetry survey is shown in colour and was collected by RV Tangaroa and USV Maxlimer. Dark blue graduated points and black crosses show locations where seafloor video footage documented invertebrate abundance (in count) during RV Tangaroa survey in April 2022. **c** Example images of the seabed across the study area are shown in insets A-C, and each image is located in (**b**). The white scale bar in each A-C inset is 10 cm, as indicated in the middle inset (**B**).

## Wider impacts of the eruption on ocean ecosystems and the role of topographic refugia

The density currents also exerted dramatic, and differential impacts on seafloor ecosystems over a large area. Towed camera footage of the seabed after the eruption revealed an absence of epifaunal biota on the flanks of Hunga Volcano, in the chutes and across most of the surrounding seafloor, indicating widespread loss of life following the eruption (Figs. 3, S9, 10, Supplemental Data 2). Close to the caldera, seafloor substrate was almost completely removed, indicating erosional regimes where animals would have been physically removed with the substrata (Fig. S10).

While the seafloor in most of the remaining survey area was smothered with volcanic deposits linked to volcaniclastic density currents and subsequent ashfall, areas of potential refugia from the impacts of density currents appear to have been created locally by the high and irregular seafloor relief provided by several seamounts (Figs. 3, S11, S19, Supplemental Data 2). We observed relatively undisturbed and diverse benthic communities on seamounts in the southern part of our survey area, at distances >50 km from the caldera. The density and diversity of these ecosystems aligns with our expectations for the warm, oxygenated and highly productive waters that exist adjacent to seamounts and volcanic islands of the Tonga-Tofua-

Kermedec Arc[41–43]. Closer to Hunga Volcano, abundant benthic invertebrate and fish communities were also observed on seamounts in the area, presumably protected from direct eruption impacts (Fig. 3; Figs. S11, S19). The locations of these refugia match closely the thickest (in m) flow paths of the density current modelling, which shows that deflection of flow pathways around irregular relief and seamounts left some areas relatively unaffected (Fig. 3, S6, S19; Supplemental Movie 1). Pre-eruption deep-sea benthic surveys for the Hunga Volcano edifice and adjacent seamounts were not available to compare with our new post-eruption data, except for discrete observation of mussel beds on a southern seamount within our study area[26]. The mussel beds visualised in 2007 and 2017 on the southern seamount were still present, and of similar density to prior observations[26], indicating that these communities were not affected by the eruption of Hunga Volcano (Fig. S11). Such refugia from pyroclastic density currents have been reported on land, where vegetation and people have been sheltered by local relief[44–46]; however, to our knowledge, this is the first time such a sheltering effect from volcaniclastic density current has been reported in a marine setting.

The removal and smothering of benthic communities and habitats have been documented following previous volcanic eruptions, albeit in different settings[47–52]. Ashfall from Mount Pinatubo reduced the density and diversity of foraminifera assemblages nearby in the South China Sea[53], while larger animal communities experienced mass mortalities following the underwater eruption of the caldera of Deception Island in 1967-1970 (South Shetland Islands, Antarctica)[54]. Ocean entering pyroclastic density currents from the 2008 eruption of Kasatochi Volcano (Aleutian Islands, USA) smothered shallow-water benthic communities[55]. In each of these cases, benthic recovery started within a few years and, in the case of the Deception Island eruption, shallow-water (<200 m) benthic communities were thought to have recovered 14 years after the event[56]. In a deeper setting (700–750 m) on Vailulu'u Seamount (east of American Samoa) there was higher abundance of hormathiid anemones, hydroids, a demosponge, brittle stars and octocorals on older basaltic seafloor following a series of eruptions[52], consistent with ecological succession of benthic communities, which occurred over a period of 12 years. A much slower and less complete recovery of macrofaunal communities was reported at vent-proximal sites on the East Pacific Rise at 2500 m[51]. We do not know the timescale over which the seafloor communities in the Hunga Volcano may recover, but we speculate that the process may be aided by re-colonisation from the refugia indicated above.

Persistent ocean impacts of Hunga Volcano were also observed within the water column. High turbidity, related to very fine volcanic ash, was recorded 3 months after the eruption at 200 m water depth up to 20 km to the NW of the caldera (Fig. S12). Future research will seek to determine if this water column turbidity is related to ongoing venting from plumes within the caldera. Near-seabed ash layers (e.g., nepheloid layers, a layer of water above the ocean floor with a significant amount of suspended sediment) were detected above the seafloor across many of our sampling sites, in some cases extending hundreds of metres up from the seafloor (Fig. S13). The turbidity anomalies (both mid-water and near-seabed layers) were sampled by Conductivity, Temperature, Depth (CTD) rosette and in all instances were dominated by ash particles, identified by scanning electron microscope imagery, with electron microprobe analysis revealing a similar composition to the volcaniclastic density current deposits and post-eruption ashfall samples from Tongatapu discussed above (Figs. S4, S14–17; Supplemental Data 1). Long-lived particulate suspensions of volcanic material have been reported for many other eruptions, with diverse ecological impacts[7]. Therefore, assessing the true impacts of a large eruption like the 15 January eruption of Hunga Volcano will require holistic and continued monitoring over several years to determine both short-term resilience, and longer-term recovery from the effects of the eruption.

Our data illustrate how an eruption of a single submerged volcano can have wide-reaching and diverse impacts. Volcaniclastic density currents resulting from the Hunga Volcano eruption were steered by antecedent morphology, further excavating pre-existing chutes and depositing the majority of the erupted volume on the seabed within 20 km of the Hunga caldera rim. Modelling and benthic observations indicate that the majority of the density currents originated within the caldera and flowed around bathymetric relief as they moved to deeper waters, with only local preservation of benthic communities on topographic refugia. Outside of these topographic refugia the seafloor ecosystems around Hunga Volcano were decimated by the eruption and related density currents.

While ocean impacts resulting from volcanic eruptions are typically hidden from view, we show they can have major consequences, including widespread loss of marine life and damage to critical seafloor telecommunication links, with knock-on socioeconomic impacts. The domestic cable that connects the Tongan island groups took 18 months to repair, leaving island communities without high-speed connections. The enduring effects of suspended volcanic material in the water column and the localised decimation of benthic communities may have as-yet unknown implications for ecosystem functioning, which may in turn have wider impacts on food security. Future monitoring (of both the volcanic edifice itself and the surrounding seafloor and habitats) is necessary to robustly determine the resilience and recovery of both human and natural systems to major submarine eruptions, and to more broadly assess the risks posed by the many similar submerged volcanoes that exist worldwide.

## Methods

### Field survey

Most of the post-eruption data included in this manuscript is from a voyage aboard the RV Tangaroa in April and May 2022. A portion of the bathymetric data is also from a survey by the USV Maxlimer in July-September 2022 as detailed below. These surveys together formed the NIWA-Nippon Foundation Tonga Eruption Seabed Mapping Project (TESMaP). The survey design aboard the RV Tangaroa was a combination of sampling at specific topographic and event locations (e.g., sites of breaks along the international and domestic communications cable and other volcanic seamounts and knolls in the immediate vicinity of Hunga Volcano), and sites along a distance (and impact) gradient on the flanks of Hunga Volcano. These latter sites comprised two transects aligned southwest and southeast from the Hunga Volcano caldera.

### Seafloor imagery collection and analysis

Seafloor imagery was collected with NIWA's Deep Towed Instrument System (DTIS), a battery-powered towed camera frame which records continuous high-definition digital video (1080p, @ 60 fps) and simultaneously takes high definition still images (24 megapixel) at 15 s intervals. The seabed position and depth of DTIS were tracked in real time using the KONGSBERG ultra-short baseline (USBL) transponder system. The seabed position of DTIS was plotted in real time using OFOP software (Ocean Floor Observation Protocol, www.ofop-by-sams.eu), with all navigation data, camera commands, and spatially-referenced observations of seabed type and the occurrence of visible mega-epifaunal invertebrates (animals living on or above the seafloor with sizes ≥2 cm) recorded to OFOP log files and captured by the ship's Data Acquisition System (DAS). For initial analyses presented here various levels of taxonomic identification were amalgamated into higher taxa: Hexactinellida (glass sponges), Demospongiae (sponges), Alcyonacea (various gorgonian octocorals, whip corals, stony corals (reef-forming), cup corals), Hydrozoa (stylasterid hydrocorals), Pennatulacea (seapens), Holothuroidea (sea cucumbers), Ophiuroidea (brittle stars), Asteroidea (seastars), Crinoidea (crinoids), Bivalvia (vent mussels). The abundance of invertebrate taxa were counted in real

time from the live video feed (totalling 31 h over a distance of 32 km) and hence the data included here should only be regarded as provisional estimates of abundance. The abundance data is available in Supplemental Data 2.

### Sediment core collection and analysis

Seafloor sediment samples were collected with the Ocean Instruments MC-800 multicorer system. The cores described here were then subsampled from the multicore tubes with push core tubes. These were 80 cm long polycarbonate core liners which were inserted down the core barrel, then sealed with absorbent foam, capped and taped at either end, and stored upright in a 4 °C refrigerator on board and onshore until analyses. Sediment core liners were then cut longitudinally, and the sediment parted using cheese wire to create a flat surface on the split core halves. Once split, cores were immediately photographed and subject to sedimentological core logging that comprised visual identification of sedimentary structures, grain size and texture. Grain size was visually characterised through use of a grain size comparator, which has been shown by prior studies to relate to the coarsest 5% of the grain size distribution[57]. Core sample observations were synthesised in a visual sedimentological log (Figs. 3, S1, S2). Deposits were visually differentiated on the basis of their composition, colour, sedimentary textures and grain size, which included the identification of two main facies that are linked to the Hunga volcano eruption. These are (i) Volcaniclastic density current deposits, which comprise dominantly sand to granule-sized volcanic material that generally fines upwards and a very sharp basal contact (where it was sampled); and (ii) A thin veneer of orange-brown oxidised ash-rich deposits that has no obvious internal structure and that was sampled at seafloor, consistently overlying the density current deposits. As continued suspension and settling of ash was observed by video and sampling of the water column 3 months after the eruption (Figs. S13–S15), we consider these surficial seafloor deposits to be most likely related to ash fall out; however, it is possible that this facies may relate to the fall out of ash-sized material that was suspended by the volcaniclastic density current as a dilute cloud and subsequently settled out after the deposition of the coarsest load of the current, or that this relates to settling from a surface plume. Regardless, this does not affect the key conclusions of this present study.

To establish if ash preserved in the marine cores collected during the TAN2206 voyage were sourced from Hunga Volcano, nine samples of ash from four cores were analysed for major elements by quantitative bench-top X-Ray Fluorescence (XRF) spectrometry and compared with Tongatapu ash deposits, samples from the water column (both described below) and literature data. Samples were selected from core tops, with an additional four samples picked from different phases of the eruption preserved in core 90, as well as a basal bioturbated layer representative of the seafloor prior to the 2022 eruption.

Samples were prepared by soaking in deionised water (18.2 MΩ) for 5 days, with periodic (1–2 h/day) agitating in a warm ultrasonic bath. As the sample material was very fine the ash was allowed to settle overnight, drained and fresh water added. Salinity was tested each day and after 5 days the water was found to have a conductivity consistent with deionised water. The samples were then drained of water and dried in an oven at 110 °C. For sample 90-1, aliquots of sample were taken after 1, 4 and 5 days washing to test the effect of washing on ash composition.

The dried samples were crushed in a ceramic mortar and pestle, which was then cleaned with detergent and dilute citric acid added to the bowl overnight to avoid contamination. Fused glass disks of the samples were prepared by mixing 1 g of sample, 10 g of flux (35% lithium tetraborate, 65% lithium metaborate) and an NH$_4$I tablet in a platinum crucible. The mixture was then heated in a Claisse LeNeo Fluxer at 1075 °C for 25 min to produce a fused glass disk suitable for analysis by XRF spectrometry.

Analyses were conducted on a Panalytical Minipal-4 energy-dispersive XRF spectrometer at NIWA, Wellington. Each analytical run consisted of three analyses at different keV energy to optimise analytical conditions for each element. Two international standards (USGS basaltic standards BHVO-2 and BCR) and an internal standard (TP-1) produced at Victoria University of Wellington, New Zealand[58] were analysed together with Hunga Volcano samples. The resultant data was processed using a calibration curve derived from nine international standards. Results for samples and standards corrected for loss on ignition (LOI) by summing the elements to 100% are given in Supplementary Data 1. LOI was determined as the difference between the original total and 100 % and is also given in the table. Average values for the standards are in close agreement with certified values (generally within 1-3% relative, although P$_2$O$_5$ is up to 9% due to its low abundance), and standard deviation analytical precisions for each element (other than Na$_2$O and P$_2$O$_5$) and are generally <2% (2 SD) relative.

### Water Column Analysis

Water column data (particle density) was collected with sensors on a conductivity, temperature, depth (CTD) rosette equipped with turbidity optical sensors, offering an insight into the particle makeup of the water. Niskin water sampling bottles attached to the CTD rosette were fired at relevant depths throughout the water column for discrete water samples. This water was then filtered onto 25 mm polycarbonate filters and dried before analysis with the Scanning Electron Microscope.

Shards of material filtered from CTD samples were identified as volcanic glass based on their optical properties. To constrain their origin the samples were analysed for major elements together with a sample of ash collected from Tongatapu ('Ash' samples) a few days after the eruption. Analyses were performed at the University of Otago using energy-dispersive spectroscopy (EDS) on a Zeiss Sigma VP FEG SEM, JEOL FE-SEM6700 scanning electron microscope. The analyses were semi-quantitative with the aim of analysing as many shards as possible to investigate their origin.

### Post-eruption multibeam surveys

Complete seabed mapping of the post-eruption seabed around Hunga Volcano was one of the primary goals of TESMaP. The seabed mapping was undertaken in two parts. The first was by the RV Tangaroa (TAN2206) in April-May 2022 using a Kongsberg EM302 multibeam echosounder system which mapped the flanks on Hunga Volcano and the surrounds. The second part of the TESMaP seabed survey was undertaken by an uncrewed surface vessel, the USV Maxlimer (MAX2201), in July-September 2022 utilising a Kongsberg EM710 multibeam echosounder system. USV Maxlimer was operated from a Remote Operating Center in Essex, UK while a team of surveyors from around the world (Ireland, Poland, Egypt, Australia, and New Zealand) remotely operated the EM710 multibeam.

The Kongsberg EM302 multibeam echosounder system operates at a frequency of 30 kHz and comprises 288 beams/432 soundings per swath, with real-time beam steering compensating for ship motion. The beam width is 1° along-track and 2° across-track, producing small acoustic footprints.

Positioning was provided by GPS, differentially corrected by the Fugro SeaStar XP Wide Area Differential GPS (WADGPS). Heave and attitude data were provided by an Applanix POS/MV 320 motion sensor. Measurements of roll, pitch, and heading are accurate to 0.02° or better. Heave measurements supplied by the POS/MV maintain an accuracy of ±5% of the measured vertical displacement or ±5 cm (whichever is larger) for movements that have a measured period up to 20 s. No significant heave artefacts were observed in the processed bathymetric data.

The Kongsberg EM710 multibeam echosounder system operates at a frequency range of 70 to 100 kHz and comprises 200 beams/

**Table 1 | Sources of bathymetry used to generate the pre-eruption DTM**

| Dates | Instrument | Vessel | Voyage Id | PI |
|---|---|---|---|---|
| Sep 2017 | SAR and satellite imagery | | DEM_20170919_5m.tif | Garvin |
| May 2016 | EM302 and EM710 MBES | *RV* Falkor | FK160407 | Ferrini |
| Nov 2015 | WASSP MBES | small vessel | | Cronin |
| 2017 | Satellite-derived bathymetry | | AY14_TongaAOI15_2m_82-1048_m | LINZ/PRNI |

400 soundings per swath, with real-time beam steering compensating for ship motion. The beam width is 1° along-track and 1° across-track, producing small acoustic footprints. Heave and attitude data were provided by a Seapath 130 with MRU-5 + MK-11 motion sensor.

These multibeam bathymetry data were edited and processed during the voyage using QPS Qimera v2.3.1 software, then gridded to a 50 m cell-size.

### Pre-eruption Multibeam surveys

To generate the pre-eruption Digital Elevation Model (herein referred to as bathymetry) we searched all available bathymetry repositories for survey data conducted around Hunga Volcano prior to 2022. These data repositories include the IHO DCDB, the Seabed 2030 Pacific Data Center hosted at NIWA, AusSeabed, EMODnet, and JAMSTEC. For this search we identified three surveys that were suitable to generate this pre-eruption surface (Table 1). These surveys were by the RV Falkor (FK160407) in May 2016 using a Kongsberg EM302, a small boat survey in November 2015 using a WASSP multibeam sounder, and 2017 satellite-derived bathymetry from Land Information New Zealand as part of the Pacific Regional Navigation Initiative.

These multibeam bathymetry data were edited and processed using QPS Qimera v2.3.1 software, then gridded to a 50 m cell-size. Corrections for sound velocity were provided either by the direct SV measurement by the deployment of sound velocity probes or by water sound speed calculated from conductivity, temperature and pressure-depth (CTD) values. Refraction residuals for these surveys were further reduced using the TU Delft Speed Sound Inversion tool within Qimera[59].

To determine the spatial extent and vertical change in the seafloor from the 15 January eruption we used the TESMaP and the pre-eruption bathymetry surfaces to perform a change detection analysis. The pre- and post-eruption DTM surfaces are co-registered to further reduce vertical and horizontal uncertainties on any analysis. Analysis of the DTMs was undertaken using ArcPro GIS software where many bathymetric derivative surfaces (for example slope, aspect, and curvature) were generated to understand the geomorphology. To determine the spatial extent and vertical change in the seafloor from the 15 January eruption, the post-eruption bathymetry was subtracted from the pre-eruption bathymetry. Volumetric calculations of differences between the pre- and post-eruptions DTMs were undertaken using the Cut Fill tool, on the summit and flanks of the Hunga Volcano (Fig. S18).

Vertical uncertainties for multibeam acquired bathymetric data are depth dependent and are usually reported as being 1% of depth (ranging from 20 cm to 20 m). The variable uncertainty makes any quantitative volume estimation difficult. However, morphology as measured by multibeam echo sounders is typically not affected by vertical uncertainties, and if so we would expect artifacts in the difference maps which we do not see. While quantitively estimates of gain/loss of seafloor will be impacted by this uncertainty range, morphological changes are not. Pre- and post-eruption digital elevation models can be accessed at 10.5281/zenodo.7456324. To determine the percentage of the volcano area that is subaerially exposed, the area of the submerged edifice was defined by a change in slope of 3° from the base of the edifice. This result was compared to the area of the pre-eruption island area to demonstrate that <1% of the volcano is subaerially exposed.

### Volcaniclastic density current modelling

The turbulent gravity-driven current generated when the volcaniclastic density current mixed with seawater and the ocean above was modelled with the Basilisk modelling software[60] using a two-layer model with the lower layer representing turbulent density current and the upper layer representing the overlying sea water[61]. This approach allows us to model the submarine density current flowing over the complex bathymetry at the same time as we model its effects on the seawater above it. The vertically-averaged density current was modelled as a dense incompressible Newtonian fluid using the Saint Venant (Shallow Water) Equations affected by the lighter fluid (the sea water) overtop. The vertically-integrated equations for the density current are:

$$\frac{\partial h_s}{\partial t} + \frac{\partial}{\partial x}(h_s u_s) + \frac{\partial}{\partial y}(h_s v_s) = 0 \tag{1}$$

$$\frac{\partial h_s u_s}{\partial t} + \frac{\partial}{\partial x}\left(h_s u_s^2 + \frac{1}{2}g h_s^2\right) + \frac{\partial}{\partial y}(h_s u_s v_s)$$
$$+ g h_s \frac{\partial}{\partial x}\left(\frac{\rho}{\rho_s}h + z_b\right) = \mathscr{F}_{sx} + \frac{\tau_{sx}}{\rho_s} \tag{2}$$

$$\frac{\partial h_s v_s}{\partial t} + \frac{\partial}{\partial x}(h_s u_s v_s) + \frac{\partial}{\partial y}\left(h_s v_s^2 + \frac{1}{2}g h_s^2\right)$$
$$+ g h_s \frac{\partial}{\partial y}\left(\frac{\rho}{\rho_s}h + z_b\right) = \mathscr{F}_{sy} + \frac{\tau_{sy}}{\rho_s} \tag{3}$$

Where $h_s$ is the thickness of the volcaniclastic density current and $h$ is the depth of the sea water above it. The top equation is derived by vertically integrating the mass conservation equation and the bottom two equations the momentum. The vertically-averaged velocity of the volcaniclastic density current in the $x$ (east) and $y$ (north) directions are $u_s$ and $v_s$. $\rho_s$ and $\rho$ are the density of the volcaniclastic density current and sea water respectively, $g$ is the acceleration due to gravity. $\mathscr{F}_s$ is the contribution of friction and $\tau_s$ shear of the turbidity current with $x$ and $y$ denoting the direction. In these equations the flow rheology is prescribed by the friction, in this model a simple quadratic friction is used and the coefficient is set to $10^{-4}$ for the density current, similar to the water as physical modelling results suggest these volcaniclastic density currents are highly mobile and act very similarly to dense fluids[62]. We assume a free slip condition in the bottom boundary when deriving these equations, but other boundary conditions can be accommodated by $\tau_s$.

The seawater was modelled using the Saint Venant Equations influenced by the denser submarine density current layer underneath:

$$\frac{\partial h}{\partial t} + \frac{\partial}{\partial x}(hu) + \frac{\partial}{\partial y}(hv) = 0 \tag{4}$$

$$\frac{\partial hu}{\partial t} + \frac{\partial}{\partial x}\left(hu^2 + \frac{1}{2}gh^2\right) + \frac{\partial}{\partial y}(huv)$$
$$+ gh\frac{\partial}{\partial x}(h_s + z_b) = \mathscr{F}_x + \frac{\tau_x}{\rho} + h\left(\frac{g}{\alpha}\frac{\partial \eta}{\partial x}\right) \tag{5}$$

$$\frac{\partial hv}{\partial t} + \frac{\partial}{\partial x}(huv) + \frac{\partial}{\partial y}\left(hv^2 + \frac{1}{2}gh^2\right)$$
$$+ gh\frac{\partial}{\partial y}(h_s + z_b) = \mathscr{F}_y + \frac{\tau_y}{\rho} + h\left(\frac{g}{\alpha}\frac{\partial \eta}{\partial y}\right) \quad\quad (6)$$

Where $u$ and $v$ are the vertically averaged velocity in the $x$ and $y$ directions, $\mathscr{F}$ is the contribution of friction and $\tau$ shear in the $x$ and $y$ directions. $\eta$ is the water surface. These equations are very similar to the two-layer equations solved by Giachetti et al.[63] within the VolcFlow software, however, with a more mobile dense flow following results from physical modelling of aerated granular flows entering water[62]. The lateral boundary condition for the volcaniclastic density current (the lower layer) is a solid boundary (no flux over the boundary) but the boundary of the model is sufficiently far from the volcano that the density currents that are not stopped by the bathymetry do not reach the boundary within the simulation time (the boundary is beyond the geographic limit of all figures and outputs here, and thus is not shown in any density current model outputs included). The sea water (upper layer) is solved with a radiating boundary condition at the edge of the domain, which allows for waves to propagate out of the domain and not reflect back in.

The density current was initialised as an 80 m thick, 4 km radius column of dense fluid[62] (density 1600 kg/m$^3$ and total volume of 4 km$^3$ which for instance could represent a density current made up of 1.6 km$^3$ of volcaniclastic material of density 2500 kg/m$^3$ mixed with seawater, at a volumetric ratio of 2:3) on top of the Hunga Volcano edifice (Table S2 model simulation 1–4 km$^3$). This could represent either the point where the eruptive column has collapsed and mixed with the sea water or where an effusive density current has mixed with the seawater. The upper sea water layer is initialised as a flat surface ($\eta = 0$) which then evolves according to the movement of the density current beneath it.

The TESMaP data was used for the bathymetry data around the edges of the edifice but the centre of the caldera was filled into a maximum depth of 200 m to represent the bathymetry before the caldera collapse. Topography data for the islands of Hunga Tonga and Hunga Ha'apai was taken from the 2017 bathymetry and used to fill in where these islands remain above sea level after the 15 January 2022 eruption. This near-field bathymetry was blended into a coarser bathymetry grid made up of GEBCO 2021, Seabed 2030 soundings, and LiDAR for Tongatapu.

This modelling is a very simple representation of a very complicated process. While it captures some aspects well it is not able to resolve other aspects. The density current is represented as a dense Newtonian fluid flowing over an unerodable bed. It does not erode sediment as it flows down the flanks of the edifice and neither does sediment deposit when the speed decreases, it just flows to the lowest point in the bathymetry. Thus, this model cannot predict where the density current will finally stop (i.e., runout distances), except in depressions or hollows, but it can indicate where the density current may have enough momentum to overcome bathymetric barriers (such as the saddles to the south of Hunga Volcano). If the volcaniclastic density current is created by a column collapsing back into the ocean and mixing with the water, then this model will also underestimate the surface waves generated as it does not consider the effect of the volcanic material falling back into the water. It can, however, capture the motion of the density current as it cascades down the sides of the edifice and the wave generation process in the upper layer that occurs due to this (see Supplemental Movie 2).

We have provided a description in the Supplemental information of the differences and outcomes of the different model runs undertaken to summarize impacts of varied bathymetry, fountaining within the crater or outside, varied volumes, changed densities of the volcaniclastic density current, and changes to friction of volcaniclastic density

current (Table S2). To investigate the sensitivity of the density current flow paths to the location of the originating volcaniclastic density current, an alternate initialisation was modelled starting with a annulus of material between five and seven km from the centre of the caldera (thus originating outside the main caldera) and 53 m thick (so also ~4 km$^3$ volume of material) but with a radial velocity of 10 m/s to ensure that it flowed outwards rather than falling back inwards to the caldera. This initialisation shows that the strong channelisation of the flow on the flanks of the Hunga edifice that match with the chutes, erosional scars, and channelisation observed in the bathymetric data from the voyage does not occur when the volcaniclastic density current originates outside the caldera (compare Fig. S7a which is initialised outside the caldera with Fig. S6a which is initialised within). Because this channelisation was clearly observed, that is evidence that the volcaniclastic density current must have originated within the caldera.

The total volume of the volcaniclastic density current specified in our final model was chosen based on the conditions that resulted in a density current that was able to reach the international cable, which is known to have been extensively damaged following the Hunga eruption (at least 89 km was buried or damaged by a volcaniclastic density current[21]). Small volume density currents that were modelled with an initial volume of 2 km$^3$ or less had insufficient inertia to overcome the topographic relief (specifically the saddles in the ridges to the south of Hunga edifice; see Fig. S7b) and were thus incapable of reaching the international cable. The model runs presented in the manuscript are based on an initial density current volume of 4 km$^3$ that resulted in flows that reached, and ran out beyond, the international cable. Sensitivity tests on the density of the density current show that the distance the flow reaches is relatively insensitive to the density of the density current (i.e., the ratio of volcanic material to seawater) with only minor differences being observed for densities of 1400 kg/m$^3$ and 1800 kg/m$^3$ (see Fig. S7, c and d).

## Data availability
Geochemical data, core logs, photographs, relevant coordinates, and raw faunal counts are provided within the manuscript or in the supplementary information or supplementary datasets. The pre- and post-eruption bathymetric data can be accessed via Zenodo (https://doi.org/10.5281/zenodo.7456324)[64].

## Code availability
The code for density current modelling was developed using the Basilisk software available at (http://basilisk.fr/). The initialisation code for the model used within this manuscript is available from the corresponding author upon request.

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

## Acknowledgements

We thank the Kingdom of Tonga for allowing us to undertake this research, and in particular acknowledge the support of The Nippon Foundation through the NIWA-Nippon Foundation Tonga Eruption Seabed Mapping Project for providing financial support; and The Nippon Foundation-GEBCO Seabed 2030 project and their alumni for their support. We thank Semisi Panuve from Tonga Cable Limited for his assistance in understanding impacts to seafloor cables. We extend our appreciation to Prof Shane Cronin and Dr Marta Ribo for data-sharing and continued collaboration with this work, as well as for providing airborne ash samples, and to Dr Suzanne Bull, Dr Jess Hillman, and Dr Malcolm Arnot for assisting with core CT splitting, scanning and ITRAX analysis. This work would not have been possible without the captain, crew, and scientists aboard RV Tangaroa (Voyage TAN2206) and the use of SEA-KIT International's Uncrewed Surface Vessel Maxlimer for mapping the caldera. We extend our gratitude to everyone involved in these voyages and for assistance in processing the results. SS acknowledges co-funding and support from NIWA SSIF Oceans 'Structure and function of marine ecosystems' and 'Sedimentary systems'. MAC, IAY, and JEH acknowledge funding from the Natural Environment Research Council (NE/X00239X/1 and NE/X003272/1) and support from the International Cable Protection Committee.

## Author contributions

Conceptualization of the work: S.S., S.J.W., M.A.C., I.A.Y., T.K., R.W., and K.M. Voyage and sample collection: S.S., K.M., L.J.H., E.A., and M.R.C. Sample analysis, data analysis, illustrative outputs: S.S., K.M., S.J.W., M.A.C., J.E.H., I.A.Y., E.M.L., M.R.C., A.A., R.W., L.J.H., and E.A. Funding acquisition: M.J.M.W., K.M., M.A.C., I.A.Y., and J.E.H. All authors contributed to the writing of the manuscript.

## Competing interests

The authors declare no competing interests.
