## [Peer Review File · Nature Communications]

Volcaniclastic density currents explain widespread and diverse seafloor impacts of the 2022 Hunga Volcano eruptionREVIEWER COMMENTS

Reviewer #1 (Remarks to the Author):

This article describes timely observations made of the 2022 eruption site and accompanying analyses. Fortuitously, multibeam sonar data of good quality were available from before the eruption, collected sufficiently recently that the depth changes are almost certainly due to the eruption. There has been a lot of interest in using depth changes around volcanic areas at smaller scale to study effects of volcanism (e.g., Casalbore et al.; Watts), which this study adds to. Seabed photographs from prior expeditions were available also for comparison with new bed photos. The authors document sediment core and bottom photographic evidence that indicate the pathways travelled by the submarine pyroclastic flows. The area will further become a benchmark for studying how seabed fauna recover from such events.

Given the widespread media coverage of this event, I would not be at all surprised if this article will become highly viewed. I hope I don't appear to be diminishing the role of the Tonga eruption and the damage it caused, though the importance to hazards research probably lies with it helping to understand the risks posed by similar magma bodies nearer to major population centres, such as the Phlegraean Fields near Naples. To the north of San Miguel Island of the Azores, another field of giant sediment waves can be found (Weiss et al. (2016); Chang et al. (2022)). They lie north of the Furnas suspected magma body of similar concern. The article mentions the scale of the eruption, though it might have been useful to read a magnitude comparison with other known eruptions, rather than leaving that to other publications.

Despite my enthusiasm, I found reading the article somewhat frustrating, as it dips a toe into many topics without exploring them in great detail. At the end of reading it, I would certainly feel happy citing this article for the effect on the seabed around the vent and as another indicator of the scale of the eruption, though I'm not sure it will get cited so much for specific insights. This is illustrated by the abstract, which emphasizes both the geological event and biological impact. As a geoscientist, selfishly I might have preferred more focus on the geological event. For example, only the optimum results of modelling of pyroclastic density currents are shown, but this seems a rare example where we can be sure the deposits can be associated with an event (almost unequivocally). Sensitivity tests (varying volume and density of the originating flow) might have been explored and inferences drawn from comparison of those results with the field data. Although the modelling was very simple as the authors admit, I wondered if the spatial pattern of differences may allow at least a discussion of possible range of scale of the submarine PDCs. The flows evidently lacked sufficient kinetic energy to over-top the surrounding seamounts and friction (bed shear and momentum transfer from water entrainment) - this seems a basic constraint on them. Do the multibeam data suggest the elevation that the PDCs reached? The areas of bed loss in Figure 2 around the seamounts suggests that they may have. Equating potential energy gain for kinetic energy implies the velocity would be at least $\sqrt{2gh}$ ['at least' because this neglects friction]. Given that velocity (from which bed shear stress can be estimated), is it surprising or unsurprising that widespread erosion occurred over the edifice and along the NW channel? This is just an illustration - it seems to me a punchier outcome could be developed.

Detailed suggestions:

Line #

31 Is it possible to be more specific concerning biological communities? Fauna? To be pedantic, if sampling for bacteria, there may have been some carried in by the ocean. On the other hand, it might not be surprising not to see whales as they are not so abundant.

40 The ash and chemicals from large volcanic...

48 height

60 data from surveys before and after eruptions that can be used to quantify ..

77 The text could be made more succinct in places. Here for example:

began erupting in late ...

Put a full stop after 2021.

83 I would not put quantities like 99% unless they have actually been measured. Instead, use "majority"

85 A map of bathymetry change derived by subtracting ... from ...

88 Integrating depth changes over selected areas of the map, this change ...

Shouldn't these polygons be shown in the electronic supplement? How could these results be reproduced?

90 small features

97 I would think the volume ejected in the eruption column would also be quite uncertain. How does that change this assessment?

99 varied

100-101 I found it hard working out the details from the maps shown (I can see the areas of sediment waves in Figure 1 roughly correspond with the major depth changes, but not much more than that). It might be useful to plot some depth contours from the post-eruption survey over the depth changes in Figure 2.

101 Up to 70 m of entrenchment was ...

106 Could the NNW entrenched channel have initiated as a landslide? Might landslides have occurred in the area marked as "diffusive flow" in Figure 2? They look difficult to rule out. Maybe instead say that there is no superficial morphology typical of landslides remaining, e.g., crown/headwall.

122 I was left not knowing really what was meant by "powerful density flows" in terms of magnitude. Can the scale (wave spacing and height) be compared with features created by known (measured) currents or even inferred from cable breaks? The gravel waves produced by the Grand Banks failure have been well studied for example.

125 Without dates, strictly speaking we don't know if the deposit was from the eruption (though likely) or an earlier eruption. Is there anything further about the deposit in the core that indicates it is young? Perhaps lack of any iron-manganese coating of grains near the surface, lack of bioturbation, glass shards are unaltered, etc?

131 Bathymetry difference mapping ... were predominantly erosional on the

135 ... of sediment laden flow ...

? or particle volume (DRE)?

139 Do we know the timing of the cable breaks (presumably yes)? If so, what was the delay between appearance of the atmospheric eruption column and these breaks? Does that tell us anything about how the base of the eruption column collapsed?

195 In my opinion, the conclusions ought to focus more on the results of the marine work and modelling, not broader social issues that have not really been developed in the article.

Figure 1 - quite hard seeing much detail in these maps. I can see the sediment waves. Can the maps be shown enlarged?

I also wondered if a traditional figure with overlain depth contours might also be useful, as the contours can help accentuate the channels and minor gullies.

366 insert (grey areas) before "2017".

Figure 2 is annotated with the author's interpretations of processes, rather than features in the map. Traditionally, readers like to see the evidence presented in a non-genetic way so that they can assess it alongside basic observations, before arriving at the interpretation. I would suggest replacing these terms with observations.

The scale bar on the map needs values even if the same as in the cross-sections graph below (or otherwise say so in the figure caption).

372 Please be more specific about what is meant by flow trajectories. E.g., do these represent motion of a virtual particle in the flow or flow front?

Figure 3

377 modelling results

381 and each image is located in...

Sediment core is located by ...

383 with interpreted pre-eruption ...

405 I don't know what "abundance of biological assemblages" means exactly - different species, individuals?

416 please write in the past tense, as it is unclear as written if the CAT scans were useful or if useful in general (maybe not there).

425 In our study of cores from the Azores, volcanoclastic turbidites were found to have no organic carbon (well below threshold) - Chang et al. (2021). It seems obvious that there wouldn't be, though was organic carbon detected? This is a minor further check.

479 It might be useful for readers to know what this means in terms of location precision and accuracy.

486 Were water-column data recorded with the EM710? This might be useful information for later studying impact of turbidity on pelagic fauna.

521 It also varies with beam direction and with noise (Schmitt et al.). The final difference grid (Figure 2) does not contain any obvious known artifacts (e.g., Hughes Clarke et al.), which is good.

523 This is not quite true, some can look like geology (de Moustier & Kleinrock) but the lack of them in the change map is reassuring.

535 This is a high density, it seems to me, if the deposits are nearer 2500 kg/m³. I thought it a shame the volume and density were not explored further with sensitivity tests (ie., what range of these values leads to a breakdown in match to the observations and what can we learn about the flow (or limits of the modelling) from that comparison?).

The areas of compliance and non-compliance of the model results have not really been explored.

536 no need for "back"

Electronic supplement:

Figure S10. Please mention the sonar and frequency used to produce the water column backscatter image lower-right.

I very much enjoyed reading the results of this study and look forward to seeing a revised version in print.

Neil C Mitchell, University of Manchester, January 2023.

Casalbore D, Di Traglia F, Bosman A, Romagnoli C, Casagli N, Chiocci FL (2021) Submarine and subaerial morphological changes associated with the 2014 eruption at Stromboli Island. *Remote Sensing* 13:article 2043

Chang Y-C, Mitchell NC, Hansteen TH, Schindlbeck-Belo JC, Freundt A (2021) Volcaniclastic deposits and sedimentation processes around volcanic ocean islands: the central Azores. In: Chang Y-C, Mitchell NC, Quartau R, Hübscher C, Rusu L, Tempera F (2022) Why are submarine sediment waves more common on the north sides of the Azores volcanic islands? *Marine Geology* 449:art. 106837

de Moustier C, Kleinrock MC (1986) Bathymetric artifacts in Sea Beam data: How to recognize them and what causes them. *J. Geophys. Res.* 91:3407-3424

Di Capua A, De Rosa R, Kereszturi G, Le Pera E, Rosi M, Watt SFL (eds) *Volcanic Processes in the Sedimentary Record: When Volcanoes Meet the Environment*. Geol. Soc. Lond., London

Hughes Clarke JE, Mayer LA, Mitchell NC, Godin A, Costello G (1993) Processing and interpretation of 95 kHz backscatter data from shallow-water multibeam sonars. In: *OCEANS'93 Proceedings*. IEEE, Victoria, B.C., pp 437-442

Schmitt T, Mitchell NC, Ramsay ATS (2008) Characterizing uncertainties for quantifying bathymetry change between time-separated multibeam echo-sounder surveys. *Cont. Shelf Res.* 28:1166-1176

Watts AB, Nomikou P, Moore JDP, Parks MM, Alexandri M (2015) Historical bathymetric charts and the evolution of Santorini submarine volcano, Greece. *Geochem. Geophys. Geosys.* 16:847-869

Weiss BJ, Hübscher C, Lüdmann T, Serra N (2016) Submarine sedimentation processes in the southeastern Terceira Rift/São Miguel region (Azores). *Marine Geology* 374:42-58

Reviewer #2 (Remarks to the Author):

This manuscript represents an important contribution, marking the first documentation of the impacts to the benthos of the Jan 2022 Hunga Tonga eruption and providing some key observations on the physical processes of subsea dispersal of volcanic material. I think the paper could be improved by refining the message about the hazards posed by this eruption relative to the potential hazards of all submarine volcanoes. In addition, the authors should consider more description of the data and models and de-emphasizing some of the interpretation - given that there may not yet be sufficient information. Below I describe a number of suggestions that I think would significantly improve the paper. If addressed, I highly recommend its publication. The eruption itself and its clear impacts on local populations were highly publicized and of broad interest to the scientific community and the public. In addition, there were well-documented global impacts that captured the interest of the world (e.g., pressure waves). These results will likewise be of high interest because they reveal the subsea impacts in some detail for the first time. The paper is well-constructed and reports high-quality data that advances our understanding of this important volcanic event and its impacts. Below I describe some of the potential improvements to the manuscript.

1. There are many superlative claims about the eruption and data collection that I view as distracting at best and disingenuous at worst. The high interest of the topic speaks for itself does not require the authors to go out on such limbs. Nature should accept some responsibility for this as, intentionally or not, it seems to encourage such language in its publications. When it is accurate, I'm all for it, but in this case I don't think there is a strong enough case to merit it. Some specific examples...

line 28 "largest seaborne volcanic eruption in nearly 150 years" is based on a citation of a letter that does not provide any documentation of eruption volume, but instead cites a YouTube presentation. The authors note that a significant portion of the volume erupted is removed from the existing edifice. Perhaps it is the largest submarine eruption of the last 150 years, but I don't feel like that it is currently well-supported and it would be simple enough to say "one of the largest documented seaborne volcanic eruptions in perhaps a century or more".

line 86 "reveal the seafloor impacts of a major submarine eruption for the first time" seems to ignore several decades of work documenting submarine eruptions. Most noticeably there is no reference to Carey et al., 2014; 2018 that documented seabed impacts on Havre volcano that erupted in 2012.

2. The authors have made special note of the hazards from this eruption, with good reason. The impact to the local island communities through volcanogenic tsunami and failure of the subsea telecom cables are very important. However, I thought that the authors implied that these risks are common to all submarine volcanoes and that by juxtaposing those statements next to the fact that there are so many submarine volcanic features, that they implied that the risks are exponentially greater than reality. The hazards from this eruption are very real, but are predicated on it being a very large, shallow-water eruption in a populated area. Eruptions of this nature can and will happen again and being prepared for that is important. However, most submarine eruptions are too deep and too remote to pose similar hazards. The authors cited the documentation of 100,000 of seamounts as increasing this risk, but the

vast majority of those are extinct volcanos more than 100 Ma. I would suggest caveating the discussion of submarine volcanic hazards by pointing out what, when, and where those hazards are most prominent (e.g., Lines 47-49). In contrast to my statement above, the risks to benthic ecosystems are likely valid wherever submarine volcanic eruptions occur and have been documented in many places. It would be worth citing some of that literature:

- Vroom, Peter S., and Brian J. Zgliczynski. "Effects of volcanic ash deposits on four functional groups of a coral reef." *Coral Reefs* 30.4 (2011): 1025-1032.
- Angulo-Preckler, Carlos, et al. "Volcanism and rapid sedimentation affect the benthic communities of Deception Island, Antarctica." *Continental Shelf Research* 220 (2021): 104404.
- del Moral, Roger. "The importance of long-term studies of ecosystem reassembly after the eruption of the Kasatochi Island Volcano." *Arctic, Antarctic, and Alpine Research* 42.3 (2010): 335-341.
- Shank, Timothy M., et al. "Temporal and spatial patterns of biological community development at nascent deep-sea hydrothermal vents (9 50' N, East Pacific Rise)." *Deep Sea Research Part II: Topical Studies in Oceanography* 45.1-3 (1998): 465-515.
- Marcus, Jean, Verena Tunnicliffe, and David A. Butterfield. "Post-eruption succession of macrofaunal communities at diffuse flow hydrothermal vents on Axial Volcano, Juan de Fuca Ridge, Northeast Pacific." *Deep Sea Research Part II: Topical Studies in Oceanography* 56.19-20 (2009): 1586-1598.

3. I'm somewhat concerned about the description of the origin of the PDCs. The authors did a very good job of explaining their use of the terminology and that these submarine mass flows are different from terrestrial PDCs given that they are water supported, but they make the claim that they likely originated by collapse of the eruption column and I don't think they've supported that claim (e.g., Line 114, 118-119). The alternative, that they originated from the submarine environment either via primary magmatic explosivity or hydrovolcanic explosions or simply mass wasting of the edifice, is not explored. If there is evidence that these originated from column collapse, the authors should include it, or at least note the alternatives.

4. No Tongan authors? I'm surprised that there are no Tongan authors on the manuscript. Surely they provided data or observations or significant science support for the work. I would think they should be on the author list.

5. More about the cores: The cores collected seem a primary piece of evidence for this study, but the data presented seems limited. There is a single core that is imaged and described in the main paper, but there were more cores taken. For the single core described, the image of the core needs to be much better. It is impossible to see the structure and a portion of the community would probably like to see it and interpret the potential processes it indicates for themselves. I would suggest that the authors add a visualization of ash fall and PDC thickness to the map of stations where total thickness is indicated by dot size and fraction of fall, flow in a pie chart at each site. This information should be used to help support the arguments in the paper of erosion and deposition inferred from the bathymetric differencing and modeling. Also, it would be useful to know whether any cores were taken that did not show evidence for deposition (in the refugia, for example).

Some more specific comments are detailed below...

Line Comment

23 It might be more appropriate to say more than 3/4 of all volcanic eruptions occur in the submarine environment, or more precisely more than 3/4 of all volcanic output. It would be good to cite Crisp et al., 1984 and maybe White et al., 2006 for this statement.

28 I would suggest: 'what some have suggested is the largest seaborne volcanic eruption'. I don't know that the erupted volumes have been accurately determined and I'm not sure if I would count edifice loss in the magmatic output that would make it a VEI 5.

30-31 Where does the information on the length of damaged telecom cable come from. I spoke with the crew who was fixing it and I did not get the impression that they were laying 100km of cable.

46 Again, suggest citing Crisp et al. and White et al. The cited papers aren't really the go-to literature for documenting global submarine volcanic output.

46-49 I would suggest removing the statement about risk in the Pacific Ocean. The vast majority of the 100,000 seamounts are extinct and are so remote as to not pose any real hazards even where they are active.

61-68 There is some missing literature here for studies that look at pre- and post-eruption seafloor conditions (notwithstanding that this study does not really have a comprehensive pre-eruption survey of benthic biology). I would include Carey et al, 2014 & 2018; Embley, 2014; Soule et al., 2009; Chadwick et al., 2008; Watts et al., 2015. It's up to you if you want to say that it hasn't been done on an eruption the scale of HT, but it doesn't seem necessary.

78 Should cite Garvin et al., 2018 that documented some of the bathymetric change at the early stages of this eruptive episode.

114, 118 As mentioned, the authors should explain why they think the 'PDCs' originated subaerially.

120 It's unclear what 'observation' is being supported by observations in this statement. Is it the origin of the PDCs from column collapse? It might be better to just say 'We observe'.

135 The statement about the volume needed to recreate the observed PDC distribution should be clarified. The model itself is not described, which is fine in the context that it is used - to define flow pathways. Because of this, I'm not sure the volume estimate is particularly relevant. However, if this would be considered a minimum estimate because of the model assumptions, the authors could indicate that and leave it in.

146 The mention of the Lau Basin here is a bit tricky. The authors have been in contact with a group that visited the Lau Basin and found evidence for HT ash deposition there, but nothing in the work described in the paper would necessarily predict that. To avoid undercutting the observations from the other group, it would be appropriate to simply say that the flows likely went well beyond the study area and not include mention of the Lau Basin.

150 Correct wording. Should be the discrete observation of a mussel bed or discrete observations of mussel beds.

182-184 The statement about suspended sediment and hydrothermal discharge needs some more explanation. First, there should be a reference to the supplement otherwise it is unclear what is being referred to in terms of suspended sediment. I'm also unsure why it is expected that suspended sediment will be a long-lasting effect. It should clear up in a matter of weeks to months unless something else is happening. In addition, I'm not sure why hydrothermal discharge is viewed to have a primarily toxic

effect? In most cases it can have a positive impact on ecosystems around the hydrothermal vents and a pretty limited impact on suppressing ecosystems locally and certainly not regionally.

Figures

Figure 2 This is a great figure that gives a very nice view of how density currents progressed and evolved. There is no description of the methods used to make the difference map or uncertainty associated with the estimates. There is clearly some error associated with rough terrain on features that are known not have changed, such as flank domes, that can arise from minor navigation offsets. It looks like that difference may not be stochastic (i.e., distributed around zero), but has a somewhat positive bias. It would be worth taking a look at the net gain/loss in areas that are known to not have changed to make sure the gain/loss is not biased. I'm also a little dubious of the spur to the NW, where pre-eruption data is sparse. There seem to be patterns in the gain/loss that look more like artifacts than real signal. I would be careful interpreting that and it may be better to leave it out of the analysis.

Figure 3 The modeled PDC paths in Figure 3 should be made less prominent. They are best guesses at flow paths and are rather complex and distracting in the figure. It would be nice to see the presence/absence and abundance of benthic life against the loss-gain map, or at least the average for that area could be shown next to the point. It seems that there are some points with benthic biota present near the caldera (to the east) that seem contradictory to the idea of complete destruction of habitat. Also, please use a higher resolution image of the core. It is impossible to see.

Figure S3 It is hard to see the symbols on the geochemical plots. A higher resolution version should be included. Also it seems that the T2206 data is just plunked on an older figure and covers up some of the previous data. Also, the legend is missing a symbol for T2206 shards.

Reviewer #3 (Remarks to the Author):

This manuscript compiles observations from ship-based cruises to describe the seafloor impacts of the January 15, 2022 Hunga Tonga-Hunga Ha'apai eruption. In addition, they describe results from a PDC model that they plot against observations. Observations include changes in seafloor bathymetry, changes to biologic communities, and reports on the characteristics of the sedimentary deposits. Due to the remarkable nature of the 15 January Hunga Volcano eruption, I suggest that these results are worthy of publication in a high impact journal such as Nature Geoscience. The changes in seafloor bathymetry, existence of pyroclastic density current deposits, and runout distances of these pyroclastic current deposits will be of widespread interest to the volcanological and oceanographic communities, in my opinion. That said, some of the interpretations are not currently well-supported by information provided in the ms although I do not doubt that this data was collected. I also have significant questions about the PDC model. After these and other changes are addressed, I would highly recommend publication as I believe these results are of particularly high impact and importance. Please see my attached comments.

-Kristen Fauria

Review for Pyroclastic density currents explain far-reaching and diverse seafloor impacts of the 2022

Hunga Tonga Hunga Ha'apai eruption

This manuscript compiles observations from ship-based cruises to describe the seafloor impacts of the January 15, 2022 Hunga Tonga-Hunga Ha'apai eruption. In addition, they describe results from a PDC model that they plot against observations. Observations include changes in seafloor bathymetry, changes to biologic communities, and reports on the characteristics of the sedimentary deposits. Due to the remarkable nature of the 15 January Hunga Volcano eruption, I suggest that these results are worthy of publication in a high impact journal such as Nature Geoscience. The changes in seafloor bathymetry, existence of pyroclastic density current deposits, and runout distances of these pyroclastic current deposits will be of widespread interest to the volcanological and oceanographic communities, in my opinion. That said, some of the interpretations are not currently well-supported by information provided in the ms. I also have significant questions about the PDC model. After these and other changes are addressed, I would highly recommend publication as I believe these results are of particularly high impact and importance.

General Comments

First, I want to congratulate the authors on their incredible work surveying the Hunga Volcano post-eruption. Their findings on PDC runout and the lowering of the caldera floor are remarkable. My comments and suggestions below are primarily aimed at clarifying my own questions and making sure their observations can be well understood and utilized by readers and scientists studying the Hunga eruption.

The change in bathymetry is a major finding of this manuscript. Because of its importance I would suggest clarifying both the bathymetric observations in the text and improving the quality of the difference map in Figure 2. Specifically, there is no absolute scale on the colors used in the difference map. I cannot see a difference between the remarkable >800 m loss in the caldera and more modest loss along the flanks. I also cannot tell, from the text or figure, how much variation there is in deposition or scour along the flanks. The ms could also benefit from a brief description of the pre-eruption Hunga Volcano morphology (i.e., that it is a caldera volcano) before a discussion of the topographic changes is introduced.

The authors claim that the areas of deposition on the volcano's flanks are due to PDC deposition. Although this is reasonable, the claim is not well-supported by sedimentological observations in the text or supplement. More information along the lines of the stratigraphic section in Fig. 3 would be helpful. For example, the authors discount sedimentation by landslides without explaining why or defining what they consider to be a "lack of evidence." The deposits are interpreted to be PDCs in part due to bedforms and sediment cores, but the features within the cores or bedforms that distinguish the deposit as PDC are not mentioned or defined. Although thorough work on the sedimentary structures within their cores and from video observations may be outside the scope of the paper, I suggest that the authors describe

what specific aspects of the cores and seafloor observations allow them to define the deposits as PDC related (rather than mass flow by other flank collapse, for example). In the end, I agree that these are PDC-type flows but based on their spatial distribution rather than the sedimentological evidence provided.

I found the authors findings on post-eruption venting compelling. That said, the descriptions of phenomena such as post-eruption venting are only vaguely described. The ms would benefit for more clear (even if in the supplement) descriptions of where venting was and was not observed and how.

I am pleased that the authors employed a PDC model and compare the model to observations. I have major questions about the model, which I detail at the end of this review. One main concern is that it is not clear however, what aspect of the PDC model they are comparing to what characteristic of the observations. There is some discussion of runout. However, how is PDC runout characterized in the model and in the observations? How do the authors determine the flow direction (arrows on Figs. 2 and 3) from the model? The authors acknowledge that the model does not include deposition. How, then, is the ultimate flow distance in the models determined (what process stops the PDC in the model)? I agree with the authors approach of employing a general model, but it is not at all clear how their model works.

Line by line comments

Line 63: I suggest being specific about the types of data used earlier in the ms and on this line.

Line 68: Add “mass” to the description of “loss.”

Line 79: The eruption on Jan 13-14 could be described as a single large event rather than a series of events before the 15th (e.g., Gupta et al., 2022).

Line 87-91: Here the Hunga summit is mentioned for the first time and it would benefit with a brief summary of context. I suggest briefly describing the pre-eruption bathymetry to give the readers context for where bathymetric change occurs.

One of the most notable findings is the >800 m change in the caldera. It is mentioned here, but the reader would not know Hunga is a caldera volcano, without prior knowledge, and would not know the pre-eruption caldera depth. I suggest adding those things here. Could you also comment on whether this change in caldera floor elevation is uniform across the caldera? It also seems crater and caldera are used interchangeably. Is there a difference between the crater and caldera? What kind of change occurred along the caldera rim?

Lines 89-91: Could you expand upon the volume loss along the caldera rim. Was the mass loss uniform along the flanks or localized? If uniform, what was the average change in vertical elevation along the flanks? How far from the caldera rim did the mass loss extend? That is,

where was the transition from erosional to depositional features and how uniform was that distance from the caldera rim?

Line 93-94: Ashfall is mentioned here and subsequently. However, the ashfall portion of this eruption is not described in your ms. Could you briefly describe what is known about the presence of ashfall?

Line 95-97: The supplementary data table is cited as evidence here. I am so glad you have included these supplementary data. However, what aspect of the data are you suggesting makes this point? Could you perhaps provide a plot of the data in the supplement to make this point?

Line 98: "There is consistent loss" is rather vague. What is consistent about the mass loss (spatially, mean change in elevation?). Why choose a water depth threshold? Does this correspond to a mean distance from the caldera rim? If so, what is that distance?

Line 102-103: I assume here you mean chute termini. Could you make more clear the transition in discussion from erosional to depositional features.

Line 106: What do you consider an absence of evidence of landsliding? Could you briefly list the features you would look for to indicate landsliding?

How much erosion and deposition occurs outside of the "chutes". Could you also describe the change in bathymetry in the areas not defined by "chutes".

Lines 113-114: The process-based description of PDC formation here is a suggestion or interpretation rather than an observation. I suggest, therefore, caveating this sentence with a phrase such as "we suggest". I also wonder the extent to which the flows may have never entered the atmosphere and remained entirely submerged since the vent was submarine? This possibility could be acknowledged even if only for a portion of the flows.

Line 118: Please describe the plume collapse as "partial" since the plume attained a height of 57 km.

Line 120-121: Could you please remind the reader how these observations were made. Overall, these scours are remarkable and could be discussed or emphasized more than they are at present (purely optional).

Line 126: Could you please remind the reader the scale of a granule? Using what methods was grain size evaluated and quantified?

Line 124-125: How far was this from the vent?

Line 128: Here an earlier description of ashfall could be helpful. What sample did you compare against to determine they were from Hunga Tonga?

Overall, the finding that the PDCs go from erosional to depositional is very interesting. As mentioned earlier I would love more information on the dimensions (thicknesses, distance from the caldera) of the erosional and depositional components. Could you also put the vent location in the context of these observations. It seems that you assume all material was sourced at the vent, which was on one side of the caldera, yet the erosional and depositional features are evenly distributed around the caldera? Could you comment, confirm, or clarify in the ms?

What were the maximum and characteristic runout distances of the PDC for both the channelized and unchannelized components?

Line 135-136: Please describe the basic attributes of your model here. Was the model for a dense or dilute flow? What were the basic assumptions or initial conditions? Did you model a PDC in water or in air, for example? Here you mention replicating the extent, are you referring to the depositional extent? How did you do this comparison in practice, especially if the PDC model does not include deposition?

Line 137: Here you introduce the concept of simulated flow pathways. How are these defined? I am unfamiliar with PDC models being simplified into the arrows shown on your figures. On what basis are you defining these pathways?

Line 140: What is meant by broken cables along >30 km and >100 km lengths? Is this distance from a location? If the cables were not broken in one location, could you please describe?

Line 144: I do not understand this comparison. Are you comparing the model results to observations from areas other than Hunga?

Line 144: What was the longest PDC runout?

Line 163: What aspect of the PDC model is being compared to locations of the refugia?

Line 185-186: How was turbidity measured and where and when? Was there excessive turbidity everywhere or in specific locations (both spatially and in water depth)? Could you expand upon the definition of nepheloid layers as this may be a new concept to many volcano readers? Could you make Fig. S10 easier to interpret to substantiate this point. For example, the acoustic image is not annotated and has no scale, date/time, or specific location (lat/lon). The ongoing venting is of particular interest and currently it is only vaguely described. Where and when was venting found? Can you determine if/when the venting shut off?

In the conclusions, I suggest providing a brief summary of your bathymetry change and PDC erosion/deposition observations.

I am not a petrologist and so cannot evaluate the details of their methods for determining the ash composition. It could be more clear what sample they compare against to determine the true Hunga composition.

Fig. 1: Can you please make clear if the two A to A' transects and B to B' transects are the same or different. If location A is location B and location A' is location B', I suggest using the same label for both.

Fig. 2: How do you choose how many flow trajectories to plot here from the model, why not 3 or 300? Please provide an absolute scale for mass gain/loss. What is meant by "diffusive flow"? I also don't see any evidence for interaction with topography as annotated on the figure.

Fig. 3: Is this sedimentary log based on a single point or a general summary? To what extent is this log representative? Overall, this is a nice figure. I have the same issue with the flow pathways. I would like to see scale bars on A-C and the other panels labelled with letters as well.

Line 406-407: How much footage (distance or time)?

Line 505: Why 50 cm gridding of the data. This seems very high resolution for ship-based bathymetry. Wouldn't the natural resolution be much lower.

Please add scales to all images.

Could you please make sure there are geographic coordinates included in all sampling locations (e.g., Fig. S1 and data tables).

PDC Modeling Comments:

PDC modeling is challenging and I think it is neat that the authors undertook this challenge in addition to their substantial ship-based work. Overall, I cannot understand the model based on the information provided. Please see the questions below.

- Did you consider the current dilute or dense in your modeling effort? Your choice would result in different conservation equations and assumptions.
- I am unfamiliar with this modeling software for PDCs. Can you describe how common it is for PDCs compared to other types of models such as VolcFlow, Titan2D, and other known codes? Why did you choose this modeling software?
- What are the governing equations for mass, momentum, and energy in your model?
- The two-layer choice is interesting, but I don't understand how it is set up. Why two layers with a water layer on top? The water layer is the ambient fluid so shouldn't be considered part of the flow. How does this work?
- How was flow through water dealt with since most PDC models consider flow through air. That is, how did you treat the ambient fluid (as water or air)?

- Was the model 1D, 2D, or 3D?
- Is the model single-phase?
- What controls runout distance in the model. How does the model stop? This is particularly important since runout is one of the features you compare.
- How is the model coupled to topography?
- Was the initial volume of PDC material placed on the seafloor, at the water surface?
- How did you parameterize key aspects of the model (e.g., friction coefficient, Froude number, Richardson number, or entrainment coefficient)? I don't understand enough about your model to know what you needed to parameterize, but models that aren't 3D include these types of parameterizations.
- Can you clarify your boundary conditions.

We thank the reviewers for their time spent reviewing this manuscript. We have endeavoured to address every comment, with responses in bold and italicized text following each comment. We have in-line track changes in the revised manuscript, with corresponding text addition and line numbers indicated here. We have provided additional supplemental information as well as the code (in a zip file) for your review as well. We thank the reviewers for their useful feedback which we believe has made this manuscript clearer, more robust, and better all around.

RESPONSE TO REVIEWER COMMENTS

Reviewer #1 (Remarks to the Author):

This article describes timely observations made of the 2022 eruption site and accompanying analyses. Fortuitously, multibeam sonar data of good quality were available from before the eruption, collected sufficiently recently that the depth changes are almost certainly due to the eruption. There has been a lot of interest in using depth changes around volcanic areas at smaller scale to study effects of volcanism (e.g., Casalbore et al.; Watts), which this study adds to. Seabed photographs from prior expeditions were available also for comparison with new bed photos. The authors document sediment core and bottom photographic evidence that indicate the pathways travelled by the submarine pyroclastic flows. The area will further become a benchmark for studying how seabed fauna recover from such events.

We thank R1, Dr Mitchell, for recognizing the importance of this study and the contributions that this work will make to the field.

Given the widespread media coverage of this event, I would not be at all surprised if this article will become highly viewed. I hope I don't appear to be diminishing the role of the Tonga eruption and the damage it caused, though the importance to hazards research probably lies with it helping to understand the risks posed by similar magma bodies nearer to major population centres, such as the Phlegraean Fields near Naples. To the north of San Miguel Island of the Azores, another field of giant sediment waves can be found (Weiss et al. (2016); Chang et al. (2022)). They lie north of the Furnas suspected magma body of similar concern. The article mentions the scale of the eruption, though it might have been useful to read a magnitude comparison with other known eruptions, rather than leaving that to other publications.

We agree that there may be larger populations exposed to such hazards in other regions and we recognize the importance of this work to understanding potential events at similar magma bodies of concern. Therefore we have highlighted this further in lines 52-54 to expand on the wider implications to more populous areas.

"In particular, shallow water active volcanic centres that are near populated islands expose a major blind spot regarding risk assessment and response preparedness³⁻⁵."

To place this in a more useful context, we have also added the Volcanic Explosivity Index to line 57 to provide a magnitude comparison to other known eruptions.

"With a Volcanic Explosivity Index of 5.7, the 15 January eruption was the largest eruption since Mount Pinatubo in 1991 and is one of the most explosive submarine eruptions ever recorded and largest documented in a century or more."

However, we are also keen not to remove too much focus from small island developing states in the South Pacific that may contain smaller numbers of people, but may be far more exposed and vulnerable, due to them lying so close to sea level, with a far greater reliance on one or relatively few telecommunications connections, and with a much reduced or a total lack of hazards monitoring capacity (thus inhibiting early warning).

Despite my enthusiasm, I found reading the article somewhat frustrating, as it dips a toe into many topics without exploring them in great detail. At the end of reading it, I would certainly feel happy citing this article for the effect on the seabed around the vent and as another indicator of the scale of the eruption, though I'm not sure it will get cited so much for specific insights. This is illustrated by the abstract, which emphasizes both the geological event and biological impact. As a geoscientist, selfishly I might have preferred more focus on the geological event.

As R1 indicates, a key aim of this paper is to highlight the wider impacts of the Hunga eruption, encapsulating geologic and biologic impacts and shining a new light on the diverse, and typically-ignored, seafloor impacts of submarine eruptions. The broad interest that we seek to attract with this manuscript construction is why we think it is an ideal fit for Nature Communications, with more specific insights about the Hunga eruption saved for follow-up papers in a more specialist geologic journal. The multidisciplinary data acquired from this event will enable a host of follow on papers, but we believe it is important that this paper, which synthesises our new understanding of those diverse impacts, is published first to set the scene for more discipline-specific and specialised studies.

For example, only the optimum results of modelling of pyroclastic density currents are shown, but this seems a rare example where we can be sure the deposits can be associated with an event (almost unequivocally). Sensitivity tests (varying volume and density of the originating flow) might have been explored and inferences drawn from comparison of those results with the field data.

This is an entirely reasonable comment and we have now provided further information on the PDC modelling, which includes the outcomes of different model runs in the supplementary material that provide a sensitivity assessment as asked for by the reviewer. We investigate the sensitivity of the results on the initial volume and the density of the PDC gravity current and an alternative fountaining initialisation. We would not claim that the results we show are optimum as the model is very simple and the constraints fairly wide, but it is a plausible scenario and we have expanded on what we can infer from the modelling results, as seen throughout the revised text.

Although the modelling was very simple as the authors admit, I wondered if the spatial pattern of differences may allow at least a discussion of possible range of scale of the submarine PDCs. The flows evidently lacked sufficient kinetic energy to over-top the surrounding seamounts and friction (bed shear and momentum transfer from water entrainment) - this seems a basic constraint on them.

To the south of the Hunga edifice, the PDCs did in fact overcome bathymetric constraints and make it over submarine knolls into the valley where the international cable lay. We have added a sentence to clarify this point in the main text (lines 172-173: "... and the overtopping of two submarine knolls to the south heading towards the international cable").

Smaller PDCs that were modelled did not reach that far and mainly ended up in the valley that the domestic cable was in, this can be seen in the different model run outcomes that we have provided in supplemental material (Figure S6 top right panel).

Do the multibeam data suggest the elevation that the PDCs reached? The areas of bed loss in Figure 2 around the seamounts suggests that they may have. Equating potential energy gain for kinetic energy implies the velocity would be at least $\sqrt{2gh}$ ['at least' because this neglects friction]. Given that velocity (from which bed shear stress can be estimated), is it surprising or unsurprising that widespread erosion occurred over the edifice and along the NW channel? This is just an illustration - it seems to me a punchier outcome could be developed.

Due to the water depths in which the multibeam data were acquired, the resolution of these data does not permit the robust identification of trim lines on the seamounts to infer flow thickness nor to confidently resolve the absolute elevation that the PDCs reached (i.e. not in the same manner

as has been done on other studies such as Stevenson et al., 2018. Reconstructing the sediment concentration of a giant submarine gravity flow. Nature communications, 9(1), p.2616. **There the acquisition of side-scan sonar data and greater coverage of sediment cores enabled mapping of trimlines. We agree with the reviewer that this is an interesting area to explore and we do propose to perform more detailed analysis on the sedimentological characteristics of the PDCs in a follow-up paper that is under review. In time, we may be able to provide deeper insight into some of these points; however, this is beyond the scope of this present manuscript as we are seek to provide a broad, multi-disciplinary view of the seafloor impacts of the Hunga eruption and demonstrate the widespread but otherwise unseen impacts of the eruption.**

Detailed suggestions:

Line #31 Is it possible to be more specific concerning biological communities? Fauna? To be pedantic, if sampling for bacteria, there may have been some carried in by the ocean. On the other hand, it might not be surprising not to see whales as they are not so abundant.

We have adjusted the text accordingly (lines 34-35): *Biological (mega-epifaunal invertebrate) seafloor communities...*

40 The ash and chemicals from large volcanic...

Line 43 now reads: *The ash and chemicals from large volcanic...*

48 height

Line 52 now reads: *...volcanoes of >1km in elevation...*

60 data from surveys before and after eruptions that can be used to quantify ..

Lines 67-68 now read: *...and data from surveys before and after eruptions that can be used to quantify...*

77 The text could be made more succinct in places. Here for example:
began erupting in late ...
Put a full stop after 2021.

Lines 94-96 now read: *After this eruption, Hunga Volcano was largely inactive until it began erupting in late 2021, producing a steam-rich gas and ash plume. Following this, Hunga Volcano continued erupting intermittently until two large eruptions...*

83 I would not put quantities like 99% unless they have actually been measured. Instead, use "majority"

The following sentence has been added to supplementary material (Pre-eruption multibeam section in the methods, lines 656-659) to explain how the quantity 99% was measured:

"To determine the percentage of the volcano area that is subaerially exposed the area of the submerged edifice defined by a change in slope of 3° the base of the edifice. This was compared to the area of the pre-eruption island area to demonstrate that <1% of the volcano is subaerially exposed".

85 A map of bathymetry change derived by subtracting ... from ...

Line 101-102 now reads: *A map of bathymetric change (derived by subtracting new bathymetric data from pre-eruption surveys in 2015-2017^{6,7}) reveal the seafloor morphologic fingerprint of a major eruption.*

88 Integrating depth changes over selected areas of the map, this change ...

Lines 104-105 now read: *Integrating depth changes over selected areas of the map, this change accounted for 6.0km³ seafloor loss from within the caldera..*

Shouldn't these polygons be shown in the electronic supplement? How could these results be reproduced?

A shapefile with the polygons used has been included as a figure within the supplemental information (Fig S18) as indicated now in the methods section : Line 641: "*...for select areas of the map (shown in Fig. S18).*"

90 small features

Line 108 now reads: *with many small features, such as..*

97 I would think the volume ejected in the eruption column would also be quite uncertain. How does that change this assessment?

We refer to the initial estimates of volume ejected from the volcano. We acknowledge there are updated values and uncertainty in these values, and we have modified the sentence to reflect this: (Line 111: *Prior studies suggest that ~1.9km³... remaining material (~1.3km³)...*). This uncertainty also includes the thin deposit on the seabed that we cannot resolve using modern echosounder techniques, which we have further clarified in this paragraph (Lines 112-113: *widespread thin deposits below the detection limit of what can be resolved with modern echosounder techniques*)

99 varied

We have addressed this in the manuscript, line 117 now.

100-101 I found it hard working out the details from the maps shown (I can see the areas of sediment waves in Figure 1 roughly correspond with the major depth changes, but not much more than that). It might be useful to plot some depth contours from the post-eruption survey over the depth changes in Figure 2.

We have added contour lines to Figure 1. Given the broad readership of Nature Communications we prefer to keep Figure 2 as a difference map without contour lines as we worry contour lines over the difference map could add confusion.

101 Up to 70 m of entrenchment was ...

Line 120 now reads: *Up to 70m entrenchment was..*

106 Could the NNW entrenched channel have initiated as a landslide? Might landslides have occurred in the area marked as "diffusive flow" in Figure 2? They look difficult to rule out. Maybe instead say that there is no superficial morphology typical of landslides remaining, e.g., crown/headwall.

We have addressed this in the manuscript. The sentence (lines 126-129) now reads:

“There is no superficial morphology consistent with seafloor landslides (i.e., headscarps and/or scars) on the volcano flanks despite the scale of the eruption and associated seismicity, which makes this event distinct from partially-submerged eruptions that generated large slope collapses such as Anak Krakatau (Indonesia) in 2018³⁷.”

122 I was left not knowing really what was meant by "powerful density flows" in terms of magnitude. Can the scale (wave spacing and height) be compared with features created by known (measured) currents or even inferred from cable breaks? The gravel waves produced by the Grand Banks failure have been well studied for example.

The scale of bedforms is comparable to those that have been formed by large volume and fast-moving sediment density flows such as the 1929 Grand Banks and the 2016 Kaikoura Canyon events. We therefore infer that these flows were also similarly powerful. We also know that they were locally capable of running up slope so must have had sufficient inertia to do so. We are working on a more detailed follow up paper that provides further insight into the initiation timing, cable breaks and relates the bedform morphodynamics to likely flow velocities and properties and there is not sufficient space here to properly expand. To address this point we have added the following text to explain by what is meant by “powerful”, providing context with published studies on the Kaikoura and Grand Banks events

Lines 149 to 154 : “We observe crescentic scours (up to 30 m deep) and bedforms (up to 1,200 m wavelength, 20 m wave height; Fig. 2), commonly observed in other deep-sea settings affected by powerful density flows (including those triggered by large magnitude earthquakes, that severed seafloor cables, comprised a dense basal layer, and have been shown to attain speeds of up to 20 m/s), and further corroborated by seafloor sediment coring and imagery, geochemical analysis and numerical modelling”.

125 Without dates, strictly speaking we don't know if the deposit was from the eruption (though likely) or an earlier eruption. Is there anything further about the deposit in the core that indicates it is young? Perhaps lack of any iron-manganese coating of grains near the surface, lack of bioturbation, glass shards are unaltered, etc?

We have added a sentence (lines 165-167) to clarify points that indicate the deposit are from this most recent eruption: “A lack of bioturbation within the volcanoclastic deposit and no indication of oxidation indicate that it is a relatively fresh deposit, as well as the absence of any hemipelagic layering on top of the volcanoclastic deposit.”

It is also worth noting that we have sampled locations where there is clearly resolvable change from the bathymetric difference mapping (up to 22 m thickness of new deposits); hence any seafloor deposit there cannot pre-date the eruption.

131 Bathymetry difference mapping ... were predominantly erosional on the

We have addressed this in the manuscript, lines 169-170.

135 ... of sediment laden flow ...
? or particle volume (DRE)?

Line 174 now reads: ...of sediment laden flow

139 Do we know the timing of the cable breaks (presumably yes)? If so, what was the delay between appearance of the atmospheric eruption column and these breaks? Does that tell us anything about how the base of the eruption column collapsed?

This is a very good question and is the subject of ongoing research that specifically focuses on this aspect of the event (rather than the compounded impacts as a whole). As a result, this cannot be explored in detail in this present manuscript.

We have modified the text to reflect the model outputs as well (lines 179-183):

“The model is consistent with the locations and extensive damage to the international and domestic telecommunication cables, damaged along 89 km and 105 km length, respectively (Fig. 3), and explains the observed spatial trends in erosion and flow runout.”

195 In my opinion, the conclusions ought to focus more on the results of the marine work and modelling, not broader social issues that have not really been developed in the article.

The conclusion section has been modified to address this comment.

Lines 263-271: “Our data illustrate how an eruption of a single submerged volcano can have wide-reaching and diverse impacts. PDCs resulting from the Hunga Volcano eruption were steered by antecedent morphology, further excavating pre-existing chutes and depositing the majority of the erupted volume on the seabed within 20 km of the Hunga caldera rim. Modelling and benthic observations indicate that the majority of the PDCs originated within the caldera and flowed around bathymetric relief as they moved to deeper waters, with only local preservation of benthic communities on topographic refugia. Outside of these topographic refugia the seafloor ecosystems around Hunga Volcano were decimated by the eruption and related PDCs.

Figure 1 - quite hard seeing much detail in these maps. I can see the sediment waves. Can the maps be shown enlarged?

The intention of this overview map is to illustrate that the dominant change to the volcano is within the caldera and that the changes to the flank are much more subtle. Given the scale of the eruption, the relatively minor morphological changes to the flanks, compared to the caldera, are a key result – an initial surprise to us and in stark contrast to other submerged volcanoes such as the half-island collapse of Anak Krakatau in 2019. The details on the local changes on the flank are provided in Figure 2. However, in order to make it easier to see, we have increased the size of panels b and c so that they occupy a full-page width.

I also wondered if a traditional figure with overlain depth contours might also be useful, as the contours can help accentuate the channels and minor gullies.

We have added depth contours to Figure 1.

366 insert (grey areas) before "2017".

We have addressed this in the manuscript, with a sentence added within the figure caption for Figure 1 at line 485: *This is overlain on GEBCO_2022 bathymetric grid (greyscale..).*

Figure 2 is annotated with the author's interpretations of processes, rather than features in the map. Traditionally, readers like to see the evidence presented in a non-genetic way so that they

can assess it alongside basic observations, before arriving at the interpretation. I would suggest replacing these terms with observations.

We have removed the annotations from Figure 2

The scale bar on the map needs values even if the same as in the cross-sections graph below (or otherwise say so in the figure caption).

We have added values to the scale bar on the map in Figure 2

372 Please be more specific about what is meant by flow trajectories. E.g., do these represent motion of a virtual particle in the flow or flow front?

We have clarified this in the figure caption, lines 491-493: "*Indicative PDC flow paths (i.e., the passage of the flow fronts as estimated from model animations and snapshots (see Supplemental Video 1 and Fig. S6))*"

Figure 3
377 modelling results

We have addressed this in the manuscript (line 497-498): "*with indicative PDC flow paths*"

381 and each image is located in...
Sediment core is located by ...

We have addressed this in the manuscript, lines 502-503: *The sediment core is located by light blue star.*

383 with interpreted pre-eruption ...

We have addressed this in the manuscript, line 488.

405 I don't know what "abundance of biological assemblages" means exactly - different species, individuals?

We have clarified this in the text, lines 527-528. "*The biological assemblages are mega-epifaunal invertebrates, living on or above the seafloor with sizes >2cm.*" and lines 533-534 "*abundance of invertebrate taxa were counted*"

416 please write in the past tense, as it is unclear as written if the CAT scans were useful or if useful in general (maybe not there).

We removed this text from the manuscript as we agree it was confusing as written and not necessary. The removed text is seen between lines 542-547.

425 In our study of cores from the Azores, volcanoclastic turbidites were found to have no organic carbon (well below threshold) - Chang et al. (2021). It seems obvious that there wouldn't be, though was organic carbon detected? This is a minor further check.

Organic Carbon was not analysed from the cores. For this study only major elements were measured to establish that the material was sourced from HTHH. Full analyses of the cores,

including carbon, will be undertaken as part of a more detailed study on the composition of the ash layers.

479 It might be useful for readers to know what this means in terms of location precision and accuracy.

This is provided in the methods section, line 649 *“uncertainties for multibeam acquired bathymetric data are depth dependent and are usually reported as being 1% of depth (ranging from 20 cm to 20 m).”*

486 Were water-column data recorded with the EM710? This might be useful information for later studying impact of turbidity on pelagic fauna.

Yes, and this will be looked at in works in preparation and released for public use.

521 It also varies with beam direction and with noise (Schmitt et al.). The final difference grid (Figure 2) does not contain any obvious known artifacts (e.g., Hughes Clarke et al.), which is good.

523 This is not quite true, some can look like geology (de Moustier & Kleinrock) but the lack of them in the change map is reassuring.

To address the two points above we added to lines 636-637: .. *“However, morphology as measured by multibeam echo sounders is typically not affected by vertical uncertainties, and if so we would expect artifacts in the difference maps which we do not see.”*

535 This is a high density, it seems to me, if the deposits are nearer 2500 kg/m³. I thought it a shame the volume and density were not explored further with sensitivity tests (ie., what range of these values leads to a breakdown in match to the observations and what can we learn about the flow (or limits of the modelling) from that comparison?).

The PDC modelled here is actually a underwater density current made up of a mixture of the dense part of the PDC and water. The density is taken to be 1,600kg/m³ which assumes that the PDC flow is 40% (by weight) dense pyroclastic material and 60% water. Sensitivity tests were undertaken and the results of some of these are shown in the supplementary material (SI Fig 6,7). The model is fairly insensitive to changes in density of the PDC flow. The bottom two panels of Figure S6 show model runs with lower and higher density at time t=2,400s. It can be seen that the flow does not reach quite as far for a less dense flow (1,400kg/m³, 26.7% pyroclastic material, bottom left panel of Figure S6) and reaches slightly further for a denser flow (1,800kg/m³, 53.3% pyroclastic material, bottom right panel of Figure S7) but in both cases the differences are fairly subtle.

The areas of compliance and non-compliance of the model results have not really been explored.

The simplicity of the model means that there is limited use in more than a qualitative comparison between the model and the observations, but we now discuss some of the implications of where versions of the model do or do not agree with the observations in the Methods Section. (lines 710-730)

“To investigate the sensitivity of the PDC flow paths to the location of the originating PDC, an alternate initialisation was modelled starting with a annulus of material between five and seven

km from the centre of the caldera (thus originating outside the main caldera) and 53m thick (so also approximately 4km³ volume of material) but with a radial velocity of 10m/s to ensure that it flowed outwards rather than falling back inwards to the caldera. This initialisation shows that the strong channelisation of the flow on the flanks of the Hunga edifice that was observed during the voyage does not occur when the PDC originates outside the caldera (see Fig. S7 top left panel). Because this channelisation was clearly observed, that is evidence that the PDC must have originated within the caldera.

The total volume of PDC material in this model was chosen so that the gravity current reached the international cable with sufficient force to be able to cause the damage observed. Because of the simplicity of the initialisation, we have to be careful with interpretation, but smaller volumes were also modelled and volumes of 2 km³ or less do not extend that far (see Fig. S7). This suggests that a significant proportion of the volcanic material must have been ejected within a short period of time as a slower, more continuous, rate of PDC formation would not have the energy to reach the international cable. Sensitivity tests on the density of the PDC show that the distance the flow reaches is relatively insensitive to the density of the gravity current with only minor differences being observed for densities of 1,400kg/m³ and 1,800kg/m³ (see Fig. S7, bottom panels)."

536 no need for "back"

Lines 688-690 : "This could represent either the point where the eruptive column has collapsed and mixed with the sea water or where an effusive PDC has mixed with the seawater."

Electronic supplement:

Figure S10. Please mention the sonar and frequency used to produce the water column backscatter image lower-right.

This has been added to the supplemental figure caption : "(collected with the Kongsberg EM710, which has 65-100 kHz frequency)."

I very much enjoyed reading the results of this study and look forward to seeing a revised version in print.

Neil C Mitchell, University of Manchester, January 2023.

Casalbore D, Di Traglia F, Bosman A, Romagnoli C, Casagli N, Chiocci FL (2021) Submarine and subaerial morphological changes associated with the 2014 eruption at Stromboli Island. Remote Sensing 13:article 2043

Chang Y-C, Mitchell NC, Hansteen TH, Schindlbeck-Belo JC, Freundt A (2021) Volcaniclastic deposits and sedimentation processes around volcanic ocean islands: the central Azores. In: Chang Y-C, Mitchell NC, Quartau R, Hübscher C, Rusu L, Tempera F (2022) Why are submarine sediment waves more common on the north sides of the Azores volcanic islands? Marine Geology 449:art. 106837

de Moustier C, Kleinrock MC (1986) Bathymetric artifacts in Sea Beam data: How to recognize them and what causes them. J. Geophys. Res. 91:3407-3424

Di Capua A, De Rosa R, Kereszturi G, Le Pera E, Rosi M, Watt SFL (eds) Volcanic Processes in the Sedimentary Record: When Volcanoes Meet the Environment. Geol. Soc. Lond., London

Hughes Clarke JE, Mayer LA, Mitchell NC, Godin A, Costello G (1993) Processing and interpretation of 95 kHz backscatter data from shallow-water multibeam sonars. In: OCEANS'93 Proceedings. IEEE, Victoria, B.C., pp 437-442

Schmitt T, Mitchell NC, Ramsay ATS (2008) Characterizing uncertainties for quantifying bathymetry change between time-separated multibeam echo-sounder surveys. Cont. Shelf Res. 28:1166-1176

Watts AB, Nomikou P, Moore JDP, Parks MM, Alexandri M (2015) Historical bathymetric charts and the evolution of Santorini submarine volcano, Greece. *Geochem. Geophys. Geosys.* 16:847-869
Weiss BJ, Hübscher C, Lüdmann T, Serra N (2016) Submarine sedimentation processes in the southeastern Terceira Rift/São Miguel region (Azores). *Marine Geology* 374:42-58

Reviewer #2 (Remarks to the Author):

This manuscript represents an important contribution, marking the first documentation of the impacts to the benthos of the Jan 2022 Hunga Tonga eruption and providing some key observations on the physical processes of subsea dispersal of volcanic material. I think the paper could be improved by refining the message about the hazards posed by this eruption relative to the potential hazards of all submarine volcanoes. In addition, the authors should consider more description of the data and models and de-emphasizing some of the interpretation - given that there may not yet be sufficient information.

Below I describe a number of suggestions that I think would significantly improve the paper. If addressed, I highly recommend its publication. The eruption itself and its clear impacts on local populations were highly publicized and of broad interest to the scientific community and the public. In addition, there were well-documented global impacts that captured the interest of the world (e.g., pressure waves). These results will likewise be of high interest because they reveal the subsea impacts in some detail for the first time. The paper is well-constructed and reports high-quality data that advances our understanding of this important volcanic event and its impacts. Below I describe some of the potential improvements to the manuscript.

We thank Reviewer 2 for their review of this manuscript, their constructive feedback that has improved the quality of this manuscript, and their encouragement. We have done our best to address all comments as detailed below.

1. There are many superlative claims about the eruption and data collection that I view as distracting at best and disingenuous at worst. The high interest of the topic speaks for itself does not require the authors to go out on such limbs. Nature should accept some responsibility for this as, intentionally or not, it seems to encourage such language in its publications. When it is accurate, I'm all for it, but in this case I don't think there is a strong enough case to merit it. Some specific examples...

line 28 "largest seaborne volcanic eruption in nearly 150 years" is based on a citation of a letter that does not provide any documentation of eruption volume, but instead cites a YouTube presentation. The authors note that a significant portion of the volume erupted is removed from the existing edifice. Perhaps it is the largest submarine eruption of the last 150 years, but I don't feel like that it is currently well-supported and it would be simple enough to say "one of the largest documented seaborne volcanic eruptions in perhaps a century or more".

line 86 "reveal the seafloor impacts of a major submarine eruption for the first time" seems to ignore several decades of work documenting submarine eruptions. Most noticeably there is no reference to Carey et al., 2014; 2018 that documented seabed impacts on Havre volcano that erupted in 2012.

We have adjusted statements in the text with these points in mind, in order to reduce superlative claims. We have changed "largest" to "most explosive" where required (lines 30 and 58) to separate this eruption from other large volume but less explosive events and we have added a reference for this explosivity in line 30. We have also qualified the section around revealing the impacts with regards to other eruptions. What makes this study unique is the availability of pre and post eruption bathymetry, which allows an analysis for an eruption this size that has not previously been possible. This is now stated in lines 67-69: "*Most submerged volcanoes are poorly mapped, and data from surveys before and after eruptions that can be used to quantify pre- and post-event seafloor conditions are rare*²⁹⁻³². *No such data currently exist for an event on the scale of the 2022 Hunga Volcano eruption.* "

Other related changes:

Lines 57-58: We added reference to the VEI of the eruption and reworded the original sentence so that it now reads: " *With a Volcanic Explosivity Index of 5.7, the 15 January eruption was the largest eruption since Mount Pinatubo in 1991 and is one of the largest documented seaborne eruptions in a century or more.*"

We removed "for the first time" from line 103.

We added reference to Carey et al., 2018 in line 68, and reference to Carey et al., 2014 in line 48.

2. The authors have made special note of the hazards from this eruption, with good reason. The impact to the local island communities through volcanogenic tsunami and failure of the subsea telecom cables are very important. However, I thought that the authors implied that these risks are common to all submarine volcanoes and that by juxtaposing those statements next to the fact that there are so many submarine volcanic features, that they implied that the risks are exponentially greater than reality. The hazards from this eruption are very real, but are predicated on it being a very large, shallow-water eruption in a populated area. Eruptions of this nature can and will happen again and being prepared for that is important. However, most submarine eruptions are too deep and too remote to pose similar hazards. The authors cited the documentation of 100,000 of seamounts as increasing this risk, but the vast majority of those are extinct volcanos more than 100 Ma. I would suggest caveating the discussion of submarine volcanic hazards by pointing out what, when, and where those hazards are most prominent (e.g., Lines 47-49).

We have added to lines 50-54 statements to clarify these points, as well as additional citations this now reads:

"This observational bias is severe in regions under-represented by scientific study, such as the South Pacific Ocean where the majority of the world's 100,000 uncharted underwater volcanoes of >1 km in elevation lie. In particular, shallow water active volcanic centres that are near populated islands expose a major blind spot regarding risk assessment and response preparedness³⁻⁵."

In contrast to my statement above, the risks to benthic ecosystems are likely valid wherever submarine volcanic eruptions occur and have been documented in many places. It would be worth citing some of that literature:

- Vroom, Peter S., and Brian J. Zgliczynski. "Effects of volcanic ash deposits on four functional groups of a coral reef." *Coral Reefs* 30.4 (2011): 1025-1032.
- Angulo-Preckler, Carlos, et al. "Volcanism and rapid sedimentation affect the benthic communities of Deception Island, Antarctica." *Continental Shelf Research* 220 (2021): 104404.
- del Moral, Roger. "The importance of long-term studies of ecosystem reassembly after the eruption of the Kasatochi Island Volcano." *Arctic, Antarctic, and Alpine Research* 42.3 (2010): 335-341.
- Shank, Timothy M., et al. "Temporal and spatial patterns of biological community development at nascent deep-sea hydrothermal vents (9 50' N, East Pacific Rise)." *Deep Sea Research Part II: Topical Studies in Oceanography* 45.1-3 (1998): 465-515.
- Marcus, Jean, Verena Tunnicliffe, and David A. Butterfield. "Post-eruption succession of macrofaunal communities at diffuse flow hydrothermal vents on Axial Volcano, Juan de Fuca Ridge, Northeast Pacific." *Deep Sea Research Part II: Topical Studies in Oceanography* 56.19-20 (2009): 1586-1598.

We added the impact to benthic ecosystems to line 27. We have added these references as well as more detail and a few other relevant references to the section between lines 194 – 271 and throughout the text as appropriate.

3. I'm somewhat concerned about the description of the origin of the PDCs. The authors did a very

good job of explaining their use of the terminology and that these submarine mass flows are different from terrestrial PDCs given that they are water supported, but they make the claim that they likely originated by collapse of the eruption column and I don't think they've supported that claim (e.g., Line 114, 118-119). The alternative, that they originated from the submarine environment either via primary magmatic explosivity or hydrovolcanic explosions or simply mass wasting of the edifice, is not explored. If there is evidence that these originated from column collapse, the authors should include it, or at least note the alternatives.

We have adjusted the language to reflect the alternative mechanisms, now lines 139-142, and added supporting evidence from a recent publication (Purkis et al., 2023)

“The PDCs most likely initiated from the partial collapse of the eruptive column of Hunga Volcano, with this hypothesis similarly supported by the timing and intensity of the tsunami waves³⁹. However primary magmatic explosivity, hydrovolcanic explosions, or mass wasting of the edifice could have also been formative mechanisms.”

We would like to note that sensitivity tests of our PDC model (see supplementary material, Fig. S6 and S7) confirm that whatever initialisation mechanism caused these PDCs it must have occurred within the caldera in order to cause the strong channelisation that was observed on the flanks of Hunga volcano. An initialisation mechanism such as a fountaining PDC where the PDC and water mixed outside of the caldera (i.e. on the flanks of the Hunga edifice or even further out) would not have caused this channelisation. Additionally, this possibility has been suggested in relation to data regarding the tsunami waves in a recent publication which we have now cited (indicated above, Purkis et al., 2023).

On the basis of the eroded channel / flanks that we see on the flanks – footprint of the cladera; point to shanes paper and the video

4. No Tongan authors? I'm surprised that there are no Tongan authors on the manuscript. Surely they provided data or observations or significant science support for the work. I would think they should be on the author list.

We agree with R2 on this point, and have since included Mr. Taaniela Kula, the Deputy Secretary of the Tonga Geological Services, as a co-author. Mr. Kula has remained a key part of planning, coordination, and implementation of the field research programmes that has led to this work and he should have been included as a co-author from the beginning.

5. More about the cores: The cores collected seem a primary piece of evidence for this study, but the data presented seems limited. There is a single core that is imaged and described in the main paper, but there were more cores taken. For the single core described, the image of the core needs to be much better. It is impossible to see the structure and a portion of the community would probably like to see it and interpret the potential processes it indicates for themselves. I would suggest that the authors add a visualization of ash fall and PDC thickness to the map of stations where total thickness is indicated by dot size and fraction of fall, flow in a pie chart at each site. This information should be used to help support the arguments in the paper of erosion and deposition inferred from the bathymetric differencing and modeling. Also, it would be useful to know whether any cores were taken that did not show evidence for deposition (in the refugia, for example).

We have added a figure in the supplemental information which details locations of sediment cores, detailed visual sedimentological core logs and core photography along a transect that extends to the south-west from the caldera, with distance away from caldera indicated. This supplemental figure shows that even 80.5 km away from the caldera, we were still sampling density flow deposits 23 cm thick. Closer to the caldera we were unable to retrieve long enough

cores to constrain the extent of the flow deposit (the base was not able to be sampled). In the areas of refugia, we were unable to successfully retrieve cores due to the steepness of the seafloor and the rocky terrain. A follow-up paper (in review now) contains a more extensive record of sediment cores collected during the survey, and specific focus on the behaviour and evolution of the pyroclastic density currents. This level of detail is beyond this scope of this manuscript which is intended to be a multidisciplinary overview of the impacts of the event, rather than specifically focusing on specific sedimentological aspects. Additionally, we have added an improved image of the core to Figure 3.

Some more specific comments are detailed below...

Line Comment

23 It might be more appropriate to say more than 3/4 of all volcanic eruptions occur in the submarine environment, or more precisely more than 3/4 of all volcanic output. It would be good to cite Crisp et al., 1984 and maybe White et al., 2006 for this statement.

We have addressed this in the manuscript, lines 25-276 now reads: “..three quarters of all volcanic output worldwide...”

28 I would suggest: ‘what some have suggested is the largest seaborne volcanic eruption’. I don’t know that the erupted volumes have been accurately determined and I’m not sure if I would count edifice loss in the magmatic output that would make it a VEI 5.

Line 30 now reads: “...what some have suggested is the largest seaborne volcanic eruption in nearly...”

Similarly,

Lines 56-60 now reads: “With a Volcanic Explosivity Index of 5.7, the 15 January eruption was the largest eruption since Mount Pinatubo in 1991 and is one of the most explosive submarine eruptions ever recorded and largest documented in a century or more.”

30-31 Where does the information on the length of damaged telecom cable come from. I spoke with the crew who was fixing it and I did not get the impression that they were laying 100km of cable.

Tonga Cable Ltd (the cable owner) has recently commented that “The actual break distance between the 2 recovered ends of the international cable was actually 89km rather than 35km”. The domestic cable was damaged along a stretch of 105 km and has not yet been repaired as of the writing of this response. We have updated the numbers in the manuscript accordingly (lines 179-183: *The model is consistent with the locations and extensive damage to the international and domestic telecommunication cables, damaged along 89 km and 105 km length, respectively (Fig. 3), and explains the observed spatial trends in erosion and flow runout.*)

46 Again, suggest citing Crisp et al. and White et al. The cited papers aren’t really the go-to literature for documenting global submarine volcanic output.

We have addressed this in the manuscript.

46-49 I would suggest removing the statement about risk in the Pacific Ocean. The vast majority of the 100,000 seamounts are extinct and are so remote as to not pose any real hazards even where they active.

We have modified the statement to this (lines 50-54):

“This observational bias is severe in regions under-represented by scientific study, such as the south Pacific Ocean where the majority of the world’s 100,000 uncharted underwater volcanoes of >1 km in elevation. In particular, the shallow water active volcanic centres that are near populated islands expose a major blind spot regarding risk assessment and response preparedness.”

61-68 There is some missing literature here for studies that look at pre- and post-eruption seafloor conditions (notwithstanding that this study does not really have a comprehensive pre-eruption survey of benthic biology). I would include Carey et al, 2014 & 2018; Embley, 2014; Soule et al., 2009; Chadwick et al., 2008; Watts et al., 2015. It’s up to you if you want to say that it hasn’t been done on an eruption the scale of HT, but it doesn’t seem necessary.

We have added more references including these throughout the document. We were constrained by reference limitations in original submission but have asked for additional references to be allowed.

78 Should cite Garvin et al., 2018 that documented some of the bathymetric change at the early stages of this eruptive episode.

We have added this reference, now line 88.

114, 118 As mentioned, the authors should explain why they think the ‘PDCs’ originated subaerially.

Eye witness imagery has shown collapses from within the plume below the umbrella cloud shortly after the first major eruption on 15th January, which generated subaerial pyroclastic density currents; hence, we assume this is the most likely origin mechanism. However, we recognise this is not definitive, and as discussed above, we have now revised the language to be more speculative in the initiation of the flows (i.e.: lines 139-148 *“The PDCs most likely initiated from the partial collapse of the eruptive column of Hunga Volcano, with this hypothesis similarly supported by the timing and intensity of the tsunami waves³⁹. However primary magmatic explosivity, hydrovolcanic explosions, or mass wasting of the edifice could have also been formative mechanisms. Sensitivity tests of our PDC model (see Supplementary Material, Fig. S6), however, confirm that whatever initialisation mechanism caused these PDCs it most likely occurred within the caldera in order to cause the strong channelisation that was observed on the flanks of Hunga volcano. An initialisation mechanism such as a fountaining PDC where the PDC and water mixed outside of the caldera (i.e. on the flanks of the Hunga edifice or even further out) would not have caused this channelisation.”*)

120 It’s unclear what ‘observation’ is being supported by observations in this statement. Is it the origin of the PDCs from column collapse? It might be better to just say ‘We observe’.

We have addressed this in the manuscript, now line 149-154: *“We observe crescentic scours (up to 30 m deep) and bedforms (up to 1,200 m wavelength, 20 m wave height; Fig. 2), commonly seen in other deep-sea settings affected by powerful density flows⁴⁰, (including those triggered by large magnitude earthquakes, that severed seafloor cables, comprised a dense basal layer, and have been shown to attain speeds of up to 20 m/s)⁴¹⁻⁴³, and further corroborated by seafloor sediment coring and imagery, geochemical analysis and numerical modelling.”*

135 The statement about the volume needed to recreate the observed PDC distribution should be clarified. The model itself is not described, which is fine in the context that it is used - to define

flow pathways. Because of this, I'm not sure the volume estimate is particularly relevant. However, if this would be considered a minimum estimate because of the model assumptions, the authors could indicate that and leave it in.

We have added clarification on this point to lines 173-182: *“Numerical modelling of the submerged PDC (see Methods section for further details) suggest that approximately 3-4 km³ of sediment laden flow is required originating from inside the caldera rim to replicate the extent of the observed PDC deposits, especially the erosion channels on the flanks of Hunga edifice and the overtopping of two submarine knolls to the south heading towards the international cable (Fig. 3). While our relatively simple model does not account for seafloor erosion, incorporation of additional material, nor multi-phase flow, the flow pathways that the simulated PDC follows align remarkably well with the areas of most pronounced seafloor change. The model is consistent with the locations and extensive damage to the international and domestic telecommunication cables, damaged along 89 km and 105 km length, respectively (Fig. 3), and explains the observed spatial trends in erosion and flow runout.”*

In this we have also directed readers to the Methods section for further details. Importantly, in our model simulations, smaller volumes of sediment laden flow did not recreate what was observed following the eruption, including the fact that the PDCs appear to have overtopped two submarine cols in the south before reaching the international cable. We have added other model simulations to the supplemental to allow the reader to see these differences.

146 The mention of the Lau Basin here is a bit tricky. The authors have been in contact with a group that visited the Lau Basin and found evidence for HT ash deposition there, but nothing in the work described in the paper would necessarily predict that. To avoid undercutting the observations from the other group, it would be appropriate to simply say that the flows likely went well beyond the study area and not include mention of the Lau Basin.

The flow clearly extended towards the northwest towards in the direction of the Lau Basin which lies downslope, but we have no data to indicate how much further it went. As a result, we have removed mention of the Lau Basin from the document as suggested.

150 Correct wording. Should be the discrete observation of a mussel bed or discrete observations of mussel beds.

Corrected wording, now lines 217-222: *Pre-eruption deep-sea benthic surveys for the Hunga Volcano edifice and adjacent seamounts were not available to compare with our new post-eruption data, except for discrete observation of mussel beds on a southern seamount within our study area³³. The mussel beds visualised in 2007 and 2017 on the southern seamount were still present, and of similar density to prior observations³³, indicating that these communities were not affected by the eruption of Hunga Volcano (Fig. S11).*

182-184 The statement about suspended sediment and hydrothermal discharge needs some more explanation. First, there should be a reference to the supplement otherwise it is unclear what is being referred to in terms of suspended sediment. I'm also unsure why it is expected that suspended sediment will be a long-lasting effect. It should clear up in a matter of weeks to months unless something else is happening. In addition, I'm not sure why hydrothermal discharge is viewed to have a primarily toxic effect? In most cases it can have a positive impact on ecosystems around the hydrothermal vents and a pretty limited impact on suppressing ecosystems locally and certainly not regionally.

This sentence has now been removed.

Figures

Figure 2 This is a great figure that gives a very nice view of how density currents progressed and evolved. There is no description of the methods used to make the difference map or uncertainty associated with the estimates. There is clearly some error associated with rough terrain on features that are known not have changed, such as flank domes, that can arise from minor navigation offsets. It looks like that difference may not be stochastic (i.e., distributed around zero), but has a somewhat positive bias. It would be worth taking a look at the net gain/loss in areas that are known to not have changed to make sure the gain/loss is not biased. I'm also a little dubious of the spur to the NW, where pre-eruption data is sparse. There seems to be patterns in the gain/loss that look more like artifacts than real signal. I would be careful interpreting that and it may be better to leave it out of the analysis.

Rev 2 is correct that the color range we previously used was heavier on the gain, and so we have adjusted the color bar to be evenly distributed about zero. Additionally, the neutral color that was used in the previous figure to show no change was a warmer color that appeared to be more towards the orange side and skewed the image. We have corrected this. If you view the histogram included below you can see that the gain/loss appears roughly evenly distributed and does not appear to be biased.

Additionally, in the manuscript, we speak mostly of qualitative morphological change, not quantitative change. The only quantification that we provide is on the order of cubic kilometers. In this case, the error associated with the echosounders, even in the most erroneous scenarios, would be 1% of that quantification. The maximum error in the deepest portion of our survey area (associated with echosounder error) would be 2m vertically, and in the shallowest portions of our

survey area it would be 0.1m. This error would not significantly change the estimates we have provided on the order of cubic kilometers and we consider the specific uncertainties associated with the difference map estimates (which would require a depth variable uncertainty grid) to be beyond the scope of this manuscript, and not significant enough to impact the cubic kilometer volumes that we report in this manuscript.

Figure 3 The modeled PDC paths in Figure 3 should be made less prominent. They are best guesses at flow paths and are rather complex and distracting in the figure. It would be nice to see the presence/absence and abundance of benthic life against the loss-gain map, or a least the average for that area could be shown next to the point. It seems that there are some points with benthic biota present near the caldera (to the east) that seem contradictory to the idea of complete destruction of habitat. Also, please use a higher resolution image of the core. It is impossible to see.

The modelled PDC paths have been made less prominent in Figure 3. We think it is best to keep the benthic biota observations in Fig 3 and not include with the loss/gain map, however we have included a simplified abundance table as a supplemental table to show what groups of invertebrates were present in each location, and the shift in diversity of invertebrate groups, e.g., more diversity in areas of refuge. Additionally, this table shows orders of magnitude difference in the abundance of invertebrates near to the caldera and in areas of refugia. Further, these points were there are some benthic biota still present are within areas that have minimal seafloor change noted in Figure 2 (and appear outside of PDC paths).

We now include a higher resolution core image as well in Fig. 3 and in the Supplemental core log.

Figure S3 It is hard to see the symbols on the geochemical plots. A higher resolution version should be included. Also it seems that the T2206 data is just plunked on an older figure and covers up some of the previous data. Also, the legend is missing a symbol for T2206 shards.

We have made a higher resolution version of Figure S4 (updated numbering) and corrected issues with data plotting which was resulting from the original figure being corrupted during conversion to a PDF.

Reviewer #3 (Remarks to the Author):

This manuscript compiles observations from ship-based cruises to describe the seafloor impacts of the January 15, 2022 Hunga Tonga-Hunga Ha'apai eruption. In addition, they describe results from a PDC model that they plot against observations. Observations include changes in seafloor bathymetry, changes to biologic communities, and reports on the characteristics of the sedimentary deposits. Due to the remarkable nature of the 15 January Hunga Volcano eruption, I suggest that these results are worthy of publication in a high impact journal such as Nature Geoscience. The changes in seafloor bathymetry, existence of pyroclastic density current deposits, and runout distances of these pyroclastic current deposits will be of widespread interest to the volcanological and oceanographic communities, in my opinion. That said, some of the interpretations are not currently well-supported by information provided in the ms although I do not doubt that this data was collected. I also have significant questions about the PDC model. After these and other changes are addressed, I would highly recommend publication as I believe these results are of particularly high impact and importance. Please see my attached comments.

-Kristen Fauria

We thank Reviewer 3 for their time spent in reviewing our manuscript and offering constructive feedback and encouragement. We have addressed concerns and made changes as indicated below.

General Comments (from attached document)

First, I want to congratulate the authors on their incredible work surveying the Hunga Volcano post-eruption. Their findings on PDC runout and the lowering of the caldera floor are remarkable. My comments and suggestions below are primarily aimed at clarifying my own questions and making sure their observations can be well understood and utilized by readers and scientists studying the Hunga eruption.

We thank the reviewer for their positive feedback.

The change in bathymetry is a major finding of this manuscript. Because of its importance I would suggest clarifying both the bathymetric observations in the text and improving the quality of the difference map in Figure 2. Specifically, there is no absolute scale on the colors used in the difference map. I cannot see a difference between the remarkable >800 m loss in the caldera and more modest loss along the flanks.

We have added an absolute scale to the colours used in the difference map, this is +/- 30m as you will now see in Figure 2. We are intentionally trying to not focus on the >800m loss in the caldera, instead focusing instead on the loss on the flanks. The loss within the caldera is a focus of other studies by our colleagues. We have included the cross-section showing the loss within the caldera in Figure 1 to allow the reader to appreciate the massive loss within the caldera, which compares to the more modest changes on the flanks. We have also made additional changes to Figure 2 following recommendations of Reviewer 2 which should make some of the loss/gain that we have measured more obvious.

I also cannot tell, from the text or figure, how much variation there is in deposition or scour along the flanks.

The changes we have made to Figure 2 should make the variation in deposition/scour along the flanks more obvious.

The ms could also benefit from a brief description of the pre-eruption Hunga Volcano morphology (i.e., that it is a caldera volcano) before a discussion of the topographic changes is introduced.

We have added the following sentences / reworded text in lines 87-96 with this in mind:

“The islands of Hunga Tonga and Hunga Ha’apai are remnants of the rim of a roughly 3- to 4-km large submarine caldera of Hunga Volcano, with eruptions noted in various parts of the caldera throughout the past century. The most recent eruption series (in 2015) formed a 120m high and 2km wide near-circular tephra cone which joined the two islands together^{6,7,26}. After this eruption the Hunga Volcano was largely inactive until it began erupting in late 2021, producing a steam-rich gas and ash plume. Following this, Hunga Volcano continued erupting intermittently until two large eruptions 14-15 January 2022, with the largest on the evening of 15 January³⁴.”

The authors claim that the areas of deposition on the volcano’s flanks are due to PDC deposition. Although this is reasonable, the claim is not well-supported by sedimentological observations in the text or supplement. More information along the lines of the stratigraphic section in Fig. 3 would be

helpful For example, the authors discount sedimentation by landslides without explaining why or defining what they consider to be a “lack of evidence.” The deposits are interpreted to be PDCs in part due to bedforms and sediment cores, but the features within the cores or bedforms that distinguish the deposit as PDC are not mentioned or defined. Although thorough work on the sedimentary structures within their cores and from video observations may be outside the scope of the paper, I suggest that the authors describe 2 what specific aspects of the cores and seafloor observations allow them to define the deposits as PDC related (rather than mass flow by other flank collapse, for example). In the end, I agree that these are PDC-type flows but based on their spatial distribution rather than the sedimentological evidence provided.

We have modified the sentence in lines 126-129 to be more inclusive of the possibility of land sliding:

“There is no superficial morphology consistent with seafloor landslides (i.e., headscarps and/or scars) on the volcano flanks despite the scale of the eruption and associated seismicity, which makes this event distinct from partially-submerged eruptions that generated large slope collapses such as Anak Krakatau (Indonesia) in 2018³⁷”

As we state in lines 132-134, it is the spatially variable change that we are relating to the PDCs first and foremost, as well as the crescentic scours that are indicative of a large sediment-laden flow (lines 149-150). The sediment coring and other analysis corroborate this, but detailed analysis of the sediment cores collected as part of this broader work is beyond the scope of the manuscript and will be included in a follow-up manuscript that is in review.

I found the authors findings on post-eruption venting compelling. That said, the descriptions of phenomena such as post-eruption venting are only vaguely described. The ms would benefit for more clear (even if in the supplement) descriptions of where venting was and was not observed and how.

We have adjusted the text to reduce the focus on potential venting. Instead, in lines 251-252: (*Future research will seek to determine if this water column turbidity is related to ongoing venting from plumes within the caldera*) and in the figure caption for Supplemental Fig. S12 we specifically say that the particle layer in the water column, and the ‘plume’ image indicated in Fig. S12, are potentially a result of ongoing venting within the caldera, with the nature of this to be determined by future research. We are specifically trying to not describe these phenomena as there is ongoing work to comprehensively characterise plumes detected within the caldera that we are trying to not supersede.

I am pleased that the authors employed a PDC model and compare the model to observations. I have major questions about the model, which I detail at the end of this review. One main concern is that it is not clear however, what aspect of the PDC model they are comparing to what characteristic of the observations. There is some discussion of runout. However, how is PDC runout characterized in the model and in the observations?

Minimum runout is best constrained by the coring and cable damage. The model (as described in the Methods Section) is a fairly simple two-layer model where the turbidity current from the PDC is modelled as a dense Newtonian fluid underneath the water layer. As such it will not correctly predict the maximum extent of the runout but will over-estimate this. The model is, however, able to show that a minimum volume of PDC is required to over-top two saddles to the south of Hunga to reach the international cable. We have clarified this in the text.

How do the authors determine the flow direction (arrows on Figs. 2 and 3) from the model? The authors acknowledge that the model does not include deposition. How, then, is the ultimate flow distance in the models determined (what process stops the PDC in the model)? I agree with the authors approach of employing a general model, but it is not at all clear how their model works.

The flow directions in the figures are merely indicative. They were added to the figures from animations of the flows. The supplemental material contains snapshots of these flows at various times (Figure S6,S7) as well as an animation of the flow in the electronic supplementary material (Supplemental Video 1). As mentioned above, the model provides no information as to the ultimate flow distance as the PDC is modelled as a Newtonian fluid and therefore only stops when it reaches a depression. The model is, however, able to distinguish between the volume of PDC required to overtop the saddles to the south of Hunga and reach the international cable, which we see from the sea-floor observations.

Details of the equations used in the model are now given in the Methods Section.

Line by line comments

Line 63: I suggest being specific about the types of data used earlier in the ms and on this line.

We have addressed this in the manuscript, the sentence now reads (now lines 73-76):

“Combining these datasets (including multibeam echosounder and geological data, video footage and water column samples), reveals the wide-reaching ocean impacts of such a large eruption, ranging from topographic changes and damage to critical infrastructure, to widespread loss of seafloor life.”

Line 68: Add “mass” to the description of “loss.”

We have addressed this in the manuscript (line 75-76: *widespread loss of seafloor life.*).

Line 79: The eruption on Jan 13-14 could be described as a single large event rather than a series of events before the 15th (e.g., Gupta et al., 2022).

We have clarified this in the manuscript (lines 93-100: *“After this eruption, Hunga Volcano was largely inactive until it began erupting in late 2021, producing a steam-rich gas and ash plume. Following this, Hunga Volcano continued erupting intermittently until two large eruptions on 13-15 January 2022, with the largest on the evening of 15 January³⁴. Satellite imagery on 15 January, prior to the largest eruptive episode in the evening, showed the islands had been disconnected by the eruption on 13-14 January³⁴ (Fig. S5). After the 15 January eruption, only small remnants of the islands of Hunga Tonga and Hunga Ha'apai remained above water. However, the modification of the submerged 99% of the volcano edifice remained unknown.”*

Line 87-91: Here the Hunga summit is mentioned for the first time and it would benefit with a brief summary of context. I suggest briefly describing the pre-eruption bathymetry to give the readers context for where bathymetric change occurs. One of the most notable findings is the >800 m change in the caldera. It is mentioned here, but the reader would not know Hunga is a caldera volcano, without prior knowledge, and would not know the pre-eruption caldera depth. I suggest adding those things here. Could you also comment on whether this change in caldera floor elevation is uniform across the caldera?

We have added lines at the beginning of this paragraph (starting at line 82) describing that Hunga Volcano is a caldera volcano that has had various eruptive phases, that most recent of which

formed a 120m high and 2km wide near-circular tephra cone which joined the two islands together. It is this difference between the tephra cone and the new caldera depth that we describe in the next paragraph and illustrate in Figure 1. The depth of the remainder of the volcano that was submerged is shown in Figure 1, with contour lines added to aid in the visual analysis of differences in depth pre- and post eruption.

It also seems crater and caldera are used interchangeably. Is there a difference between the caldera and caldera? What kind of change occurred along the caldera rim?

We have changed all instances of crater to caldera. With regards to change along the caldera rim, change on the caldera rim or within the caldera is not the focus on this manuscript but will be extensively detailed in follow-on work. This manuscript is focused on change on the flanks of the volcano and the surrounding seafloor, and we want to keep this focus in the manuscript.

Lines 89-91: Could you expand upon the volume loss along the caldera rim. Was the mass loss uniform along the flanks or localized? If uniform, what was the average change in vertical elevation along the flanks? How far from the caldera rim did the mass loss extend? That is, where was the transition from erosional to depositional features and how uniform was that distance from the caldera rim?

We have adjusted the colour-scale of Figure 2 which makes the mass loss / gain along the rim of the caldera and along the flanks of the volcano more obvious. We have added a sentence at line 124 to indicate how far from the caldera rim mass loss extended:

“Most of the measured loss was within 6.5 km of the caldera rim, with this distance greatest in the NW direction, beyond 6.5 km there is predominantly a depositional environment.”

Line 93-94: Ashfall is mentioned here and subsequently. However, the ashfall portion of this eruption is not described in your ms. Could you briefly describe what is known about the presence of ashfall?

Now line 155-156 – we have updated to : “*..overlain by fine volcanic ash deposits tens of centimetres thick..*” as the ash is not necessarily from ash fall but could also be from submarine ash release and deposition.

Line 95-97: The supplementary data table is cited as evidence here. I am so glad you have included these supplementary data. However, what aspect of the data are you suggesting makes this point? Could you perhaps provide a plot of the data in the supplement to make this point?

A plot of the data is provided in the Supplementary Material (under Expanded Geochemical Results). This section shows plots of the data compared to previously analyses of material from Hunga volcano (Brenna et al., 2022) as well as other nearby volcanoes. The data match those from Brenna et al (2022) and are distinct from other volcanic material sourced from other nearby volcanoes, indicating a Hunga volcano source. A brief statement reflecting this has now been added to the main text (specifically lines 161-165: “*Scanning electron microprobe analysis and X-Ray fluorescence spectrometry indicate that PDC and ashfall deposits have the same chemical composition as samples previously analysed from Hunga volcano⁴⁴, suggesting that they were sourced from the Hunga Volcano eruption (Supplemental Information; Figs. S3, S4; Supplemental Data 1).*”)

Line 98: "There is consistent loss" is rather vague. What is consistent about the mass loss (spatially, mean change in elevation?). Why choose a water depth threshold? Does this correspond to a mean distance from the caldera rim? If so, what is that distance?

The word "consistent" has been changed to "widespread" (now line 116). This is more of an observation, above 600 m we see erosion and below we see deposition. We have also included reference to the distance (6.5km) from the caldera rim that we see a transition from predominantly mass loss to mass deposition (lines 124-126: "*Most of the measured loss was within 6.5 km of the caldera rim, with this distance greatest in the NW direction, beyond 6.5 km there is predominantly a depositional environment.*")

Line 102-103: I assume here you mean chute termini. Could you make more clear the transition in discussion from erosional to depositional features.

We have added the word "chute" here to clarify (now line 121), as well as distance as indicated above (6.5km). Additionally, the changes made in Figure 2 should make this transition more obvious.

Line 106: What do you consider an absence of evidence of landsliding? Could you briefly list the features you would look for to indicate landsliding? How much erosion and deposition occurs outside of the "chutes". Could you also describe the change in bathymetry in the areas not defined by "chutes".

We have added: "*There is no superficial morphology consistent with no seafloor landslides (i.e., headscarps and/or scars)*" to lines 126-127.

Lines 113-114: The process-based description of PDC formation here is a suggestion or interpretation rather than an observation. I suggest, therefore, caveating this sentence with a phrase such as "we suggest". I also wonder the extent to which the flows may have never entered the atmosphere and remained entirely submerged since the vent was submarine? This possibility could be acknowledged even if only for a portion of the flows.

We have modified this sentence (line 134-136) to read: "*We suggest these flows initiated as PDCs that mixed with the ambient seawater, becoming turbulent underwater density flows..*"

Additionally, in response to Reviewer 2 we have also adjusted the language elsewhere to reflect the alternative mechanisms, now lines 139-142.

"The PDCs most likely initiated from the partial collapse of the eruptive column of Hunga Volcano, with this hypothesis similarly supported by the timing and intensity of the tsunami waves³⁹. However primary magmatic explosivity, hydrovolcanic explosions, or mass wasting of the edifice could have also been formative mechanisms."

Line 118: Please describe the plume collapse as "partial" since the plume attained a height of 57 km.

We have modified this sentence accordingly (now lines 139-140: *The PDCs most likely initiated from the partial collapse of the eruptive column of Hunga Volcano..*).

Line 120-121: Could you please remind the reader how these observations were made. Overall, these scours are remarkable and could be discussed or emphasized more than they are at present (purely optional).

This is the focus of follow-on work so we do not analyse them in detail here.

Line 126: Could you please remind the reader the scale of a granule? Using what methods was grainsize evaluated and quantified?

Grain size was identified using a visual grain size comparator as outlined in the Methods section. We now specify “up to 4 mm-diameter scoria” rather than “granule-sized” for clarity (line 159).

Line 124-125: How far was this from the vent?

Have added distance (now line 154 : “Multicoring 80 km from the caldera”)

Line 128: Here an earlier description of ashfall could be helpful. What sample did you compare against to determine they were from Hunga Tonga?

The data was compared to that of Brenna et al (2022) who analysed volcanic material from the islands, the data match extremely well (see Supplementary Material – “Expanded Geochemical Results”. In addition, the data was compared to other nearby volcanoes and is distinct from them.

Overall, the finding that the PDCs go from erosional to depositional is very interesting. As mentioned earlier I would love more information on the dimensions (thicknesses, distance from the caldera) of the erosional and depositional components.

We have added the distance from the caldera of the erosional and depositional components as indicated above. The chute-specific changes will be described in more detail in follow-on manuscripts.

Could you also put the vent location in the context of these observations. It seems that you assume all material was sourced at the vent, which was on one side of the caldera, yet the erosional and depositional features are evenly distribution around the caldera? Could you comment, confirm, or clarify in the ms?

We are treating the caldera as the centre, with the flows radiating out. This intensity of this radiation does vary around the caldera rim, with some pre-existing chutes (e.g. the NW direction) having more focused erosion.

What were the maximum and characteristic runout distances of the PDC for both the channelized and unchannelized components?

We cannot state this with certainty as the model is not appropriate to define runout for areas where it is not bathymetrically constrained. We can infer from observations (i.e. cores and cable damage) where the PDCs definitely reached and runout distance were likely considerably further than these locations based on the depths of deposits measured and the amount of damage. From this we can estimate minimum volumes of the PDC required to reach this far (the international cable lies over two submarine saddles). However, our interpretation remains speculative (e.g., lines 190-192: The longest observed PDC runout likely exploited bathymetric lows to the NW of the caldera. Modelling and post-eruption bathymetry indicate this PDC went well-beyond the northwest limit of our survey area). *This will be investigated in more detail in a follow-up manuscript.*

Line 135-136: Please describe the basic attributes of your model here. Was the model for a dense or dilute flow? What were the basic assumptions or initial conditions? Did you model a PDC in water or in air, for example? Here you mention replicating the extent, are you referring to the depositional

extent? How did you do this comparison in practice, especially if the PDC model does not include deposition?

We describe the model in the Methods section. What we actually model is the underwater flow characteristics of the dense current created from the mixture of the dense PDC and the ambient water. The PDC and the water above it are both modelled as Newtonian fluids which is why we are not able to make predictions about the maximum extent of the PDC. When we refer to replicating the event we are referring to the fact that we are able to replicate the flow paths seen in the observations and our maximum flow depths compare well with locations where benthic fauna was destroyed. We also specifically refer to the fact that a minimum volume of PDC was required to overtop the two submarine knolls to the south of Hunga and reach the international cable.

Line 137: Here you introduce the concept of simulated flow pathways. How are these defined? I am unfamiliar with PDC models being simplified into the arrows shown on your figures. On what basis are you defining these pathways?

These flow path lines are simply an attempt to illustrate complex time varying flows over 2D topography in a single diagram. Snapshots of the modelling are also shown in the supplementary material as well as an animation being provided (Supplemental Video 1). These show where the flow fronts and deepest material flowed in the simulation.

Line 140: What is meant by broken cables along >30 km and >100 km lengths? Is this distance from a location? If the cables were not broken in one location, could you please describe?

We have modified the sentence for clarity (now lines 180-182: “*extensive damage to the international and domestic telecommunication cables, damaged along 89 km and 105 km length, respectively*”). These lengths are the points between which the cables were damaged. As mentioned earlier, we have since received further clarification from the cable owner that the cables were actually broken along 89 km (international) and 105 km (domestic cable) length.

Line 144: I do not understand this comparison. Are you comparing the model results to observations from areas other than Hunga?

We have clarified this in text by saying: “*As the submarine PDC flows away from the caldera, it is steered by the local bathymetry and in some locations separate PDC flows converge back together and accelerate. This has previously been observed subaerially but only inferred in submarine settings*” (now lines 184-190).

Line 144: What was the longest PDC runout?

As we indicate above, the model does not allow us to accurately provide this information, nor do we have sufficient observational data to estimate the longest PDC runout (as this likely went beyond our survey area as we indicate).

Line 163: What aspect of the PDC model is being compared to locations of the refugia?

We have modified this to read: *The locations of these refugia match closely the thickest flow paths of the PDC modelling, which shows that deflection of flow pathways around irregular relief and seamounts, leaving some areas relatively unaffected (Fig 3, S6; Supplemental Video 1).*” (now lines 215-217).

Line 185-186: How was turbidity measured and where and when? Was there excessive turbidity everywhere or in specific locations (both spatially and in water depth)? Could you expand upon the definition of nepheloid layers as this may be a new concept to many volcano readers?

Turbidity was measured with a transmissometer on a CTD rosette. This had been added to the figure caption of Fig S12 and the new Fig S13. The locations of nepheloid layers are now shown in the new Fig S13, with the spatial and water depth variation shown. We have expanded upon the definition of nepheloid layers in the main text as well as added clarity to this section (lines 251-257: *Future research will seek to determine if this water column turbidity is related to ongoing venting from plumes within the caldera Near-seabed ash layers (e.g., nepheloid layers, a layer of water above the ocean floor with a significant amount of suspended sediment) were detected above the seafloor across many of our sites, in some cases extending hundreds of metres up from the seafloor (Fig. S13). In all instances (mid-water and near-seabed ash layers), the turbidity anomalies were identified as being ash particles with SEM imaging and analysis (Figs. S14-17, Supplemental Table 1).*)

Could you make Fig. S10 easier to interpret to substantiate this point. For example, the acoustic image is not annotated and has no scale, date/time, or specific location (lat/lon). The ongoing venting is of particular interest and currently it is only vaguely described. Where and when was venting found? Can you determine if/when the venting shut off?

We have added additional text, and SEM images to support the beam transmission anomalies shown in Fig S12 and reported in the manuscript.

In the conclusions, I suggest providing a brief summary of your bathymetry change and PDC erosion/deposition observations.

We have added this to the conclusions, lines 263-271: *Our data illustrate how an eruption of a single submerged volcano can have wide-reaching and diverse impacts. PDCs resulting from the Hunga Volcano eruption were steered by antecedent morphology, further excavating pre-existing chutes and depositing the majority of the erupted volume on the seabed within 20 km of the Hunga caldera rim. Modelling and benthic observations indicate that the majority of the PDCs originated within the caldera and flowed around bathymetric relief as they moved to deeper waters, with only local preservation of benthic communities on topographic refugia. Outside of these topographic refugia the seafloor ecosystems around Hunga Volcano were decimated by the eruption and related PDCs.*

I am not a petrologist and so cannot evaluate the details of their methods for determining the ash composition. It could be more clear what sample they compare against to determine the true Hunga composition.

This is provided in the Supplemental Material 'Expanded Geochemical Results' which shows the data compared to other studies

Fig. 1: Can you please make clear if the two A to A' transects and B to B' transects are the same or different. If location A is location B and location A' is location B', I suggest using the same label for both.

We have made this change.

Fig. 2: How do you choose how many flow trajectories to plot here from the model, why not 3 or 300? Please provide an absolute scale for mass gain/loss. What is meant by “diffusive flow”? I also don’t see any evidence for interaction with topography as annotated on the figure.

Per recommendation of Reviewer 1 we have removed the text entirely that was describing the flow types. An absolute scale for mass/gain loss has been added as well. The flow trajectories are simply indicative flow paths interpreted from model animations and snapshots (see Figure S6 and Supplemental Video 1) and they represent the main channels of flow seen in these. The number was basically chosen as a minimum thought to be able to represent the time-varying movement of the flow on a static map. We have clarified this in the caption. Lines 491-494: “*Indicative PDC flow paths (i.e., the passage of the flow fronts as estimated from model animations and snapshots (see Supplemental Video 1 and Fig. S6)) are indicated with dashed grey lines plotted over the difference map with arrows indicating flow direction.*”

Fig. 3: Is this sedimentary log based on a single point or a general summary? To what extent is this log representative? Overall, this is a nice figure. I have the same issue with the flow pathways. I would like to see scale bars on A-C and the other panels labelled with letters as well.

The sedimentary log is based on a single point and is representative of the general findings. The differences in deposits measured at each core site is indicated in Table S1, to compare with the detailed core log in Fig 3 and Fig S1. We have added scale bars to A-C, but prefer to keep the other panels without letter labels. We have also added a description about how the flow pathways shown were drawn (Lines 504-508 “*Indicative PDC flow paths taken from modelling results (see Supplemental Video 1 and Fig. S6) are shown in solid black lines and locations of damaged submarine cable are depicted with dashed pink lines. In order to reach the international cable, the PDC must overtop two knolls directly E and ESE of location C, which partially constrains the minimum volume of material in the PDC.*”)

Line 406-407: How much footage (distance or time)?

We have added this (now lines 533-536: *The abundance of invertebrate taxa were counted in real time from the live video feed (totalling 31 hours over a distance of 32km) and hence the data included here should only be regarded as provisional estimates of abundance.*

Line 505: Why 50 cm gridding of the data. This seems very high resolution for ship-based bathymetry. Wouldn’t the natural resolution be much lower.

The gridding was to a 50 m resolution, not 50 cm. This is the standard gridding that we use.

Please add scales to all images. Could you please make sure there are geographic coordinates included in all sampling locations (e.g., Fig. S1 and data tables).

We have added scales to all images. Additionally, we have added geographic coordinates to Table S1 (which also covers Fig S1) and the Supplemental data table. We also have locations for the invertebrate observations in the new Supplemental Table 2.

PDC Modeling Comments: PDC modeling is challenging and I think it is neat that the authors undertook this challenge in addition to their substantial ship-based work. Overall, I cannot understand the model based on the information provided. Please see the questions below. - Did you consider the current dilute or dense in your modeling effort? Your choice would result in different conservation equations and assumptions. - I am unfamiliar with this modeling software for PDCs. Can you describe how common it is for PDCs compared to other types of models such as VolcFlow, Titan2D, and other known codes? Why did you choose this modeling software? - What are the governing equations for mass, momentum, and energy in your model? - The two-layer choice is interesting, but I don't understand how it is set up. Why two layers with a water layer on top? The water layer is the ambient fluid so shouldn't be considered part of the flow. How does this work? - How was flow through water dealt with since most PDC models consider flow through air. That is, how did you treat the ambient fluid (as water or air)? Was the model 1D, 2D, or 3D? - Is the model single-phase? - What controls runout distance in the model. How does the model stop? This is particularly important since runout is one of the features you compare. - How is the model coupled to topography? - Was the initial volume of PDC material placed on the seafloor, at the water surface? - How did you parameterize key aspects of the model (e.g., friction coefficient, Froude number, Richardson number, or entrainment coefficient)? I don't understand enough about your model to know what you needed to parameterize, but models that aren't 3D include these types of parameterizations. - Can you clarify your boundary conditions.

We have expanded the explanation of the model in the Methods section. As we explain in lines 132-139 (*We relate these spatially variable changes in seafloor elevation on the volcano flanks to particle-laden gravity flows that radiated downslope from Hunga Volcano in multiple directions, exploiting and enhancing pre-existing topography (Figs. 2A, S6; Supplemental Video 1). We suggest these flows initiated as PDCs that mixed with the ambient seawater, becoming turbulent underwater density flows (e.g., turbidity flows) that were steered by the complex topographic relief around the volcano. For simplicity, and due to their formative mechanism, here we also refer to the underwater component of these flows as PDCs, although we recognise that their behaviour changes as they travel underwater³⁸.*), what we are actually modelling is not a classic PDC but a turbidity current generated by the mixing of the dense portion of the PDC with the sea water. The model is a two-layer model which, by its nature, must have the more dense layer below the less dense layer. The simplicity of the model means that we must initialise it at the point where the PDC has already mixed with the water to create the turbidity current.

REVIEWER COMMENTS

Reviewer #2 (Remarks to the Author):

I'm speaking on behalf of reviewers 1 & 2, since reviewer 1 was unable to re-review.

This paper describes the initial submarine observations of the 2022 Hunga Tonga eruption and will be an important touchstone for ongoing and future studies of this and other similar shallow water explosive eruptions. The reviewers had a lot of comments seeking clarification, addition, justification, and the authors did a great job of addressing those comments, sometimes directly and sometimes by addition or subtraction of information. The comments were wide-ranging and, I presume, not an easy task to address them all. Kudos! Overall I think the paper is nearly ready for publication, and only have a few minor points that should be addressed, along with a few specific wording suggestions. The authors should consider these comments as burnishing of a well-written paper and can feel free to address or ignore as they see fit.

- I found the first few sentences of the introduction a bit confusing. Are the impacts of large explosive eruptions on land and in the marine environment being described, or just terrestrial eruptions? I was thrown off by the mention of tsunamis (which of course could refer to terrestrial eruptions - but are not a common impact) and the inclusion of reference 20. Perhaps I was influenced by the first two sentences of the abstract which more clearly stated 'we know a lot about large eruptions on land, but very little about those in the marine environment'. If this could be clarified just a bit to better reflect the statements in the abstract it would be much more powerful.
- line 51 - add comma 'shallow-water, active'
- line 83 - replace 'large' with 'wide'
- line 88 - replace 'until' with 'culminating with'
- line 90 - remove 'in the evening'
- line 91 - remove '99% of the volcano'. Given the comment by reviewer 1 (although addressed in the authors rebuttal) it doesn't seem necessary to put a number on this.
- line 96 - replace 'a decrease of >800m elevation' with 'an increase in depth >800m'. There is a range of terminology related to seafloor change in the paper with volume and seafloor being used interchangeably and depth and elevation being substituted. When referring to the seafloor change, I would stick with referring to changes in depth and addition or removal of volume or material.
- line 101 - 'post eruption' I think this is referring to features present in pre- and post-eruption data, but the wording is unclear.
- line 101 - replace 'following' with 'during'
- line 105 - replace 'modern echo sounder techniques' with 'ship-based echo sounders'. Modern echo sounders at high frequency operated near the seafloor could indeed resolve small changes.
- line 109 - 'elevation' to 'depth'
- line 110 - replace 'seafloor' with 'volume'
- line 111 - replace 'was focused within' to 'coincide with'
- line 114 - perhaps add a clause that the 'new chute' was not present in pre-eruption data.

- line 116-117 - 'distance greatest in the NW direction' I know what you're saying here, but I think most will find it confusing. Could leave out. Possible new sentence 'Most of the measured loss was within 6.5km of the caldera rim, beyond which deposition dominated.'
- line 130 - could also cite 'Cas, R. A., & Wright, J. V. (1991). Subaqueous pyroclastic flows and ignimbrites: an assessment. Bulletin of Volcanology, 53, 357-380' although I know you're running short on references.
- line 133-134 - remove 'or mass wasting of the edifice' as there is now a description of why this was unlikely.
- line 136 - rephrase to indicate channelization at the rim since that is the key piece of evidence supporting the hypothesis (e.g., channelization that was observed on the flanks of the volcano and extends up to the caldera rim).
- Line 138-139 - I don't think this last sentence is necessary given the argument made above. It's placement is a bit odd and reads like a response to reviewer comment.
- Line 140-144 - This very long sentence should be broken up.
- Line 148 - remove 'which we interpret to be a result of PDC's'. It doesn't seem that people need to be reminded that you're talking about the PDCs.
- Line 151-154 - I would reword the geochemistry description for clarity...'Major element chemical composition of PDC deposits was measured by XRF and overlap with those reported for recent eruptions from Hunga-Tonga Volcano (Brenna et al., 2022) and distinct from other regional volcanoes.' I think this implies the source of the eruption and you don't need to spell it out more explicitly. I also think you could leave out the SEM EDS analyses of the CTD-recovered ash shards here and move it to the section that talks about turbidity plumes/layers.
- Line 158 - 'that eventually decelerated' seems unnecessary. I see that you use it as a link to 'widespread deposition', but you could instead say 'that blanketed the flanks of the volcano and beyond'.
- Line 161 - I don't really see the point of comparing the volume to a subaerial landslide?
- Line 164 - Saying the model replicates the erosion channels is a bit misleading since the model, as I understand it, does not allow for erosion. I would remove this or rephrase.
- Line 169 - remove 'extensive' seems pretty binary to me damaged/undamaged. If you do remove it, you should also remove 'damaged' on line 170.
- Line 173 - replace 'submarine' with 'simulated'.
- Line 186 - replace 'occurred' with 'were observed'
- Line 188 - recommend breaking up this sentence, e.g., '...of our survey area, at distances >50km from the caldera. The density and diversity of these ecosystems aligns with expectations for the warm, oxygenated...
- Line 193 - what is meant by 'thickest flow paths'? Widest, deepest?
- Line 211 - confusing wording 'covering 700-750m'. Perhaps remove and put (700-750m) after Seamount.
- Line 228 - Need a period after caldera.
- Line 232 - clarify. 'The turbidity anomalies were sampled by CTD and the filtrate was dominated by ash particles as identified by SEM imagery and with a similar composition to the PDC deposits as measured by SEM EDS.'
- Line 248 - possibly replace 'substantial' with 'potentially substantial' or just remove 'substantial'.
- Line 250 - Perhaps add 'locally decimated benthic communities' to the suspended particles as impacts with potential enduring effects.

Reviewer #3 (Remarks to the Author):

Review for Pyroclastic density currents explain far-reaching and diverse seafloor impacts of the 2022 Hunga Tonga Hunga Ha'apai eruption

Thank you to the authors for their thoughtful comments and revisions on the previous ms. Most of my comments focus on the modeling aspects of the work and making the figures and data as useable for a general audience as possible. I have not provided many specific comments on the biological destruction aspects because these are outside my realm of expertise.

Although the new methods section has clarified some important aspects of the model, my opinion is that more clarity is needed in regard to both how the model works and towards how the authors are drawing specific conclusions from the modeling work (see further comments below). That is, I still do not understand many aspects of the model and specific model conclusions are not supported well by information provided in the ms, in my opinion. For example, I think Figs. Like S6 and S7 are valuable, but I do not see general differences between these two figures and that support the conclusion that flow initialization is needed inside the caldera to create channelized flows.

Overall, I think that this work is of extremely high scientific value to the community. The data sets included in this ms are large and I applaud the efforts to share them and make them broadly available with the scientific community.

Line by line comments:

Line 32: I suggest making it clear that you are inferring or providing evidence that transport and deposition was by submarine pyroclastic currents in this ms.

Lines 50-51: Would there be a reference for the claim of 100,000 uncharted submarine volcanoes?

Line 74: Add the word "submarine" before "loss/gain"

Line 76: Was the runout truly greater than 100s of kms or simply >100 km?

Line 87: Would there be a reference for the start of the eruption in late 2021 and composition?

Line 92-93: Perhaps modify to something like, "However, the modification of the submerged fraction of the volcanic edifice (~99% of Hunga volcano) remained unknown."

Line 100: Perhaps modify to, "An additional 3.5 km³ was lost from the outer flanks, without impacting Hunga's general morphology (Fig. 1). Many small features, such as peaks, ridges and gullies, were visible in the bathymetry post-eruption."

Line 105 – 107: I am unclear what this sentence is saying or suggesting, could you please clarify.

Line 108: Could you make clear that this 600 m threshold refers to the flanks of Hunga and the area around the caldera rim? Currently, the depth threshold doesn't make a lot of sense in terms of what spatial area it is referring to.

Line 111: What is meant by the term "entrenchment"? I am not familiar with the use of this term in this way.

Lines 113-114: Modify to "widespread blanketing of sediment"

Line 138-139: Is the rationale here that the channels begin close to or at the Caldera rim, and therefore the flows must have been fully formed by the time they reach the caldera rim?

Line 141: Can you point these scours out in Fig. 2 with an arrow?

Line 149-150: Are the ripples on the seafloor or within the deposit?

Line 153: Are you saying the submarine deposits have the same composition as ashfall from 2022 Hunga. Or, that both submarine and ashfall deposits have the same composition as previous Hunga eruptions.

Line 158: What was the evidence for local erosion? The negative difference in part of the bathymetry differencing map or something else?

Line 188-189: What evidence is there that the warm, oxygenated and highly productive nature of the water above the seamounts was relevant for the survival of species there, rather than the observation that currents may simply not have topped these higher areas?

Fig 3: Could you please put scale bars in parts A-C. Could you please label the other parts of the subfigure as well.

PDC Modeling Comments

I am extremely supportive of this manuscript overall, but have critiques and questions about the modeling effort as it currently stands.

First, it is not at all clear what aspect of the modeling result is being compared to the seafloor data or how the authors are evaluating the model. For example, what do the arrow in Fig. 2 and 3 represent from the model, a simplified version of the velocity field? If so, could so much be stated and could the authors explain how they performed this simplification? The authors claim that the model "explains the observed spatial trends in erosion and flow runout" but do not explain what aspect of the model they are comparing to field data to come to this conclusion. Indeed, the overall runout result from the model is not reported and the authors state that their model does not include erosional processes, which seems to contradict the claim that the model can be used to explain runout and erosional trends. More specific line by line comments are below.

Lines 171-177: In these sections, there is generally blurring between results and interpretations extrapolated from seafloor data vs. model results. Could the authors clarify what conclusions and results are supported by the data vs. models and/or clarify how they are integrating the two.

Modeling Methods section:

I think that the authors modeling may be interesting to a broad audience, but their methods are still unclear. There are general questions that the authors could answer specifically as they describe their model such as:

- Why a two layer model?
- Can the bottom layer entrain water from the upper layer and thus evolve in density?
- I believe that flow rheology is prescribed by their friction coefficient. If so, could the authors state this explicitly and tell the reader what they select for a friction coefficient.
- I also can't recall if the model flow eventually stops, or not? Could the authors explain whether the model gives a final runout distance and what that is. If not, could they explain why not.

Could the authors please label their equations and comment on which equations correspond to which conservation equations (i.e., mass, momentum, and energy for which layer in their model).

Line 476-478: This sentence is helpful in putting their modeling in context of VolcFlow, but I do not know what is meant by "with a more mobile dense flow following." Could the authors explain what they mean by "mobile dense flow following" and be specific about how their model is different from VolcFlow? That is, why not just use VolcFlow? What did they have to modify and why?

Lines 478-481: I am very glad that the authors have included a description of boundary conditions, but I cannot understand what they are based on what is written here.

For example, I thought at first from lines 478-480 that the authors were referring to the lower boundary that the flow is travelling over, but the later part of the sentence makes me think it is a lateral boundary. If a lower boundary, the no flux condition makes sense, but do the authors have a no-slip boundary condition as well?

Lines 480-481: I do not understand what is meant by this sentence. Are the authors talking about the upper layer of their model? The water mixed with particles in the lower layer? What is a radiating boundary condition?

Lines 482-487: Thank you for providing initial conditions here. This makes sense for the bottom layer. What about the initial conditions for the upper layer? Did the authors run models for more than one set of initial conditions, however? It would seem so based on your conclusion that the flow had to initialize inside the caldera. Perhaps a table that lists the conditions for each model run would be useful. Then you could refer to how specific model runs (as listed in the table) allowed you to draw specific conclusions.

Lines 496-499: Thank you for this information. It is helpful.

Lines 499-500: I am somewhat confused. The model cannot erode or deposit material except in hollows? How does this work? Can you clarify? What is the model doing to allow for deposition only in certain areas?

Lines 500-501: I do not understand this sentence. What is meant by “waves generated by this mechanism?”

Lines 503: What is the wave generation process that the model is simulating? How does it assess wave generation? That is, what you do examine in the model to make conclusions about wave generation? How do you define wave generation?

Lines 511-512: What are the “outputs” of the model that allow you to draw conclusions about channelization. Could you be clear about model outputs (in addition to the initial conditions you listed previously).

Line 514-515: How did you determine what was sufficient force to cause the observed cable damage? What do you consider to be the observed cable damage? From my understanding, our understanding of how the cables were damaged is quite vague (they may have been buried, broken, had their sheaths damaged, or some combination) (Clare et al., in review). How did you fine tune your model to this condition? That, are there other runs you aren't sharing with the reader. Did those runs have other results. If so, I think more justification is needed for showing only one (or two) selected model runs.

PDC Modeling Animation Video: This video is neat! Could you please put units on your timescale, x and y axes, and scale bar.

Supplemental Material:

Table S1: This is a terrific data set! I am skeptical, however, of the interpretation of ashfall vs. dilute flows. Could you please explain here or elsewhere in the text how PDC deposits were distinguished from ashfall and why you interpret these as ashfall rather than something like dilute PDCs? The 90/23 sample, 80 km from the Caldera, has ashfall thickness 2-3 times greater than those from Tongatapu (max thickness about 3 cm) and Tongatapu is less than 80 km from the Caldera, I believe.

Please remove highlighted text from the supplement.

Line 110 (supplement): Add figure number

Figures S6 and S7: Please include units in figure axes, scale bars, and descriptions.

Figures S6 and S7: I don't understand the conclusion that the PDC couldn't have initiated outside the caldera based on comparisons between these figures. The figures look broadly similar. I do not see what is different between the two sets of figures.

Figs S9 - S11: These are super neat! I am glad they are included. Could you please include scale bars in

these figures.

Congrats again on your ms. I am extremely excited about this work. Although some of my comments on the modeling section may appear critical, they are primarily aimed at understanding your work.

We thank the team from Nature Communications Ecology for their time reviewing our manuscript. Our responses to their queries and suggested changes are below.

The claim that the benthic fauna survived the pyroclastic flow-induced currents only on topographic refugia makes perfect sense and is discussed nicely on pp. 7-8, but we would like to ask you to give a bit more caution, given that "Pre-eruption deep-sea benthic surveys [...] were not available to compare with our new post-eruption data, except for discrete observation of mussel beds on a southern seamount within our study area"; in particular also since you don't really show data on those mussel beds*, just some (not super clear) pictures in the SI. We would ask you to change "the important role of refugia" to "the role of topographic refugia" in the subheading, and L201-202 to something more cautious like "to our knowledge, this is the first report suggesting a sheltering effect from PDCs in a marine setting".

*or are these in suppl table 2? I couldn't tell, there is no legend of what sites the rows represent.

We have made this suggested change to the subheading which now reads: “*Wider impacts of the eruption on ocean ecosystems and the role of topographic refugia*” and from previous edits L201-202 reads as suggested “*however, to our knowledge, this is the first time such a sheltering effect from PDCs has been reported in a marine setting.*”

Then, in L216-221 you discuss previous studies on microfauna like foraminifera, but here it's all about macrofauna, so that doesn't seem particularly relevant. You also claim that the communities on the refugia were "unaffected", which seems questionable in the lack of proper data (just because they weren't destroyed doesn't mean nothing happened). So, we suggest to condense L216-221 as something like: "We do not know the timescale over which the seafloor communities in the Hunga Volcano may recover, but we speculate that the process may be aided by re-colonisation from the refugia indicated above".

We have made this suggested change, removing lines 216-221, and using the recommended sentence to end the final paragraph of the section (see lines 220-223: “*We do not know the timescale over which the seafloor communities in the Hunga Volcano may recover, but we speculate that the process may be aided by re-colonisation from the refugia indicated above.*”)

We thank both Reviewers for their time reviewing our revisions for our manuscript, and for their helpful comments and suggestions. We note, as in our Cover Letter, that we have made a change in terminology from ‘Pyroclastic Density Currents’ to Volcanoclastic Density Currents’. We have made this change following revisions in an complementary manuscript by our co-authors (Clare et al., 2023, Science) where this change in nomenclature was requested, and this was supported by extended conversations with colleagues in the field. The terminology of Volcaniclastic Density Currents seems to be emerging as the most accepted by the field to refer to this phenomenon, particularly as this does not discriminate between where flows are hot and gas-supported (i.e. pyroclastic density currents) and where they mix with seawater and become cold turbidity currents. As a result, we think this modification will enhance the broader reach of this paper and ensure consistency with other literature. Our responses are below, line by line.

Reviewer #2 (Remarks to the Author):

I’m speaking on behalf of reviewers 1 & 2, since reviewer 1 was unable to re-review.

This paper describes the initial submarine observations of the 2022 Hunga Tonga eruption and will be an important touchstone for ongoing and future studies of this and other similar shallow water explosive eruptions. The reviewers had a lot of comments seeking clarification, addition, justification, and the authors did a great job of addressing those comments, sometimes directly and sometimes by addition or subtraction of information. The comments were wide-ranging and, I presume, not an easy task to address them all. Kudos! Overall I think the paper is nearly ready for publication, and only have a few minor points that should be addressed, along with a few specific wording suggestions. The authors should consider these comments as burnishing of a well-written paper and can feel free to address or ignore as they see fit.

Thank you very much, we appreciate this feedback and the suggestions that follow.

- I found the first few sentences of the introduction a bit confusing. Are the impacts of large explosive eruptions on land and in the marine environment being described, or just terrestrial eruptions? I was thrown off by the mention of tsunamis (which of course could refer to terrestrial eruptions - but are not a common impact) and the inclusion of reference 20. Perhaps I was influenced by the first two sentences of the abstract which more clearly stated ‘we know a lot about large eruptions on land, but very little about those in the marine environment’. If this could be clarified just a bit to better reflect the statements in the abstract it would be much more powerful.

- line 51 - add comma ‘shallow-water, active’

Line 51 now reads: ‘In particular, shallow-water, active volcanic centres’

- line 83 - replace ‘large’ with ‘wide’

Line 83 now reads: ‘roughly 3- to 4-km wide submarine’

- line 88 - replace ‘until’ with ‘culminating with’

Line 88 now reads: ‘intermittently culminating with two large eruptions’

- line 90 - remove ‘in the evening’

This has been removed

- line 91 - remove ‘99% of the volcano’. Given the comment by reviewer 1 (although addressed in the authors rebuttal) it doesn’t seem necessary to put a number on this.

To address this, and Reviewers 3 comment on this line, we have made a compromise between the options and line 93 now reads: ‘However, the modification of the submerged fraction of the volcanic edifice (~99% of Hunga Volcano) remained unknown.

- line 96 - replace ‘a decrease of >800m elevation’ with ‘an increase in depth >800m’. There is a range of terminology related to seafloor change in the paper with volume and seafloor being used interchangeably and depth and elevation being substituted. When referring to the seafloor change, I would stick with referring to changes in depth and addition or removal of volume or material.

Line 96 now reads: ‘where an increase in depth >800 m was observed’

- line 101 - ‘post eruption’ I think this is referring to features present in pre- and post-eruption data, but the wording is unclear.

Line 106 now reads: ‘visible in the post-eruption bathymetry.’

- line 101 - replace ‘following’ with ‘during’

Line 107 now reads: ‘removed during the eruption’

- line 105 - replace ‘modern echo sounder techniques’ with ‘ship-based echo sounders’. Modern echo sounders at high frequency operated near the seafloor could indeed resolve small changes.

Line 112 now reads: ‘resolved with ship-based echosounders’

- line 109 - ‘elevation’ to ‘depth’

Line 116 now reads: ‘the seafloor changes’

- line 110 - replace 'seafloor' with 'volume'

Line 118 now reads: 'Below 600 m water depth, which encompasses the flanks of Hunga Volcano and the area around the caldera rim, volume loss'

- line 111 - replace 'was focused within' to 'coincide with'

Line 119 now reads: 'which coincide with pre-existing'

- line 114 - perhaps add a clause that the 'new chute' was not present in pre-eruption data.

Line 123-124 now reads: 'within a new chute, which was not visible in pre-eruption data, that initiated'

- line 116-117 - 'distance greatest in the NW direction' I know what you're saying here, but I think most will find it confusing. Could leave out. Possible new sentence 'Most of the measured loss was within 6.5km of the caldera rim, beyond which deposition dominated.'

Lines 125-126 now read: 'Most of the measured loss was within 6.5 km of the caldera rim, beyond this deposition dominated.'

- line 130 - could also cite 'Cas, R. A., & Wright, J. V. (1991). Subaqueous pyroclastic flows and ignimbrites: an assessment. Bulletin of Volcanology, 53, 357-380' although I know you're running short on references.

Though applicable, we are already well over on references so we will refrain from citing this one.

- line 133-134 - remove 'or mass wasting of the edifice' as there is now a description of why this was unlikely.

Lines 146 to 147 now read: 'magmatic explosivity or hydrovolcanic explosions could have..'

- line 136 - rephrase to indicate channelization at the rim since that is the key piece of evidence supporting the hypothesis (e.g., channelization that was observed on the flanks of the volcano and extends up to the caldera rim).

Lines 150-151 now read: '..strong channelisation that was observed on the flanks of Hunga volcano and extends up to the caldera rim.'

- Line 138-139 - I don't think this last sentence is necessary given the argument made above. It's placement is a bit odd and reads like a response to reviewer comment.

We agree with this comment and have removed this sentence as such.

- Line 140-144 - This very long sentence should be broken up.

We broke up this section as such: “We observe crescentic scours (up to 30 m deep) and bedforms (up to 1,200 m wavelength, 20 m wave height; Fig. 2), commonly seen in other deep-sea settings affected by powerful density flows⁴⁰, including density flows triggered by large magnitude earthquakes that severed seafloor cables and attained speeds of up to 20 m/s.⁴¹⁻⁴³. These observations are further corroborated by seafloor sediment coring and imagery, geochemical analysis and numerical modelling.”

- Line 148 - remove ‘which we interpret to be a result of PDC’s’. It doesn’t seem that people need to be reminded that your talking about the PDCs.

We have made this change and lines 163 now read: “...as expected from a particulate density flow deposit.”

- Line 151-154 - I would reword the geochemistry description for clarity...’Major element chemical composition of PDC deposits was measured by XRF and overlap with those reported for recent eruptions from Hunga-Tonga Volcano (Brenna et al., 2022) and distinct from other regional volcanoes.’ I think this implies the source of the eruption and you don’t need to spell it out more explicitly. I also think you could leave out the SEM EDS analyses of the CTD-recovered ash shards here and move it to the section that talks about turbidity plumes/layers.

We think this is a reasonable suggestion and have reworded the sentences in lines 166-172 as such: “Major elemental chemical compositions of volcanoclastic deposits were measured by X-Ray fluorescence spectrometry and overlap with ashfall collected from Tongatapu post-eruption (measured by scanning electron microprobe analysis) as well as recent eruptions from Hunga volcano⁴⁴, and are distinct from other regional volcanoes (Supplemental Information; Figs. S3, S4; Supplemental Data 1).” **We still include the EDS analysis here, though, but only for the correlation with Tongatapu ashfall samples that were collected post-eruption as we think this is a useful comparison. The water column samples we discuss later on, as suggested, with reference to the data mentioned here.**

- Line 158 - ‘that eventually decelerated’ seems unnecessary. I see that you use it as a link to ‘widespread deposition’, but you could instead say ‘that blanketed the flanks of the volcano and beyond’.

We have made this change and line 177 now reads” .. blanketed the flanks of the volcano and beyond, resulting in widespread deposition.”

- Line 161 - I don’t really see the point of comparing the volume to a subaerial landslide?

We make this comparison to provide an example of volume that readers may be able to visualize easier as the St Helens landslide was very well documented by aerial photography.

- Line 164 - Saying the model replicates the erosion channels is a bit misleading since the model, as I understand it, does not allow for erosion. I would remove this or rephrase.

We have removed this as such, and the sentence now reads “..approximately 3-4 km³ of flow is required originating from inside the caldera rim to replicate the extent of the observed deposits, especially the overtopping of two submarine knolls to the south heading towards the international cable (Fig. 3).”

- Line 169 - remove ‘extensive’ seems pretty binary to me damaged/undamaged. If you do remove it, you should also remove ‘damaged’ on line 170.

We prefer to have extensive remain in this sentence as we are trying to indicate the notability of the length of damage to the cable, which in our follow up work we illustrate and discuss in more detail.

- Line 173 - replace ‘submarine’ with ‘simulated’.

We have made this change and lines 194-197 now read “As the simulated density currents flow away from the caldera, the model illustrates how this seafloor flow can be steered by the local bathymetry and how, in some locations, separate flows can converge back together and accelerate.”

- Line 186 - replace ‘occurred’ with ‘were observed’

Line 217 now read: ‘We observed relatively undisturbed and diverse benthic communities on seamounts”

- Line 188 - recommend breaking up this sentence, e.g., ‘...of our survey area, at distances >50km from the caldera. The density and diversity of these ecosystems aligns with expectations for the warm, oxygenated...’

Lines 218-221 now read: ‘The density and diversity of these ecosystems aligns with our expectations for the warm, oxygenated and highly productive waters that exist adjacent to seamounts and volcanic islands of the Tonga-Tofua-Kermedec Arc.’

- Line 193 - what is meant by ‘thickest flow paths’? Widest, deepest?

By thickest we are referring to the output of the modelling simulation, which shows the density current ‘thickness’ in meters. We have added clarification after mentions of thickest to include (in m) for enhanced clarity.

- Line 211 - confusing wording 'covering 700-750m'. Perhaps remove and put (700-750m) after Seamount.

We have reworded and line 262-263 now reads: 'In a deeper setting (700 -750 m) on Vailulu'u Seamount...'

- Line 228 - Need a period after caldera.

We have made this change

- Line 232 - clarify. 'The turbidity anomalies were sampled by CTD and the filtrate was dominated by ash particles as identified by SEM imagery and with a similar composition to the PDC deposits as measured by SEM EDS.'

We have reworded this, with the edits to lines 154-157 in mind, and lines 287-291 now read: "The turbidity anomalies (both mid-water and near-seabed layers) were sampled by CTD and in all instances were dominated by ash particles, identified by scanning electron microscope imagery, and electron microprobe analysis revealed a similar composition to the PDC deposits and post-eruption ashfall samples from Tongatapu discussed above (Figs. S4, S14-17; Supplemental Table 1). "

- Line 248 - possibly replace 'substantial' with 'potentially substantial' or just remove 'substantial'.

We removed substantial

- Line 250 - Perhaps add 'locally decimated benthic communities' to the suspended particles as impacts with potential enduring effects.

Lines 313-314 now read: "The enduring effects of suspended volcanic material in the water column and the localised decimation of benthic communities..."

Reviewer #3 (Remarks to the Author):

Review for Pyroclastic density currents explain far-reaching and diverse seafloor impacts of the 2022 Hunga Tonga Hunga Ha'apai eruption

Thank you to the authors for their thoughtful comments and revisions on the previous ms. Most of my comments focus on the modeling aspects of the work and making the figures and data as useable for a general audience as possible. I have not provided many specific comments on the biological destruction aspects because these are outside my realm of expertise.

Although the new methods section has clarified some important aspects of the model, my

opinion is that more clarity is needed in regard to both how the model works and towards how the authors are drawing specific conclusions from the modeling work (see further comments below). That is, I still do not understand many aspects of the model and specific model conclusions are not supported well by information provided in the ms, in my opinion. For example, I think Figs. Like S6 and S7 are valuable, but I do not see general differences between these two figures and that support the conclusion that flow initialization is needed inside the caldera to create channelized flows.

Overall, I think that this work is of extremely high scientific value to the community. The data sets included in this ms are large and I applaud the efforts to share them and make them broadly available with the scientific community.

We thank Reviewer 3 for their time spent on our manuscript and for the helpful comments and discussion that they have provided. We have done our best to take into account all points and enhance clarity and robustness of our manuscript whenever possible. We also thank Reviewer 3 for recognizing the contribution of this work to the scientific community, and the value of the dataset we are sharing with this manuscript. Responses are line by line below.

Line by line comments:

Line 32: I suggest making it clear that you are inferring or providing evidence that transport and deposition was by submarine pyroclastic currents in this ms.

We have modified lines 32-33 as such: “Almost 10 km³ of seafloor material was removed during the eruption, most of which we conclude was redeposited within 20 km of the caldera by long run-out seafloor density currents.”

Lines 50-51: Would there be a reference for the claim of 100,000 uncharted submarine volcanoes?

This citation had been lost when the sentence was restructured, we have corrected that and have provided the Wessel et al., 2010 citation. Thank you for noting this.

Line 74: Add the word “submarine” before “loss/gain”

We have made this change, using seafloor though instead over submarine, so that line 82 now reads: “for seafloor loss/gain”

Line 76: Was the runout truly greater than 100s of kms or simply >100 km?

We have modified this so that it reads >100 km as this encapsulates both and is indeed more to the point of what we show here.

Line 87: Would there be a reference for the start of the eruption in late 2021 and composition?

We have added two appropriate references here for the start of the eruption and the composition.

Line 92-93: Perhaps modify to something like, “However, the modification of the submerged fraction of the volcanic edifice (~99% of Hunga volcano) remained unknown.”

We like this suggested change, and it helps to resolve a comment from Reviewer 2 as well. As such, lines 102-104 now read “However, the modification of the submerged fraction of the volcanic edifice (~99% of Hunga Volcano) remained unknown.”

Line 100: Perhaps modify to, “An additional 3.5 km³ was lost from the outer flanks, without impacting Hunga’s general morphology (Fig. 1). Many small features, such as peaks, ridges and gullies, were visible in the bathymetry post-eruption.”

Combining this suggestion with a comment from Reviewer 2, lines 110-113 now read: “An additional 3.5 km³ was lost from the outer flanks, without impacting the general morphology of Hunga Volcano (Fig. 1). Many small features present pre-eruption, such as peaks, ridges and gullies, were visible in the post-eruption bathymetry.”

Line 105 – 107: I am unclear what this sentence is saying or suggesting, could you please clarify.

We made changes to lines 113-119 to enhance clarity in this section. These lines now read: “Of the total 9.5 km³ of material removed during the eruption, 66% (6.3 km³) of this material was deposited within 20 km of the caldera rim and 3.2 km³ remains unaccounted for. Prior studies suggest that ~1.9 km³ of this material that remains unaccounted for was ejected into the atmosphere as an eruption plume³⁷. The remaining material that is unaccounted for (~1.3 km³) was likely deposited as widespread thin deposits below the detection limit of what can be resolved with ship-based echosounders.”

Line 108: Could you make clear that this 600 m threshold refers to the flanks of Hung and the area around the caldera rim? Currently, the depth threshold doesn’t make a lot of sense in terms of what spatial area it is referring to.

We have added this clarification so that lines 124-126 now read: “Below 600 m water depth, which encompasses the flanks of Hunga Volcano and the area around the caldera rim, volume loss is confined to linear chutes that radiate from the caldera rim..”

Line 111: What is meant by the term “entrenchment”? I am not familiar with the use of this term in this way.

We have replaced entrenchment with deepening for more clarity so lines 127-128 now read: “Up to 70 m deepening occurred within the incised chutes”

Lines 113-114: Modify to “widespread blanketing of sediment”

We have made this change and line 130 now reads: “with widespread blanketing of sediment across the pre-eruption seafloor.”

Line 138-139: Is the rationale here that the channels begin close to or at the Caldera rim, and therefore the flows must have been fully formed by the time they reach the caldera rim?

We have removed this sentence following further discussions for enhanced clarity, and modified the sentence prior so that lines 157-160 now read: “Sensitivity tests of our density current model (see Supplementary Material, Fig. S6), however, confirm that whatever initiation mechanism caused them, it most likely occurred within the caldera in order to cause the strong channelisation that was observed on the flanks of Hunga volcano and extends up to the caldera rim.”

Line 141: Can you point these scours out in Fig. 2 with an arrow?

We attempted this but at the scale was difficult to see well, however we hope that the enhanced colour mapping that we have provided in response to the previous revisions, and the two A and B profile lines showing vertical gain/loss help to clarify the location of some of these features.

Line 149-150: Are the ripples on the seafloor or within the deposit?

The ripples we refer to here were within the volcanoclastic (PDC) deposit and also visible in the seafloor photos that we provide in Fig. S9

Line 153: Are you saying the submarine deposits have the same composition as ashfall from 2022 Hunga. Or, that both submarine and ashfall deposits have the same composition as previous Hunga eruptions.

Following suggestions by Reviewer 2 and this comment we have modified this part so that lines 177-182 now read: “Major elemental chemical compositions of volcanoclastic deposits were measured by X-Ray fluorescence spectrometry and overlap with ashfall collected from Tongatapu post-eruption (measured by scanning electron microprobe analysis) as well as recent eruptions from Hunga volcano⁴⁵, and are distinct from other regional volcanoes ...” We hope that these changes clarify that we are referring to the composition of the volcanoclastic (PDC) deposit, which matches the ashfall collected from Tongatapu post-eruption and data from previous eruptions, with these compositions all distinct from other regional volcanoes. We have removed comparison to the water column data from this section and just discuss it later on for clarity purposes.

Line 158: What was the evidence for local erosion? The negative difference in part of the bathymetry differencing map or something else?

You are correct that the evidence for local erosion here is the negative difference in the bathymetry difference mapping.

Line 188-189: What evidence is there that the warm, oxygenated and highly productive nature of the water above the seamounts was relevant for the survival of species there, rather than the observation that currents may simply not have toppled these higher areas?

We edited this section slightly in response to a comment from Reviewer 2 so that lines 203-213 now read: “While the seafloor in most of the remaining survey area was smothered with volcanic deposits linked to volcanoclastic density currents and subsequent ashfall, areas of potential refugia from the impacts of density currents appear to have been created locally by the high and irregular seafloor relief provided by several seamounts (Fig. 3, S11, S19, Supplemental Data 2). We observed relatively undisturbed and diverse benthic communities on seamounts in the southern part of our survey area, at distances greater than 50 km from the caldera. The density and diversity of these ecosystems aligns with our expectations for the warm, oxygenated and highly productive waters that exist adjacent to seamounts and volcanic islands of the Tonga-Tofua-Kermadec Arc ⁴⁷⁻⁴⁹. Closer to Hunga Volcano, abundant benthic and fish communities were also observed on seamounts in the area, presumably protected from direct eruption impacts (Fig 3; Fig. S11, S19).

We hope that this provides more clarity to what we are referring to here, which is that these areas appear to have not been impacted by the volcanoclastic deposits (PDC's), most likely because the currents did not reach them due to the topographic refugia provided by the seamounts.

Fig 3: Could you please put scale bars in parts A-C. Could you please label the other parts of the subfigure as well.

We have added these labels and scale bars to the figure.

PDC Modeling Comments

I am extremely supportive of this manuscript overall, but have critiques and questions about the modeling effort as it currently stands.

Thank you, it is useful to hear what aspects you don't understand as it helps us explain the model better. We have made changes to provide more clarity, as detailed below.

First, it is not at all clear what aspect of the modeling result is being compared to the seafloor data or how the authors are evaluating the model. For example, what do the arrow in Fig. 2 and 3 represent from the model, a simplified version of the velocity field? If so, could so much be stated and could the authors explain how they performed this simplification? The authors claim that the model “explains the observed spatial trends in erosion and flow runout” but do not explain what aspect of the model they are comparing to field data to come to this conclusion. Indeed, the overall runout result from the model is not reported and the authors state that their model does not include erosional processes, which seems to contradict the

claim that the model can be used to explain runout and erosional trends. More specific line by line comments are below.

While our model can't predict specifically where erosion or deposition will occur as the gravity current is simply modelled as a dense fluid, it can predict where the gravity current will flow, and erosion and deposition due to the gravity current can only occur on that flow path. These flow paths (and the thickness and speed of the gravity current along them) compare well with sea floor locations where erosion, deposition, cable damage and loss of fauna occur. We have updated wording in the manuscript for increased clarity on this, though, for example removing reference to the erosional channels on the flank of the Hunga edifice so that now lines 169-172 say:

"...approximately 3-4 km³ of flow is required originating from inside the caldera rim to replicate the extent of the observed deposits, especially the overtopping of two submarine knolls to the south heading towards the international cable..." More specific changes and clarifications follow.

The arrows on Figures 2 and 3 are an attempt to distil the time varying paths of the modelled density (PDC) currents onto a map which also includes other information. They were sketched onto the map from the animation of the density current (PDC) (which shows the evolution of its thickness, h_s). We have attempted to make this clearer by stating "Indicative density current flow paths (i.e., an accessible visualisation of the general flow pathways simplified from the model output animation provided (see Supplemental Video 1 and Fig. S6)) are indicated with dashed grey lines plotted over the difference map to enable comparison of density current flow paths with the difference map results. We also removed the arrows from Fig. 2 to decrease confusion and we have modified Fig. 2 to include snapshots from the modelling animation at 120, 300, and 600 seconds into the modelling run to allow for direct comparison in the figure, as well as stating that comparisons can be made with the Supplemental Video which shows the entirety of the modelling animation. This is stated as such: "For further context, select snapshots from the model animation (provided in Supplemental Video 1) are provided at 120, 300, and 600 seconds into the modelling output. For these density current modelling snapshots, the map frame extent is the same as for the difference map, and the background imagery is 2022 multidirectional hill shade." After careful consideration, we want to leave the lines of the indicative flow paths in Fig 2 and Fig 3, however, because we think they are very important to the readers ability to see how the differences shown in the difference map of the bathymetry correlate (or not) with the main areas crossed by the simulated PDC flows.

For Fig. 3, which shows these same indicative flow paths from the modelling animation but over a larger map frame, we have left the arrows but removed the size differences in the flow path lines. We have left the arrows because they are important for the readers ability to understand the directionality of the lines, which we can accurately interpret from the modelling animation as the PDC simulation moves through times, and also the fact that the PDC would not be ending at the end of the line provided. We agree that the size difference between the lines was confusing and unnecessary though, so this has been

removed to allow for enhanced clarity. We have also provided more detail in the figure caption to make this interpretation clearer: “Indicative density current flow paths taken from modelling results (as in Fig. 2, see Supplemental Video 1 and Fig. S6) are shown in solid black lines with the arrows indicating directionality of the flow paths interpreted from the modelling simulation. The locations of damaged submarine cable are depicted with dashed pink lines.”

Lines 171-177: In these sections, there is generally blurring between results and interpretations extrapolated from seafloor data vs. model results. Could the authors clarify what conclusions and results are supported by the data vs. models and/or clarify how they are integrating the two.

We have tried to make it clearer which parts of this refer to modelling results and which to seafloor data. The general message is that locations where the model shows the gravity current went are where sea floor effects were seen. These changes span the sections with lines 204-219:

“Furthermore, the modelled flow paths taken by the faster moving and thicker (in m) parts of the density current correlate well with the observed spatial trends in erosion and flow runout. The most pronounced erosion occurred immediately to the NNW of the caldera, where the modelling shows part of the density current being funnelled through an area of negative seafloor relief between the two islands. As the simulated density currents flow away from the caldera, the model illustrates how this seafloor flow can be steered by the local bathymetry and how, in some locations, separate flows can converge back together and accelerate. These modelled results illustrating the paths of the simulated density currents corroborate the areas of impact observed in the seafloor surveys presented here. The longest observed flow runout likely exploited bathymetric lows to the NW of the caldera. Post-eruption bathymetry (corroborated by modelling results) indicates this density current went well-beyond the northwest limit of our survey area. Flow paths like this have previously been observed subaerially but only inferred in submarine settings³⁹.

.”

Fig 3: Could you please put scale bars in parts A-C. Could you please label the other parts of the subfigure as well.

We added the scale bar to be in all A-C photos in this inset, and have added labels to all three parts of the figure (i-iii). We adjusted the figure caption to have more clarity with these changes, as well as to have more clarity in relation to the use of the lines in the figure.

“Figure 3: (i) Sediment core log with interpreted pre-eruption, volcaniclastic and ash deposits labelled, from the sediment core indicated with the light blue star in (ii). (ii) Compilation map showing seafloor multibeam survey and geologic and biological sampling after the 2022 Hunga

Volcano eruption. Indicative density current flow paths taken from modelling results (as in Fig. 2, see Supplemental Video 1 and Fig. S6) are shown in solid black lines with the arrows indicating directionality of the flow paths interpreted from the modelling simulation. The locations of damaged submarine cable are depicted with dashed pink lines. In order to reach the international cable, the density current must overtop two knolls directly E and ESE of location C, which partially constrains the minimum volume of material in the volcanoclastic density current. The extent of the post-eruption bathymetry survey is shown in colour and was collected by RV Tangaroa and USV Maxlimer. Dark blue graduated points and black crosses show locations where seafloor video footage documented invertebrate abundance (in count) during RV Tangaroa survey in April 2022. (iii) Example images of seabed across the study area are shown in insets A-C, and each image is located in main figure. The white bar scale in each A-C inset is 10 cm, as indicated in the middle inset (B).”

Modeling Methods section:

I think that the authors modeling may be interesting to a broad audience, but their methods are still unclear. There are general questions that the authors could answer specifically as they describe their model such as:

- Why a two layer model?

The lower layer of the model is the density current and the upper layer is the overlying water. This allows us to model the flow of the volcanoclastic density current over complex bathymetry with the effects of the water above them also taken into account. The assumption is that both these layers are vertically integrated. This allows for the density current to be influenced by the water above it and the water to be influenced by the movement of the density current below but there is not mixing This model was originally developed to try and understand the tsunami generated by the density current.

- Can the bottom layer entrain water from the upper layer and thus evolve in density?

No, in this simple model the two layers are impermeable.

- I believe that flow rheology is prescribed by their friction coefficient. If so, could the authors state this explicitly and tell the reader what they select for a friction coefficient.

Yes, that is correct. We have now stated this in line 528: “in this model a simple quadratic friction is used and the coefficient is set to 10^{-4} for the gravity current...”

- I also can't recall if the model flow eventually stops, or not? Could the authors explain whether the model gives a final runout distance and what that is. If not, could they explain why not.

We have clarified this further in the paragraph starting at line 569 this simple model assumes that the density current is a dense Newtonian fluid following physical experiments by Bouguin et al which showed that this captures many of the aspects of a density current. One of the outcomes of this is that the flow only comes to a stop when it reaches a depression. This is why we were careful to state that the model is not able to predict the final runout distance of the density current. This is provided in lines 569-575: “This modelling is a very simple representation of a very complicated process. While it captures some aspects well it is not able to resolve other aspects. The gravity current is represented as a dense Newtonian fluid flowing over an unerodable bed. It does not erode sediment as it flows down the flanks of the edifice and neither does sediment deposit when the speed decreases, it just flows to the lowest point in the bathymetry. Thus, this model cannot predict where the density current will finally stop, except in depressions or hollows.”

Could the authors please label their equations and comment on which equations correspond to which conservation equations (i.e., mass, momentum, and energy for which layer in their model).

We have modified these sections to provide extensive further explanation of each variable in the equation means, how it is used together, and what is gleaned from each portion.

Line 476-478: This sentence is helpful in putting their modeling in context of VolcFlow, but I do not know what is meant by “with a more mobile dense flow following.” Could the authors explain what they mean by “mobile dense flow following” and be specific about how their model is different from VolcFlow? That is, why not just use VolcFlow? What did they have to modify and why?

Thank you for pointing out that this doesn’t make sense – the sentence was supposed to read “... following Bouguin et al” but the referencing system messed it up. We have reworded this to make it clear. The results of Bouguin et al show that aerated granular flows entering water behave very similarly to dense fluids. As discussed during our meeting, we did not use VolcFlow because we do not have access to this software.

Lines 478-481: I am very glad that the authors have included a description of boundary conditions, but I cannot understand what they are based on what is written here. For example, I thought at first from lines 478-480 that the authors were referring to the lower boundary that the flow is travelling over, but the later part of the sentence makes me think it is a lateral boundary. If a lower boundary, the no flux condition makes sense, but do the authors have a no-slip boundary condition as well?

We think that it was unclear that this is a vertically-averaged equation and therefore only have lateral boundary conditions. We have further emphasised this in the manuscript. In the vertically-averaged equations the surface and bottom boundary conditions from the 3D equations are incorporated within the equations (this is how η and z_b enter into the equations).

This is given in lines 550-554: “The lateral boundary condition for the volcanoclastic density current (the lower layer) is a solid boundary (no flux over the boundary) but the boundary of the model is sufficiently far from the volcano that the density current does not reach it. The sea water (upper layer) is solved with a radiating boundary condition at the edge of the domain, which allows for waves to propagate out of the domain and not reflect back in.”

Lines 480-481: I do not understand what is meant by this sentence. Are the authors talking about the upper layer of their model? The water mixed with particles in the lower layer? What is a radiating boundary condition?

As detailed above, we have reworded this to make it clearer that we are referring to the upper sea water layer and to explain a radiating boundary condition.

Lines 482-487: Thank you for providing initial conditions here. This makes sense for the bottom layer. What about the initial conditions for the upper layer? Did the authors run models for more than one set of initial conditions, however? It would seem so based on your conclusion that the flow had to initialize inside the caldera. Perhaps a table that lists the conditions for each model run would be useful. Then you could refer to how specific model runs (as listed in the table) allowed you to draw specific conclusions.

The upper layer is just initialised as a flat surface above the density current. This has now been specified in the manuscript. While other initial conditions could affect the waves generated, they are unlikely to have a significant effect on the density current underneath.

This is given in lines 560-561: “The upper sea water layer is initialised as a flat surface ($\eta=0$) which then evolves according to the movement of the gravity current beneath it.”

Lines 496-499: Thank you for this information. It is helpful.

Lines 499-500: I am somewhat confused. The model cannot erode or deposit material except in hollows? How does this work? Can you clarify? What is the model doing to allow for deposition only in certain areas?

The model does not include erosion or deposition at all. The dense fluid will eventually come to a stop when it ends up in a depression. We have rephrased this to make it clear that we are not referring to deposition here but where the fluid might end up. This is in lines 627-639: “It does not erode sediment as it flows down the flanks of the edifice and neither does sediment deposit when the speed decreases, it just flows to the lowest point in the bathymetry. Thus, this model cannot predict where the density current will finally stop, except in depressions or hollows. If the volcanoclastic density current is created by a column collapsing back into the ocean and mixing with the water, then this model will also underestimate the waves generated as it does not consider the effect of the volcanic material falling back into the water. It can, however, capture the motion of the gravity current as it

cascades down the sides of the edifice and the wave generation process in the upper layer that occurs due to this (see Supplemental Video 2).”

Lines 500-501: I do not understand this sentence. What is meant by “waves generated by this mechanism?”

The motion of the density current creates waves in the overlying sea water layer. I have re-written this to try and make it clearer. See lines 633-638: “If the volcanoclastic density current is created by a column collapsing back into the ocean and mixing with the water, then this model will also underestimate the waves generated as it does not consider the effect of the volcanic material falling back into the water.”

Lines 503: What is the wave generation process that the model is simulating? How does it assess wave generation? That is, what you do examine in the model to make conclusions about wave generation? How do you define wave generation?

This refers to the waves that are generated in the seawater by the movement of the PDC underneath the seawater. The waves can be simply seen when looking at the surface of the sea water. We have provided an additional supplemental video (see Supplemental Video 2) to illustrate what we are talking about here.

Lines 511-512: What are the “outputs” of the model that allow you to draw conclusions about channelization. Could you be clear about model outputs (in addition to the initial conditions you listed previously).

These conclusions are mostly based on the thickness of the density current which is channelled down the erosion scars compared with the rest of the edifice. This is considered in comparison to the observations from the voyage, notably the difference mapping and evidence of chutes, erosional scars, and channelisation in the bathymetry. We have clarified this section as such (lines 650-655): “This initialisation shows that the strong channelisation of the flow on the flanks of the Hunga edifice that match with the chutes, erosional scars, and channelisation observed in the bathymetric data from the voyage does not occur when the volcanoclastic density current originates outside the caldera (see Fig. S7 top left panel). Because this channelisation was clearly observed, that is evidence that the volcanoclastic density current must have originated within the caldera.”

Line 514-515: How did you determine what was sufficient force to cause the observed cable damage? What do you consider to be the observed cable damage? From my understanding, our understanding of how the cables were damaged is quite vague (they may have been buried, broken, had their sheaths damaged, or some combination) (Clare et al., in review). How did you fine tune your model to this condition? That, are there other runs you aren’t sharing with the reader. Did those runs have other results. If so, I think more justification is needed for showing only one (or two) selected model runs.

For smaller initial conditions of density current volume the density current is not able to reach over the underwater passes to the south of the Hunga Volcano and reach where the international cable lies. The density current is strongly steered by the bathymetry, so as long as the initial volume is sufficiently large it will reach the international cable with sufficient thickness and speed to bury it or drag it. In one location the cable was moved about 5 km towards the Hunga Volcano. (Clare et al., 2023) – our model also shows the density current sweeping up in this direction.

PDC Modeling Animation Video: This video is neat! Could you please put units on your timescale, x and y axes, and scale bar.

We have added units and a scale bar as requested.

Supplemental Material:

Table S1: This is a terrific data set! I am skeptical, however, of the interpretation of ashfall vs. dilute flows. Could you please explain here or elsewhere in the text how PDC deposits were distinguished from ashfall and why you interpret these as ashfall rather than something like dilute PDCs? The 90/23 sample, 80 km from the Caldera, has ashfall thickness 2-3 times greater than those from Tongatapu (max thickness about 3 cm) and Tongatapu is less than 80 km from the Caldera, I believe.

We could see the ashfall layer clearly from the CT scans, we have tried to illustrate these separations in the diagrams provided in Figure S1 and S2, with Figure S1 detailing the descriptions of the ashfall, which differ from the descriptions of the volcanoclastic deposit. Whereas the volcanoclastic density current deposit has laminations of volcanoclastic sands, this layer was finely laminated volcanic glass, a typical ashfall deposit. You are correct that it is intriguing that some of the locations further from Hunga Volcano had more ashfall than those closer, we credit this to the transport of the ash by atmospheric and ocean currents. Indeed, since the eruption we have heard from others in the community about significant ashfall deposits observed significant distances away from Hunga Volcano which appear linked to the eruption. We think that this is really interesting, particularly when coupled with the observations of volcanic glass still potentially settling out in some portions of the water column.

Please remove highlighted text from the supplement.

We have removed this highlighted text

Line 110 (supplement): Add figure number

We have added the figure number

Figures S6 and S7: Please include units in figure axes, scale bars, and descriptions.

We have added these

Figures S6 and S7: I don't understand the conclusion that the PDC couldn't have initiated outside the caldera based on comparisons between these figures. The figures look broadly similar. I do not see what is different between the two sets of figures.

We also further explain this in the newly provided Table S2, and have modified Figures S6 and S7 to zoom in on the differences between the channelisation when the volcanoclastic density current initiates outside or inside the caldera to enable better comparison. Comparing the top left images in S6 and S7, in S6 (where the density current originates inside the caldera, the thickest parts of the current can still be seen flowing down the channels where the erosion was seen. In the image in S7 the density current quickly flows equally down all the flanks of the edifice and is not steered into the channels. Thus, this shows that if the density current forms outside the caldera it is not steered into the channels and couldn't cause the erosion that is seen from the expedition.

Figs S9 - S11: These are super neat! I am glad they are included. Could you please include scale bars in these figures.

Thank you! Unfortunately it is not possible to include scale bars in these, as some of the images do not have the lasers apparent for us to scale, and each image is a slightly different scale (as you can see in Fig 3). We provide them just to give a general look at the seafloor in these regions, and note that in the description of the DTIS we do specify that it flies about 1 – 1.5 m above the seafloor, if that helps give approximate scaling of the images. We can at request though provide raw copies of the images if there are some of interest.

Congrats again on your ms. I am extremely excited about this work. Although some of my comments on the modeling section may appear critical, they are primarily aimed at understanding your work.

Thank you very much! We really appreciate your feedback and your time in discussions with us to help make sure this manuscript is as good as possible!

REVIEWERS' COMMENTS

Reviewer #3 (Remarks to the Author):

I want to congratulate the authors again on a very exciting and nicely written paper. I especially enjoyed the introduction.

Overall, the modeling descriptions are very much improved. I think that this ms is ready for publication, especially after addressing several minor comments below.

Here I summarize my main comments. More specifics are given below.

1) The ability for the model to address flow runout could be further clarified. Specifically, it is stated that the model cannot be used to address runout because (other than pooling in topographic lows), there is no physical mechanism in the model that stops the current. The authors then later use runout distance, as demonstrated by the model, to make interpretations about flow dynamics.

2) The content and initial conditions used to create Figures S6 and S7 could be clarified. At present, it is very difficult to interpret these.

3) I disagree with the authors' interpretation of fallout deposits on the seafloor in Table S1. Specifically, the authors do not explain what they use to interpret portions of the seafloor deposits as fallout. The fallout deposits on land are much less thick (up to 3.5 cm) than those they interpret on the seafloor at further distances from the vent. There is a strong possibility, I think, that these seafloor deposits could be settling of material suspended by PDCs in the water column or deposits from the dilute and upper portions of currents. I suggest, at minimum, caveating the interpretation of these deposits as fallout and acknowledging the possibility that other processes could form the upper and more laminar portions of deposits in their cores.

I feel that these comments and questions are fairly minor but worth addressing. Congrats again on a great manuscript.

Line by line comments:

Line 121: "Below 600 m water depth" is confusing. Could you specify whether "below" refers to more or less shallow?

Line 131: Typo

Line 545-546: I am still confused by the lateral boundary condition for the volcanoclastic current. This is related to my lack of understanding about what causes the model current to stop (if it does). With a free slip condition, I would expect that the current does not stop. That is, I would expect that you do not

model net deposition (other than in low points where the current may pool). Could you clarify the processes that would cause the model current to stop or not (i.e., is the friction in your model enough to eventually stop the current?). Assuming that I am correct in that there is not process that would allow the model current to stop, I don't understand why the current wouldn't reach the lateral boundary. Do you simply stop the model run before the current could reach the boundary? If so, could you clarify this point and state clearly that your model cannot examine runout distance (since there would be no physical process that allows the current to stop). I now see that you mention this point about runout on line 569. You could also mention that you therefore cannot determine a minimum (or maximum) runout distance.

Line 551-552: Could you please add here water the mixture ratio is of seawater to particulate matter that you assume for the current?

Line 591-593: Could you please elaborate on this point? How do you determine what the minimum force would be to cause the observed damage? How do you relate that minimum force to the force of the flows (especially since, to my knowledge, there isn't a process that can extract energy from your model flows like there would be in a natural current).

Line 594 and 595: Can you clarify how you are interpreting runout distance. Earlier you state that you cannot specifically examine runout because there isn't a depositional process in your model. Here, however you state that small flows don't extend to the cables. Is this because of pooling in topographic lows? If so, can you state explicitly?

Line 595 – 598: I do not understand this sentence and how it follows from your previous one.

Table S1: This table is super important and I appreciate having it here. I would caution the authors against putting interpretive ashfall thicknesses here because: 1) It is not clearly explained in the text how the authors distinguish seafloor ashfall deposits from low energy dilute flows that could have also made deposits that appear laminar on the scale of a core; 2) The maximum ashfall thickness observed on land was ~3.5 cm (Kelly et al., in review), which is inconsistent with the large ashfall thicknesses (up to 7.5 cm) interpreted in this table. At minimum, I suggest acknowledging that these are possible fall deposits, but that similar deposits could result from settling of particles in the water column after the passing of large flows or low energy dilute currents.

Supplementary Figures S6 and S7:

The content of these figures is still confusing and unclear. Could you please put the initial conditions for each set of model runs in the caption to each figure. Currently, it is very challenging to understand the initial conditions for each set of figures. I am unclear on what "modeled" vs. "fountaining" density current means for example.

I also don't understand what is going on in the top left panel in both S6 and S7. The scale bar is not labelled and it appears that the currents extend further at 300 s than they do at 2400 s. Could you please label each subfigure as a, b, and c and explain the conditions for each subfigure in the caption. Please change only one variable at a time as you compare subfigures for clarity.

Supplementary Videos: I like the Density current model video. I can't find captions or explanations for either video, however. As a result, they are hard to understand (especially the wave animation). I don't understand what the wave animation is showing. Maybe I just missed seeing the caption somewhere.

The density current video also seems to contradict your statement in the text about the current not hitting the lateral boundary (the current appears to reach the edge of your frame).

Fig. S19 is neat!

REVIEWERS' COMMENTS

Reviewer #3 (Remarks to the Author):

I want to congratulate the authors again on a very exciting and nicely written paper. I especially enjoyed the introduction.

Overall, the modeling descriptions are very much improved. I think that this ms is ready for publication, especially after addressing several minor comments below.

Here I summarize my main comments. More specifics are given below.

- 1) The ability for the model to address flow runout could be further clarified. Specifically, it is stated that the model cannot be used to address runout because (other than pooling in topographic lows), there is no physical mechanism in the model that stops the current. The authors then later use runout distance, as demonstrated by the model, to make interpretations about flow dynamics.
We have added additional clarification to the main text and the methods to increase clarity, as detailed below.
- 2) The content and initial conditions used to create Figures S6 and S7 could be clarified. At present, it is very difficult to interpret these.
We have made several changes to Figures S6 and S7 as well as the captions and the reference to the figures in the text, as detailed below.
- 3) I disagree with the authors' interpretation of fallout deposits on the seafloor in Table S1. Specifically, the authors do not explain what they use to interpret portions of the seafloor deposits as fallout. The fallout deposits on land are much less thick (up to 3.5 cm) than those they interpret on the seafloor at further distances from the vent. There is a strong possibility, I think, that these seafloor deposits could be settling of material suspended by PDCs in the water column or deposits from the dilute and upper portions of currents. I suggest, at minimum, caveating the interpretation of these deposits as fallout and acknowledging the possibility that other processes could form the upper and more laminar portions of deposits in their cores.
We have added in a caveat in the method that the ashfall could be density current deposition or other particles, and add explanation of why we interpret it as ashfall. See below for specifics on this.

I feel that these comments and questions are fairly minor but worth addressing. Congrats again on a great manuscript.

Thank you very much! Also, thank you again for your time in reviewing this manuscript. We have responded to the three main comments above throughout the more specific comments below.

Line by line comments:

Line 121: “Below 600 m water depth” is confusing. Could you specify whether “below” refers to more or less shallow?

We have adjusted this to read: “In water depths deeper than 600 m”

Line 131: Typo

Corrected

Line 545-546: I am still confused by the lateral boundary condition for the volcanoclastic current. This is related to my lack of understanding about what causes the model current to stop (if it does). With a free slip condition, I would expect that the current does not stop. That is, I would expect that you do not model net deposition (other than in low points where the current may pool). Could you clarify the processes that would cause the model current to stop or not (i.e., is the friction in your model enough to eventually stop the current?). Assuming that I am correct in that there is not process that would allow the model current to stop, I don't understand why the current wouldn't reach the lateral boundary. Do you simply stop the model run before the current could reach the boundary? If so, could you clarify this point and state clearly that your model cannot examine runout distance (since there would be no physical process that allows the current to stop). I now see that you mention this point about runout on line 569. You could also mention that you therefore cannot determine a minimum (or maximum) runout distance.

We have edited and rearranged our explanation of this section so that is clearer and more explicit. As you note here in the comment, the model simulation is stopped before the current reaches the boundary. See lines 557 – 562: *“The lateral boundary condition for the volcanoclastic density current (the lower layer) is a solid boundary (no flux over the boundary) but the boundary of the model is sufficiently far from the volcano that the density currents that are not stopped by the bathymetry do not reach the boundary within the simulation time (the boundary is beyond the geographic limit of all figures and outputs here, and thus is not shown in any density current model outputs included).”*

We additionally further clarify later on that we cannot determine run out distance but only determine where the current could have been stopped due to bathymetric barriers (as detailed below and in lines 585-589: *“Thus, this model cannot predict where the density current will finally stop (i.e. runout distances), except in depressions or hollows, but it can indicate where the density current may have enough momentum to overcome bathymetric barriers (such as the saddles to the south of Hunga Volcano).”*

Line 551-552: Could you please add here water the mixture ratio is of seawater to particulate matter that you assume for the current?

We have adjusted lines 577-580 to include the mixture ratio so that they now read: *“The gravity current was initialised as an 80 m thick, 4 km radius circle of dense fluid⁶⁷ (density 1,600 kg/m³ and total volume of 4 km³ which for instance could represent a gravity flow made up of 1.6 km³ of volcanoclastic material of density 2,500 kg/m³ mixed with seawater, at a volumetric ratio of 2:3) on top of the Hunga Volcano edifice.”*

Line 591-593: Could you please elaborate on this point? How do you determine what the minimum force would be to cause the observed damage? How do you relate that minimum force to the force of the flows (especially since, to my knowledge, there isn't a process that can extract energy from your model flows like there would be in a natural current).

To address this point, and the following two points, we have made significant edits to the entire paragraph referenced in this three points, as you will see in the following responses. We have clarified wording that was misleading (e.g. force of flows) and more explicitly explained what dynamics of the model run we are referring to, differences between different iterations, and relations to what was observed on the seafloor.

Lines 610 – 613 now read: *The total volume of the volcanoclastic density current specified in our final model was chosen based on the conditions that resulted in a density current that was able to reach the international cable, which is known to have been extensively damaged following the Hunga eruption (at least 89 km was buried or damaged by a volcanoclastic density current²⁹).*

Line 594 and 595: Can you clarify how you are interpreting runout distance. Earlier you state that you cannot specifically examine runout because there isn't a depositional process in your model. Here, however you state that small flows don't extend to the cables. Is this because of pooling in topographic lows? If so, can you state explicitly?

As stated above, this section has been significantly rewritten. As we have clarified in edits to lines 585-589 (provided above) the model does not allow for an understanding of overall runout but can depict where flows of certain volumes and densities will deposit (e.g. hollows, depressions) and when certain flows can overcome bathymetric barriers such as saddles. We have clarified the wording in this section to make that more clear, that it is really the difference in ability to overcome the topographic relief that makes the difference between the 2 km³ and 4 km³ volume flows, and without overcoming this saddle the density current cannot reach the international cable, which we know it did.

Lines 613-619 now read: *Small volume density currents that were modelled with an initial volume of 2 km³ or less had insufficient inertia to overcome the topographic relief (specifically the saddles in the ridges to the south of Hunga edifice; see Fig. S7 (b)) and were thus incapable of reaching the international cable. The model runs presented in the manuscript are based on an initial density current volume of 4 km³ that resulted in flows that reached, and ran out beyond, the international cable.*

Line 595 – 598: I do not understand this sentence and how it follows from your previous one.

We have reworded this section to make it more clear that here we are not referring to volume differences, but differences in the density of that volume (seawater to volcanic material ratio). Lines 619 – 622 now read: *Sensitivity tests on the density of the density current show that the distance the flow reaches is relatively insensitive to the density of the gravity current (i.e., the ratio of volcanic material to seawater) with only minor differences being observed for densities of 1,400kg/m³ and 1,800kg/m³ (see Fig. S7, (c) and (d)).*

Table S1: This table is super important and I appreciate having it here. I would caution the authors against putting interpretive ashfall thicknesses here because: 1) It is not clearly explained in the text how the authors distinguish seafloor ashfall deposits from low energy dilute flows that could have also made deposits that appear laminar on the scale of a core; 2) The maximum ashfall thickness observed on land was ~3.5 cm (Kelly et al., in review), which is inconsistent with the large ashfall thicknesses (up to 7.5 cm) interpreted in this table. At minimum, I suggest acknowledging that these are possible fall deposits, but that similar deposits could result from settling of particles in the water column after the passing of large flows or low energy dilute currents.

We have added a section in the methods to detail how we are distinguishing the ashfall, acknowledging the possibility that it could be associated to other factors, and detailing why we report them to be ashfall here. In short, the fact that we were still seeing ash fall out in the water column supports our interpretation of these being ashfall, and (while not discussed here and remaining unpublished) sediment core samples from further away from the volcano sampled very thick (>15 cm) ash deposits that were correlated to the eruption. Hopefully we will see these results out soon by the team involved! As we note in our changes, the potential differences between ashfall or other particle settling does not changes any of our interpretations. See lines 395-408 for our adjustment of text: “Core sample observations were synthesised in a visual sedimentological log (Fig. 3, S1, S2). Deposits were visually differentiated on the basis of their composition, colour, sedimentary textures and grain size, which included the identification of two main facies that are linked to the Hunga volcano eruption. These are i) Volcaniclastic density current deposits, which comprise dominantly sand to granule-sized volcanic material that generally fines upwards and a very sharp basal contact (where it was sampled); and ii) A thin veneer of orange-brown oxidised ash-rich deposits that has no obvious internal structure and that was sampled at seafloor, consistently overlying the density current deposits. As continued suspension and settling of ash was observed by video and sampling of the water column three months after the eruption (Fig. S13-S15), we consider these surficial seafloor deposits to be most likely related to ash fall out; however, it is possible that this facies may relate to the fall out of ash-sized material that was suspended by the volcaniclastic density current as a dilute cloud and subsequently settled out after the deposition of the coarsest load of the current, or that this relates to settling from a surface plume. Regardless, this does not affect the key conclusions of this present study.”

Supplementary Figures S6 and S7:

The content of these figures is still confusing and unclear. Could you please put the initial conditions for each set of model runs in the caption to each figure. Currently, it is very challenging to understand the initial conditions for each set of figures. I am unclear on what “modeled” vs. “fountaining” density current means for example.

We have explanations of the initial conditions and numbering to Table S2 and have now linked each of the images in S6 and S7 to the model run they came from.

I also don't understand what is going on in the top left panel in both S6 and S7. The scale bar is not labelled and it appears that the currents extend further at 300 s than they do at 2400 s. Could you please label each subfigure as a, b, and c and explain the conditions for each subfigure in the caption. Please change only one variable at a time as you compare subfigures for clarity.

We have now labelled the subfigures. We have also more clearly stated that S6 (a) and S7 (a) are close-ups (extent shown in S6 (b)) so there is no confusion about this. The initial conditions for each figure is linked to the Table S2 to make it clear what is being modelled and we have also indicated which subfigures to compare so that only one variable is changing in the comparison.

Supplementary Videos: I like the Density current model video. I can't find captions or explanations for either video, however. As a result, they are hard to understand (especially the wave animation). I don't understand what the wave animation is showing. Maybe I just missed seeing the caption somewhere.

There should have been captions that came through with the video, but this may not be working in the system. Thus, we have now added captions to the Supplemental document.

The density current video also seems to contradict your statement in the text about the current not hitting the lateral boundary (the current appears to reach the edge of your frame).

The images and videos do not show the full domain that we modelled over. We have clarified this in the captions.

Fig. S19 is neat!

Thanks, we think so too